# Near-Optimal Streaming Heavy-Tailed Statistical Estimation with Clipped SGD

**Aniket Das***
Stanford University
aniketd@cs.stanford.edu

**Dheeraj Nagaraj**
Google DeepMind
dheerajnagaraj@google.com

**Soumyabrata Pal***
Adobe Research
soumyabratap@adobe.com

**Arun Sai Suggala**
Google DeepMind
arunss@google.com

**Prateek Varshney***
Stanford University
vprateek@stanford.edu

## Abstract

We consider the problem of high-dimensional heavy-tailed statistical estimation in the streaming setting, which is much harder than the traditional batch setting due to memory constraints. We cast this problem as stochastic convex optimization with heavy tailed stochastic gradients, and prove that the widely used Clipped-SGD algorithm attains near-optimal sub-Gaussian statistical rates whenever the second moment of the stochastic gradient noise is finite. More precisely, with $T$ samples, we show that Clipped-SGD, for smooth and strongly convex objectives, achieves an error of $\sqrt{\frac{\mathsf{Tr}(\Sigma)+\sqrt{\mathsf{Tr}(\Sigma)\|\Sigma\|_2}\ln(\ln(T)/\delta)}{T}}$ with probability $1-\delta$, where $\Sigma$ is the covariance of the clipped gradient. Note that the fluctuations (depending on $1/\delta$) are of lower order than the term $\mathsf{Tr}(\Sigma)$. This improves upon the current best rate of $\sqrt{\frac{\mathsf{Tr}(\Sigma)\ln(1/\delta)}{T}}$ for Clipped-SGD, known *only* for smooth and strongly convex objectives. Our results also extend to smooth convex and lipschitz convex objectives. Key to our result is a novel iterative refinement strategy for martingale concentration, improving upon the PAC-Bayes approach of Catoni and Giulini [8].

## 1 Introduction

A fundamental problem in machine learning and statistics is the estimation of an unknown parameter of a probability distribution, given samples from that distribution. This can be expressed as the minimization of the expected loss: $\min_{\mathbf{x}} F(\mathbf{x}) := \mathbb{E}_{\xi \sim P}[f(\mathbf{x}; \xi)]$, where $\mathbf{x}$ represents the parameter to be estimated, $P$ is the underlying probability distribution which can only be accessed through samples, and $f(\mathbf{x}; \xi)$ is a function which quantifies the loss incurred at a point $\xi$ by parameter $\mathbf{x}$. In this paper, we focus on the setting where $P$ is a heavy-tailed distribution for which the extreme values are more likely than in distributions like the Gaussian, $f(\cdot; \cdot)$ is convex and the learner only has access to $O(d)$ memory.

The heavy-tailed statistical estimation problem has received increased attention of late because of the prevalence of heavy-tailed distributions in many statistical applications dealing with real world data [19, 49, 57, 23]. The presence of such heavy-tailed distributions can significantly degrade the performance of statistical estimation and testing procedures designed under Gaussian (or sub-Gaussian) tail assumptions [30, 24, 53, 24]. This has spurred recent research efforts towards developing estimators specifically tailored for heavy-tailed settings (e.g., [10, 44, 16, 36]; see Section 1.2 for a more detailed literature review). Despite substantial progress on this problem in recent years, much of the

---

*Work done while at Google

existing work has concentrated on batch learning, where the entire dataset is available upfront, and the learner can revisit data points multiple times, without memory constraints. However, the streaming setting, where data arrives sequentially and must be processed with limited memory, is increasingly pertinent in the era of large-scale models. Consequently, in this work, we focus on understanding estimators for statistical estimation under heavy-tailed distributions, in the streaming setting.

A popular approach to study heavy-tailed streaming statistical estimation casts it as a stochastic convex optimization (SCO) problem with heavy-tailed gradients [17, 44, 52, 48] - with Clipped-SGD as the favored solution due to its simplicity [42]. Indeed, clipping has become a standard component in the training of modern deep neural networks and thus, the properties of Clipped-SGD have been studied widely in the literature [1, 56, 38, 48, 52] in various contexts. Specifically, several works have shown that Clipped-SGD has sub-Exponential or sub-Gaussian tails despite the presence of heavy tailed noise in the gradient [45, 21, 52, 49]. Despite this progress, the best known rates for Clipped-SGD with smooth and strongly convex losses, under a bounded $2^{\text{nd}}$ moment assumption on gradient distribution, are of the order $\sqrt{\frac{\text{Tr}(\Sigma)\ln(1/\delta)}{T}}$, where $\delta$ is the failure probability [52]. Note that this is still far from the optimal sub-Gaussian rates of $\sqrt{\frac{\text{Tr}(\Sigma)+\|\Sigma\|\ln(1/\delta)}{T}}$. In this work, we bridge this gap with a sharper analysis of Clipped-SGD for SCO problems, achieving nearly sub-Gaussian rates (see Section 1.1). Our approach leverages a novel technique obtained by bootstrapping the Donsker-Varadhan Variational Principle to Freedman's inequality, yielding tighter concentration inequalities for vector martingales compared to those in [8]. This enables us to derive more refined rates for a variety of settings than a direct application of Freedman's inequality as in [52].

## 1.1 Sub-Gaussian Error Guarantees for Statistical Estimation

**Mean Estimation** We motivate our style of results with the case of mean estimation. The Central Limit Theorem (CLT) posits that that the empirical mean of $T$ independent and identically distributed (i.i.d) random variables with a finite covariance, behaves roughly like the empirical mean of Gaussian random variables with the same covariance, as $T \to \infty$. That is, the empirical mean $\hat{\mu}$, the true mean $\mu$ and the covariance $\Sigma$ are such that $\lim_{T\to\infty} \mathbb{P}\left(\sqrt{T}\|\hat{\mu} - \mu\| > \sqrt{\text{Tr}(\Sigma) + \|\Sigma\|_2 \log(\frac{1}{\delta})}\right) \leq \delta$. However, these asymptotic rates need not hold with a practical number of samples. Therefore, recent works on heavy-tailed high dimensional mean estimation consider algorithms and non-asymptotic guarantees which move beyond the empirical mean (see [36, 10, 9, 27, 28, 15]). Estimators such as the clipped mean estimator [8, 55], trimmed mean estimator [45], and the geometric median-of-means estimator [39, 29] achieve an error of at-most $\sqrt{\text{Tr}(\Sigma)\log(\frac{1}{\delta})/T}$ with probability $1 - \delta$ with a finite covariance assumption. Recent ground breaking works [37, 28, 8, 36] further improve upon these results to construct estimators which can achieve the CLT convergence rates of $C\sqrt{\text{Tr}(\Sigma)+\|\Sigma\|_2 \log(\frac{1}{\delta})/T}$ for every $T$ and $\delta$. Some of these estimators work under just the assumption that the second moment is bounded [37, 28, 9] and some even provide a nearly linear time algorithm [15].

**General Statistical Estimation** In this work, we are interested in the general statistical estimation problem. Among the various approaches, framing this problem as SCO with heavy-tailed gradients has gained traction recently (see [52] and references there in). While one obvious candidate is to use SGD with state-of-the-art *optimal* mean estimators for robust gradient estimation, such methods can face significant challenges. First, most optimal mean estimators aren't designed for the streaming data setting with batch-size being $1$. Second, these estimators can be complex, frequently relying on semi-definite programming or other demanding techniques. Third, and perhaps most importantly, they don't typically provide guarantees on the bias of their estimates. This lack of bias control is problematic because SGD-style algorithms, even when equipped with accurate gradient estimates, can perform poorly if those estimates are systematically biased (See [3, Theorem 4], where bias does not cancel across iterations). Given these challenges, the clipped mean estimator of [8] has emerged as a popular choice for gradient estimation in SCO, due mainly to is simplicity. Several recent works analyze the performance of SGD with clipped mean estimator for the gradients (i.e, Clipped SGD). However, as previously mentioned, the best known analysis for clipped SGD achieves a sub-optimal rate of $\sqrt{\text{Tr}(\Sigma)\ln(1/\delta)/T}$, under bounded $2^{\text{nd}}$ moment assumption. In this work, we improve upon these rates and show that with $T$ samples, clipped-SGD obtains a sharper rate of $\frac{\text{Tr}(\Sigma)+\sqrt{\text{Tr}(\Sigma)\|\Sigma\|}\ln(\frac{\ln(T)}{\delta})}{T}$ with probability $1 - \delta$, which is closer to the truly sub-Gaussian rates.

Table 1: Sample complexity bounds (for converging to an $\epsilon$ approximate solution) of various algorithms for SCO under heavy tailed stochastic gradients. Results are instantiated for smooth and strongly convex losses, and for the case where the gradient noise has bounded covariance equal to the Identity matrix. $D_1$ is the distance of the initial iterate from the optimal solution. For readability, we ignore the dependence of rates on the condition number. Observe all prior works have $d\log\delta^{-1}$ dependence in the sample complexity.

| Method | Sample Complexity | Batchsize | Domain |
|---|---|---|---|
| Clipped SGD [21] | $\frac{d}{\epsilon}\left(\log\frac{D_1^2}{\epsilon}\left(\log\delta^{-1}+\log\log\frac{D_1^2}{\epsilon}\right)\right)$ | $O\left(\frac{d}{\epsilon}\log\left(\frac{D_1^2}{\epsilon}\right)\log\left(\frac{1}{\delta}\log\frac{D_1^2}{\epsilon}\right)\right)$ | Unbounded |
| R-Clipped SGD [21] | $\left(\frac{d}{\epsilon}+\log\frac{D_1^2}{\epsilon}\right)\left(\log\delta^{-1}+\log\log\frac{D_1^2}{\epsilon}\right)$ | $O\left(\frac{d}{\epsilon}\log\left(\frac{1}{\delta}\log\frac{D_1^2}{\epsilon}\right)\right)$ | Unbounded |
| R-Clipped SSTM [21] | $\left(\frac{d}{\epsilon}+\log\frac{D_1^2}{\epsilon}\right)\left(\log\delta^{-1}+\log\log\frac{D_1^2}{\epsilon}\right)$ | $O\left(\frac{d}{\epsilon}\log\left(\frac{1}{\delta}\log\frac{D_1^2}{\epsilon}\right)\right)$ | Unbounded |
| RobustGD [45] | $O\left(\frac{d\Phi}{\epsilon}\log\frac{\Phi}{\delta}\right)$ with $\Phi=\log\frac{D_1^2}{\epsilon}$ | $O\left(\frac{d}{\epsilon}\log\frac{\Phi}{\delta}\right)$ | Unbounded |
| proxBoost [14] | $\left(\frac{d}{\epsilon}+\log\frac{D_1^2}{\epsilon}\right)\log\delta^{-1}$ | $O\left(\frac{d}{\epsilon}\log\frac{1}{\delta}\right)$ | Unbounded |
| restarted-RSMD [40] | $\left(\frac{d}{\epsilon}+\log\frac{D_1^2}{\epsilon}\right)\left(\log\delta^{-1}+\log\log\frac{D_1^2}{\epsilon}\right)$ | $O\left(\frac{d}{\epsilon}\left(\log\delta^{-1}+\log\log\frac{D_1^2}{\epsilon}\right)\right)$ | Bounded |
| Clipped SGD [52] | $\left(\frac{d}{\epsilon}+\frac{D_1}{\sqrt{\epsilon}}\right)\log\delta^{-1}$ | 1 | Unbounded |
| Clipped SGD (**Ours**) | $\frac{d+\sqrt{d}\log\delta^{-1}}{\epsilon}+\frac{D_1\log^2(\delta^{-1}\log T)}{\sqrt{\epsilon}}$ | **1** | **Unbounded** |

## 1.2 Related Work

**Clipped SGD** Clipped SGD and it's variants have been studied under a variety of settings including convex, strongly-convex, non-convex losses, with various assumptions on the moments of stochastic gradients. The estimators of [21, 45, 14, 40] work under the assumption of bounded $2^{nd}$ moments, but require $O(1/\epsilon)$ batch size, to converge to an $\epsilon$-approximate solution. Consequently, they are not suitable for streaming setting. The recent work of [52], which is closest to our work, addresses this issue by analysing Clipped-SGD for batch size 1 for smooth, strongly convex losses. But they achieve a sub-optimal rate of $\sqrt{\text{Tr}(\Sigma)\ln(1/\delta)/T}$. These rates are improved in our work (see Table 1 for a detailed comparison). Additionally, our work provides convergence rates for convex objectives that are not strongly convex. Recent works [48, 46, 41, 34, 13] have studied Clipped-SGD with the assumption that the stochastic gradient has a finite $p$-th moment for some $p \in (1, 2]$. They derive fine-grained near optimal results in terms of dependence of $T$ and $p$ (but their dependence on $\log\delta^{-1}$ is sub-optimal). In contrast, our work specifically the case considers $p = 2$ with a focus on improving the sub-Gaussian dependence in the high probability bounds in these works from $\text{Tr}(\Sigma)\log(1/\delta)$ and approaching the truly sub-Gaussian rates for estimation 1.1.

**Heavy-tailed Estimation** Heavy-tailed estimation has a rich history in statistics and we only review some of the recent advances. Several recent works have studied the problem of heavy-tailed mean estimation, and have derived estimators that achieve sub-Gaussian rates under the bounded $2^{\text{nd}}$ moment assumption [36, 10, 9, 27, 28, 15, 45]. Among these, the works of [15, 32] are particularly relevant to our work. The algorithm of [15] runs in linear time while requiring $O(d\log\delta^{-1})$ memory. But it is not immediately clear how to use their estimator in the framework of SGD. [32] study the trimmed mean estimator (an estimator that is closely related to clipped mean estimator, where outliers are removed instead of being clipped) and show that when $T = \omega(\log^3\delta^{-1}), d = \omega(\log^2(\delta^{-1}))$, the estimator achieves the optimal rates. We not that our analysis of clipped SGD, when instantiated for mean estimation, leads to similar rates. But unlike [32] which is primarily focused on mean estimation, we focus on the more general SCO problem.

Heavy-tailed linear regression has been widely studied, with classical estimators based on Huber regression [30, 50, 33] known to provide optimal rates when the response variables are heavy-tailed, but the covariates are light-tailed. Recently, there has been a surge of interest in developing estimators when both covariates and response variables are heavy tailed [5, 44, 17, 43]. However, most of these works are in the batch setting. Another line of work has considered streaming algorithms in the Huber-contamination model, which is a much harder contamination model than heavy-tails [18]. However, these algorithms when adapted to heavy-tailed setting, do not provide optimal rates.

## 1.3 Contributions

**Iteratively Refined Martingale Concentration via PAC Bayes** Our key technical result obtains fine-grained concentration guarantees for vector-valued martingales by using the Donsker-Varadhan Variational Principle to iteratively refine baseline concentration inequalities. This allows us to sharpen the PAC Bayes bounds of Catoni and Giulini [8] (and its martingale based extensions like [11]), which were used to analyze the clipped mean estimator. We believe these iterative refinement arguments could be of independent interest for developing fine-grained concentration bounds.

**Sharp Analysis of Clipped SGD** Leveraging these fine-grained concentration results, We perform a fine-grained analysis of clipped SGD for heavy-tailed SCO problem obtain *nearly* subgaussian performance guarantees in the streaming setting with a batchsize of 1 and $O(d)$ space complexity. In particular, we demonstrate that the sub-optimality gap after $T$ steps scales as $\mathsf{Tr}(\Sigma) + \sqrt{\|\Sigma\|_2 \mathsf{Tr}(\Sigma)} \log(1/\delta)$, improving upon the best known scaling of $\mathsf{Tr}(\Sigma) \log(1/\delta)$ obtained by prior works [52] only for smooth strongly convex problems. To the best of our knowledge, we derive the first such guarantees for smooth convex and lipschitz convex problems in the streaming setting.

**Streaming Heavy Tailed Statistical Estimation** We use the above results to develop streaming estimators for various heavy-tailed statistical estimation problems including heavy-tailed mean estimation as well as linear, logistic and Least Absolute Deviation (LAD) regression with heavy tailed covariates, all of which exhibit nearly subgaussian performance. Our mean estimation results improve upon the previous best known guarantees for trimmed mean based estimators [8, 52, 32] (either in performance or in generality) For heavy-tailed linear regression under the assumption of bounded $4^{\text{th}}$ moments for the covariates and bounded $2^{\text{nd}}$ moments for the response, our rates significantly improve upon that of the previous best known streaming estimator [52]. To the best of our knowledge, we develop the first known streaming estimators for heavy-tailed logistic regression and LAD regression which attain nearly subgaussian rates

## 2 Notation and Organization

We work with Euclidean spaces $\mathbb{R}^d$ equipped with the standard inner product $\langle \cdot, \cdot \rangle$ and the induced $\ell_2$ norm $\| \cdot \|$. For any matrix $A \in \mathbb{R}^{m \times n}$, we use $\|\mathbf{A}\|_2$ to denote its Euclidean operator norm $\|\mathbf{A}\| = \sup_{\mathbf{x} \neq 0} \|\mathbf{A}\mathbf{x}\|/\|\mathbf{x}\|$. For $A \in \mathbb{R}^{d \times d}$, we denote its trace as $\mathsf{Tr}(A)$. For any random vector $\mathbf{x}$, we denote its covariance matrix as $\mathsf{Cov}[\mathbf{x}]$. We use $\lesssim, \gtrsim$ and $\asymp$ to denote $\leq, \geq$ and $=$ respectively, upto universal multiplicative constants. We use $\nabla f(\mathbf{x})$ to denote the gradient of a differentiable function For any convex function $f$, we use $\partial f(\mathbf{x})$ to denote an arbitrary subgradient of $f$ at $\mathbf{x}$.

## 3 Background and Problem Formulation

Our work studies the Stochastic Convex Optimization (SCO) problem, described as follows: Let $\mathcal{C}$ denote a closed convex subset of $\mathbb{R}^d$ and let $F : \mathcal{C} \rightarrow \mathbb{R}$ be a convex function. We aim to solve:

$$\min_{\mathbf{x} \in \mathcal{C}} F(\mathbf{x}), \tag{SCO}$$

assuming access to a convex projection oracle $\Pi_\mathcal{C}$ and a *stochastic gradient oracle*, which we define as follows: Let $P$ denote a probability measure supported on an arbitrary domain $\Xi$ from which we can draw samples. A stochastic gradient oracle for $F$ is a function $g : \mathcal{C} \times \mathcal{C}$, which, given a point $\mathbf{x} \in \mathcal{C}$ and a sample $\xi \sim P$ returns an unbiased estimate $g(\mathbf{x}; \xi)$ of $\nabla F(\mathbf{x})$ i.e., $\mathbb{E}_{\xi \sim P}\left[g(\mathbf{x}; \xi)\right] = \nabla F(\mathbf{x})$. If $F$ is nondifferentiable, $\mathbb{E}_{\xi \sim P}\left[g(\mathbf{x}; \xi)\right] = \partial F(\mathbf{x})$. Note that we do not assume direct access to $\nabla F(\mathbf{x})$, which may be expensive or intractable to compute. Our objective is to (approximately) solve SCO subject to a constraint on the number of samples we can draw from $P$.

This is an alternative formulation of the statistical estimation problem by recognizing $P$ as the data distribution, $\mathcal{C}$ as the parameter space and defining the population risk $F(\mathbf{x}) := \mathbb{E}_{\xi \sim P}[f(\mathbf{x}; \xi)]$, where $f$ denotes the sample-level loss function. The associated stochastic gradient oracle is $g(\mathbf{x}; \xi) := \nabla f(\mathbf{x}; \xi)$, $\xi \sim P$, which is usually easy to compute. As we shall discuss in Section 5, several statistical estimation problems such as mean estimation, linear regression, logistic regression and least absolute deviation regression naturally fit into the SCO framework.

We use $\mathbf{n}(\mathbf{x};\xi) = \mathbf{g}(\mathbf{x};\xi) - \nabla F(\mathbf{x})$ to denote the *stochastic gradient noise* and assume it has *finite second moment*, i.e., $\Sigma(\mathbf{x}) = \mathbb{E}_{\xi \sim P}[\mathbf{n}(\mathbf{x};\xi)\mathbf{n}(\mathbf{x};\xi)^T]$ exists for every $\mathbf{x} \in \mathcal{C}$. Our results make use of either of the following assumptions on $\Sigma(\mathbf{x})$.

**Assumption 1** (Bounded Second Moment). *The exists a positive semidefinite matrix $\Sigma$ such that:*

$$\Sigma(\mathbf{x}) \preceq \Sigma \quad \forall\, \mathbf{x} \in \mathcal{C} \qquad \text{(Bdd. 2$^\text{nd}$ Moment)}$$

Similar assumption has been made by several prior works [21, 40, 14, 45]. We also consider the following generalized assumption, which is as a refinement of the one made in Tsai et al. [52].

**Assumption 2** (Second Moment with Quadratic Growth). *There exist constants $\alpha, \beta \geq 0$ and $1 \leq d_{\text{eff}} d$ such that the following holds for every $\mathbf{x} \in \mathcal{C}$*

$$\|\Sigma(\mathbf{x})\|_2 \leq \alpha \|\mathbf{x} - \mathbf{x}^*\|^2 + \beta; \qquad \text{Tr}(\Sigma(\mathbf{x})) \leq d_{\text{eff}}\left(\alpha \|\mathbf{x} - \mathbf{x}^*\|^2 + \beta\right) \qquad \text{(QG 2$^\text{nd}$ Moment)}$$

*where $\mathbf{x}^*$ denotes any arbitrary minimizer of $F$.*

Since we consider streaming statistical estimators that are robust to heavy tailed data, we only assume the existence of the second moment of the stochastic gradient noise and *allow its higher moments to be infinite*. That is, our results hold even when $\mathbb{E}_{\xi \sim P}[|\langle \mathbf{n}(\mathbf{x};\xi), \mathbf{v}\rangle|^{2+\epsilon}] = \infty$ for every $\epsilon > 0, \mathbf{v} \in \mathbb{R}^d$

Our work analyzes SCO under either of the following structural assumptions assumptions on $F$

**Assumption 3** (Convexity). *$F : \mathbb{R}^d \to \mathbb{R}$ is a convex function if the following holds for any $t \in [0,1]$*

$$F(t\mathbf{x} + (1-t)\mathbf{y}) \leq tF(\mathbf{x}) + (1-t)F(\mathbf{y}) \quad \forall\, \mathbf{x}, \mathbf{y} \in \mathbb{R}^d \qquad \text{(Convexity)}$$

**Assumption 4** ($\mu$-Strong Convexity). *$F : \mathbb{R}^d \to \mathbb{R}$ is a $\mu$-strongly convex function for $\mu \geq 0$ if the following holds for every $t \in [0,1]$*

$$F(t\mathbf{x} + (1-t)\mathbf{y}) \leq tF(\mathbf{x}) + (1-t)F(\mathbf{y}) - t(1-t)\cdot\tfrac{\mu}{2}\|\mathbf{x} - \mathbf{y}\|^2 \quad \forall\, \mathbf{x}, \mathbf{y} \in \mathbb{R}^d \quad \text{($\mu$-Strong Convexity)}$$

In addition, we also consider either of the two regularity assumptions on $F$

**Assumption 5** ($L$-smoothness). *$F : \mathbb{R}^d \to \mathbb{R}$ is $L$-smooth for some $L \geq 0$ if $F$ is continuously differentiable and satisfies the following:*

$$\|\nabla F(\mathbf{x}) - \nabla F(\mathbf{y})\| \leq L\|\mathbf{x} - \mathbf{y}\| \quad \forall\, \mathbf{x}, \mathbf{y} \in \mathbb{R}^d \qquad \text{($L$-smoothness)}$$

**Assumption 6** ($G$-Lipschitzness). *$F : \mathbb{R}^d \to \mathbb{R}$ is $G$-Lipschitz for some $G \geq 0$, i.e., $F$ is continuous and satisfies the following:*

$$\|F(\mathbf{x}) - F(\mathbf{y})\| \leq G\|\mathbf{x} - \mathbf{y}\| \quad \forall\, \mathbf{x}, \mathbf{y} \in \mathbb{R}^d \qquad \text{($G$-Lipschitzness)}$$

# 4 Results

Under the Bdd. 2$^\text{nd}$ Moment and QG 2$^\text{nd}$ Moment assumptions, streaming algorithms for SCO such as Stochastic Gradient Descent (SGD) typically convergence bounds guarantees that hold in expectation [56, 26, 22]. However, high probability guarantees require strong assumptions on the tail behavior of the stochastic gradients (e.g. boundedness or subgaussianity) [25, 47, 31]. Our work analyzes SCO under heavy tailed stochastic gradients, which typically exhibit large fluctuations from their expected value due to its higher order moments being potentially infinite. Clipped SGD mitigates the large fluctuations typically observed in the heavy tailed stochastic gradient $g(\mathbf{x};\xi)$ by thresholding its norm as follows. The full algorithm is described in Algorithm 1.

$$\text{clip}_\Gamma(g(\mathbf{x};\xi)) := \frac{g(\mathbf{x};\xi)}{\|g(\mathbf{x};\xi)\|} \cdot \min\{\Gamma, \|g(\mathbf{x};\xi)\|\}$$

We now present our performance guarantees for clipped SGD for streaming heavy tailed SCO, wherein Algorithm 1 is subject to an $O(d)$ memory constraint and can access only one stochastic gradient sample per iteration. For the remainder, of this section, we use $\mathbf{x}^* \in \mathcal{C}$ to denote an arbitrary minimizer of $F$, which is assumed to always exist, and guaranteed to be unique if $F$ satisfies $\mu$-Strong Convexity. We use $\mathbf{x}_1$ to denote the initialization of Algorithm 1 and let $D_1 = \|\mathbf{x} - \mathbf{x}^*\|$.

---

**Algorithm 1** Clipped Stochastic Gradient Descent

---

**Input**: Initialization $\mathbf{x}_1$, Horizon $T$, Step Sizes $(\eta_t)_{t\in[T]}$, Clipping Level $\Gamma$

1: **for** $t \in [T]$ **do**
2:    $\mathbf{g}_t \leftarrow g(\mathbf{x}_t; \xi_t), \quad \xi_t \sim P$
3:    $\mathbf{x}_{t+1} \leftarrow \Pi_{\mathcal{C}}(\mathbf{x}_t - \eta_t \cdot \mathsf{clip}_\Gamma(\mathbf{g}_t))$
4: **end for**
5: **Last Iterate :** Output $\mathbf{x}_{T+1}$
6: **Average Iterate :** Output $\hat{\mathbf{x}}_T = \frac{1}{T}\sum_{t=1}^T \mathbf{x}_t$

---

### 4.1 Smooth Strongly Convex Objectives

Theorems 1 and 2, proved in, Appendix B and C respectively, derive high probability convergence bounds for smooth and strongly convex objectives with second moment assumption.

**Theorem 1** (Smooth Strongly Convex Objectives). *Let the L-smoothness, $\mu$-Strong Convexity and Bdd. $2^{\text{nd}}$ Moment assumptions be satisfied. Then, for any $\delta \in (0, 1/2)$, the last iterate of Algorithm 1 run for $T \gtrsim \ln(\ln(d))$ iterations with stepsize $\eta_t = \frac{4}{\mu(t+\gamma)}$ and clipping level $\Gamma = \frac{\mu}{\ln(\ln(T)/\delta)}\sqrt{(\gamma+1)^2 D_1^2 + \frac{(T+\gamma)}{\mu^2}(\mathsf{Tr}(\Sigma) + \sqrt{\mathsf{Tr}(\Sigma)\|\Sigma\|_2}\ln(\ln(T)/\delta))}$ satisfies the following with probability at least $1 - \delta$*

$$\|\mathbf{x}_{T+1} - \mathbf{x}^*\| \lesssim \frac{\gamma D_1}{T+\gamma} + \frac{1}{\mu}\sqrt{\frac{\mathsf{Tr}(\Sigma) + \sqrt{\mathsf{Tr}(\Sigma)\|\Sigma\|_2}\ln(\ln(T)/\delta)}{T+\gamma}} \tag{1}$$

*where* $\gamma \asymp \max\{\frac{\|\Sigma\|_2 \kappa^2 \ln(\ln(T)/\delta)^2}{\mathsf{Tr}(\Sigma)}, \kappa^{3/2}\ln(\ln(T)/\delta), \kappa\ln(\ln(T)/\delta)^2\}$

We use Theorem 1 to derive sharp rates for streaming heavy tailed mean estimation in Section 5.1 and the following result to derive sharp rates for streaming heavy tailed linear regression in section 5.2

**Theorem 2** (Smooth Strongly Convex Objectives with Quadratic Growth Noise Model). *Let Assumptions $\mu$-Strong Convexity, L-smoothness and QG $2^{\text{nd}}$ Moment be satisfied and let $\kappa = L/\mu$. For any $\delta \in (0, 1/2)$, the last iterate of Algorithm 1 run for $T \gtrsim \ln(\ln(d))$ iterations with step-size $\eta_t = \frac{4}{\mu(t+\gamma)}$ and clipping level $\Gamma = \frac{\mu}{\ln(\ln(T)/\delta)}\sqrt{(\gamma+1)^2 D_1^2 + \frac{\beta}{\mu^2}\cdot(T+\gamma)(d_{\mathsf{eff}} + \sqrt{d_{\mathsf{eff}}}\ln(\ln(T)/\delta))}$ satisfies the following with probability at least $1 - \delta$*

$$\|\mathbf{x}_{T+1} - \mathbf{x}^*\| \lesssim \frac{\gamma D_1}{T+\gamma} + \frac{1}{\mu}\sqrt{\frac{\beta(d_{\mathsf{eff}} + \sqrt{d_{\mathsf{eff}}}\ln(\ln(T)/\delta))}{T+\gamma}} \tag{2}$$

*where* $\gamma \asymp \max\{\frac{\alpha d_{\mathsf{eff}}}{\mu^2}, \frac{\alpha\sqrt{d_{\mathsf{eff}}}}{\mu^2}\ln(\ln(T)/\delta), \frac{\kappa\sqrt{\alpha}}{\mu}\ln(\ln(T)/\delta), \frac{\sqrt{\kappa\alpha d_{\mathsf{eff}}}}{\mu}\ln(\ln(T)/\delta),$
$\frac{\kappa^{2/3}\alpha^{1/3}d_{\mathsf{eff}}^{1/3}}{\mu^{2/3}}\ln(\ln(T)/\delta), \kappa^{3/2}\ln(\ln(T)/\delta), \kappa\ln(\ln(T)/\delta)^2, \frac{\kappa^2}{d_{\mathsf{eff}}}\ln(\ln(T)/\delta)\}$

**Comparison to Prior Works** To the best of our knowledge, the result closest to Theorem 2 is [52, Theorem 1] which analyzes streaming strongly convex SCO and obtains a $\frac{\zeta D_1}{T+\zeta} + \frac{1}{\mu}\sqrt{\frac{\beta d_{\mathsf{eff}}\ln(1/\delta)}{T+\zeta}}$ rate for $\zeta \asymp \frac{\alpha d_{\mathsf{eff}}\log(1/\delta)}{\mu^2}$. We note that Theorem 2 obtains a significantly better confidence bound which is closer to the optimal subgaussian rate compared [52, Theorem 1].

**Extra $\log\log T$ term:** Our bounds for the statistical error is of the form $\frac{1}{\mu}\sqrt{\frac{\beta(d_{\mathsf{eff}} + \sqrt{d_{\mathsf{eff}}}\ln(\ln(T)/\delta))}{T+\gamma}}$ which has an extra $\log\log T$ factor in the lower order term. This is still sharper than prior works with bounds of the form $\frac{1}{\mu}\sqrt{\frac{\beta d_{\mathsf{eff}}\ln(1/\delta)}{T+\gamma}}$ as long as $\log\log T \ll \sqrt{d_{\mathsf{eff}}}\log(\frac{1}{\delta})$.

### 4.2 Beyond Strongly Convex Objectives

Moving beyond strong convexity, we present Theorems 3 for smooth convex functions and 4 for Lipschitz convex function, proved in Appendix D and E respectively. To the best of our knowledge,

these are the first results for streaming heavy-tailed convex SCO that exhibits near-subgaussian concentration without strong convexity.

**Theorem 3** (Smooth Convex Objectives). *Let Convexity, L-smoothness and Bdd. $2^{\text{nd}}$ Moment be satisfied. Then, for any $\delta \in (0, 1/2)$ and $T \geq \ln(\ln(d))$, there exists an $\eta \in (0, 1/2L)$ such that the average iterate of Algorithm 1 run for $T$ iterations with step-size $\eta_t = \eta$ and clipping level $\Gamma = \sqrt{\frac{T\sqrt{\|\Sigma\|_2}(\sqrt{\text{Tr}(\Sigma)} + LD_1)}{\ln(\ln(T)/\delta)}}$ satisfies the following with probability at least $1 - \delta$:*

$$F(\hat{\mathbf{x}}_T) - F(\mathbf{x}^*) \lesssim \frac{LD_1^2}{T} + D_1\sqrt{\frac{\text{Tr}(\Sigma) + \sqrt{\|\Sigma\|_2}\left(\sqrt{\text{Tr}(\Sigma)} + LD_1\right)\ln(\ln(T)/\delta)}{T}} + o_T(L, D_1, \Sigma)$$

*where $o_T(L, D_1, \Sigma)$ represents terms that are of lower order in $T$ (explicated in Appendix D)*

**Theorem 4** (Lipschitz Convex Objectives). *Let Assumptions Convexity, G-Lipschitzness and Bdd. $2^{\text{nd}}$ Moment be satisfied. Then, for any $\delta \in (0, 1/2)$ and $T \geq \ln(\ln(d))$, there exists an $\eta \in (0, G/\sqrt{T}]$ such that the average iterate of Algorithm 1 run for $T$ iterations with step-size $\eta_t = \eta$ and clipping level $\Gamma = \sqrt{\frac{T\sqrt{\|\Sigma\|_2}(\sqrt{\text{Tr}(\Sigma)} + G)}{\ln(\ln(T)/\delta)}}$ satisfies the following with probability at least $1 - \delta$*

$$F(\hat{\mathbf{x}}_T) - F(\mathbf{x}^*) \lesssim \frac{D_1G}{\sqrt{T}} + D_1\sqrt{\frac{\text{Tr}(\Sigma) + \sqrt{\|\Sigma\|_2}\left(\sqrt{\text{Tr}(\Sigma)} + G\right)\ln(\ln(T)/\delta)}{T}} + o_T(G, D_1, \Sigma)$$

*where $o_T(G, D_1, \Sigma)$ represents terms that are lower order in $T$ (explicated in Appendix E)*

**Remark:** We use Theorem 3 to design the first known streaming estimator for logistic regression with heavy-tailed covariates in Section 5.3 and Theorem 4 to design the first known streaming estimator for LAD regression with heavy-tailed covariates in Section 5.4.

**Remark:** In Theorems 3 and 4, the leading order term in the error is of the form: $D_1\sqrt{\frac{\text{Tr}(\Sigma) + \sqrt{\|\Sigma\|_2}\left(\sqrt{\text{Tr}(\Sigma)} + \zeta\right)\ln(\ln(T)/\delta)}{T}}$, where $\zeta \in \{G, LD_1\}$. Assuming $G, D_1, \sqrt{\text{Tr}(\Sigma)} \asymp \sqrt{d}$, we note that the term dependent on the confidence level $\log(1/\delta)$ is lower order compared to $\text{Tr}(\Sigma)$. To the best of our knowledge, this is the first work which establishes strong confidence bounds in the setting of SCO without strong convexity. Interestingly, our results also improve the best known rates for sub-Gaussian gradient noise. To be precise, [35, Theorem 3.1] shows a *weaker* bound of $\sqrt{D_1^2(G^2 + \text{Tr}(\Sigma)\log(\frac{1}{\delta}))/T}$ in the setting of Theorem 4, but when the noise is sub-Gaussian.

# 5 Applications to Streaming Heavy Tailed Statistical Estimation

## 5.1 Streaming Heavy-Tailed Mean Estimation

Consider streaming heavy tailed mean estimation with clipped SGD with access to $N$ i.i.d samples from the distribution $P$. Let $\Xi = \mathcal{C}$, $\mathbb{E}_{\xi \sim P}[\xi] = \mathbf{m} \in \mathcal{C}$. We further assume $\text{Cov}[\xi] \preceq \Sigma$ and allow the higher moments to be infinite. As described in Appendix G.1, this is an SCO problem with the sample loss $f(\mathbf{x}; \xi) = \frac{1}{2}\|\mathbf{x} - \xi\|^2$. The population loss and the stochastic gradient are given by:

$$F(\mathbf{x}) = \frac{1}{2}\|\mathbf{x} - \mathbf{m}\|^2 + \text{Tr}(\text{Cov}_{\xi \sim P}[\xi]); \qquad g(\mathbf{x}; \xi) = \mathbf{x} - \xi$$

The following result, proved in Appendix G.1 via an application of Theorem 1, shows that the last iterate of clipped SGD attains near-subgaussian rates for the heavy tailed mean estimation problem

**Corollary 1** (Heavy Tailed Mean Estimation). *Under the stochastic gradient oracle described above, implemented using $N \gtrsim \ln(\ln(d))$ i.i.d samples $\xi_1, \ldots, \xi_N \sim P$, the last iterate of Algorithm 1 when run under the parameter settings of Theorem 1 satisfies the following with probability at least $1 - \delta$*

$$\|\mathbf{x}_{N+1} - \mathbf{m}\| \lesssim \frac{\gamma\|\mathbf{x}_1 - \mathbf{m}\|}{N + \gamma} + \sqrt{\frac{\text{Tr}(\Sigma) + \sqrt{\|\Sigma\|_2 \text{Tr}(\Sigma)}\ln(\ln(N)/\delta)}{N + \gamma}}$$

*where $\gamma \asymp \ln(\ln(N)/\delta)^2$*

**Comparison to Prior Works** The clipped mean estimator of [8] and the clipped-SGD based estimator in [52] come with a guarantee of the form $\|\hat{\mathbf{m}} - \mathbf{m}\| \lesssim \sqrt{\mathsf{Tr}(\Sigma)\log(\frac{1}{\delta})/N}$ with probability $1 - \delta$. Our result in Corollary 1 obtains a sharper rate of convergence. In a recent work, Lee and Valiant [32] showed that the trimmed mean estimator achieves the optimal rate of $\sqrt{\mathsf{Tr}(\Sigma)/N}$ when $N = \omega(\log^3 \delta^{-1}), d = \omega(\log^2(\delta^{-1}))$. Our result matches this optimal rate in those settings, but is considerably more general, as it holds for any $N, d$.

## 5.2 Streaming Heavy Tailed Linear Regression

In the current and subsequent sections, we use $\theta \in \mathcal{C}$ to denote the parameter of $F$. Let $\Xi = \mathbb{R}^d \times \mathbb{R}$. Given a target parameter $\theta^* \in \mathcal{C}$, $P$ defines the following linear model:

$$\mathbf{x} \sim Q, \ \mathbb{E}[\mathbf{x}] = 0, \ \mathbb{E}[\mathbf{x}\mathbf{x}^T] = \Sigma \succ 0; \qquad y = \langle \mathbf{x}, \theta^* \rangle + \epsilon, \ \mathbb{E}[\epsilon|\mathbf{x}] = 0, \ \mathbb{E}[\epsilon^2|\mathbf{x}] \le \sigma^2$$

In addition, we make the following bounded $4^{\text{th}}$ moment asumption on the covariates $\mathbf{x}$

$$\mathbb{E}[\langle \mathbf{x}, \mathbf{v} \rangle^4] \le C_4(\mathbb{E}[\langle \mathbf{x}, \mathbf{v} \rangle^2])^2 \qquad \forall \, \mathbf{v} \in \mathbb{R}^d$$

for some numerical constant $C_4 \ge 1$. Note that we allow both the covariate $\mathbf{x}$ and the target $\mathbf{y}$ to be heavy tailed, assuming only finite moments of upto order $4$ for $\mathbf{x}$ and order $2$ for $\mathbf{y}$. The assumption $\mathbb{E}[\mathbf{x}] = 0$ is only made for ease of presentation and our arguments easily adapt to $\mathbb{E}[\mathbf{x}] \ne 0$. Our task is to estimate $\theta^*$ in a streaming fashion with access to $N$ i.i.d samples from $P$. As described in Appendix G.2, we reframe this problem as SCO under the sample loss $f(\theta; \mathbf{x}, y) = \frac{1}{2}(\langle \mathbf{x}, \theta \rangle - y)^2$. The associated population loss $F(\theta)$ and the stochastic gradient oracle $g(\theta; \mathbf{x}, y)$ are given by:

$$F(\theta) = \frac{1}{2}(\theta - \theta^*)^T \Sigma (\theta - \theta^*); \qquad g(\theta; \mathbf{x}, \mathbf{y}) = (\langle \mathbf{x}, \theta \rangle - y)\mathbf{x}$$

**Corollary 2** (Heavy Tailed Linear Regression). *Under the stochastic gradient oracle described above, implemented using $N \gtrsim \ln(\ln(d))$ i.i.d samples from $P$, the last iterate of Algorithm 1 when run under the parameter settings of Theorem 2 satisfies the following with probability at least $1 - \delta$:*

$$\|\theta_{N+1} - \theta^*\| \lesssim \frac{\gamma \|\theta_1 - \theta^*\|}{N + \gamma} + \frac{\sigma}{\lambda_{\min}(\Sigma)} \sqrt{\frac{\mathsf{Tr}(\Sigma) + \sqrt{\|\Sigma\|_2 \mathsf{Tr}(\Sigma)} \ln(\ln(N)/\delta)}{N + \gamma}}$$

*where $\gamma \asymp \max \left\{ \frac{C_4 \kappa^2 \mathsf{Tr}(\Sigma)}{\|\Sigma\|_2}, C_4 \kappa^2 \sqrt{\frac{\mathsf{Tr}(\Sigma)}{\|\Sigma\|_2}} \ln(\ln(N)/\delta), \kappa \ln(\ln(N)/\delta)^2 \right\}$ and $\kappa = \frac{\|\Sigma\|_2}{\lambda_{\min}(\Sigma)}$*

To the best of our knowledge, [52, Corollary 4] is the only other streaming estimator for this problem with subgaussian-style concentration. Our result above significantly improves upon their rates of $\frac{\|\theta_1 - \theta^*\|}{N + \zeta} + \frac{\sigma}{\lambda_{\min}(\Sigma)} \sqrt{\frac{\|\Sigma\|_2 d \ln(1/\delta)}{N + \zeta}}$ with $\zeta = C_4 d\kappa^2 \ln(1/\delta)$. Furthermore, our result is much closer to the optimal subgaussian rate and gracefully adapts to the *stable rank* or effective dimension [32], i.e., $d_{\text{eff}} = \mathsf{Tr}(\Sigma)/\|\Sigma\|$, therefore implying significant speedups over [52] in settings where $d_{\text{eff}} \ll d$.

## 5.3 Streaming Heavy Tailed Logistic Regression

Let $\Xi = \mathbb{R}^d \times \{0, 1\}$ and given a target parameter $\theta^* \in \mathcal{C}$, $P$ denote the following linear-logistic model:

$$\mathbf{x} \sim Q, \ \mathbb{E}[\mathbf{x}] = 0, \ \mathbb{E}[\mathbf{x}\mathbf{x}^T] \preceq \Sigma; \qquad y \sim \mathsf{Bernoulli}(\phi(\langle \theta^*, \mathbf{x} \rangle))$$

where $\phi(t) = (1 + e^{-t})^{-1}$. The covariates $\mathbf{x}$ are heavy tailed, with only bounded second moments. The negative log likelihood of $y|\mathbf{x}$ is given by $f(\theta; \mathbf{x}, y) = \ln(1 + \exp(\langle \mathbf{x}, \theta \rangle)) - y \langle \mathbf{x}, \theta \rangle$. The objective of the logistic regression problem is to estimate $\theta^*$ by minimizing the population-level negative log likelihood:

$$F(\theta) = \mathbb{E}_{\mathbf{x}, y \sim P} \left[ \ln(1 + \exp(\langle \mathbf{x}, \theta \rangle)) - y \langle \mathbf{x}, \theta \rangle \right]$$

which is minimized at $\theta^*$. Here, the stochastic gradient oracle is $g(\theta; \mathbf{x}, \mathbf{y}) = \phi(\langle \mathbf{x}, \theta \rangle)\mathbf{x} - y\mathbf{x}$. The following result applies Theorem 3 to show that the output of clipped SGD attains near-subgaussian rates for heavy tailed logistic regression. We refer to Appendix G.3 for the proof.

**Corollary 3** (Heavy Tailed Logistic Regression). *Under the stochastic subgradient oracle described above, realized using $N \gtrsim \ln(\ln(d))$ i.i.d samples from P, the average iterate of Algorithm 1, when run under the parameter settings of Theorem 4 satisfies the following with probability at least $1 - \delta$:*

$$F(\hat{\theta}_N) - F(\theta^*) \lesssim D_1 \sqrt{\frac{\mathsf{Tr}(\Sigma) + \sqrt{\|\Sigma\|_2} \left( \sqrt{\mathsf{Tr}(\Sigma)} + \|\Sigma\|_2 D_1 \right) \ln\left(\ln(N)/\delta\right)}{N}} + o_N(\Sigma, D_1)$$

*where $o_N(\Sigma, D_1)$ represents terms that are lower order in N (explicated in Appendix G.3*

Note that the standard analysis of SGD, with the assumption that $\|\mathbf{x}\| \leq R$ almost surely leads to a bound of the form [4, Proposition 5]: $F(\hat{\theta}_N) - F(\theta^*) \lesssim \frac{R D_1 \sqrt{\log(\frac{1}{\delta})}}{\sqrt{N}}$

## 5.4 Streaming Heavy Tailed LAD Regression

Let $\Xi = \mathbb{R}^d \times \mathbb{R}$. Given a target parameter $\theta^* \in \mathcal{C}$, P defines the following linear model:

$$\mathbf{x} \sim Q, \ \mathbb{E}[\mathbf{x}] = 0, \ \mathbb{E}[\mathbf{x}\mathbf{x}^T] \preceq \Sigma; \qquad y = \langle \mathbf{x}, \theta^* \rangle + \epsilon, \ \mathsf{Median}(\epsilon|\mathbf{x}) = 0$$

We allow both the covariate $\mathbf{x}$ and target $y$ to be heavy tailed, assuming only bounded second moments for $\mathbf{x}$. We do not assume any moment bounds on $\epsilon|\mathbf{x}$. The assumption $\mathbb{E}[\mathbf{x}] = 0$ is made for the sake of clarity and can be straightforwardly relaxed. The Least Absolute Deviation (LAD) Regression problem involves estimating $\theta$ by solving SCO with the sample loss $f(\theta; \mathbf{x}, y) = |\langle \mathbf{x}, \theta \rangle - y|$. The stochastic subgradient oracle and population risk is given by:

$$g(\theta; \mathbf{x}, \mathbf{y}) = \mathsf{sgn}(\langle \theta, \mathbf{x} \rangle - \mathbf{y})\mathbf{x}, \qquad F(\theta) = \mathbb{E}\left[ |\langle \theta - \theta^*, \mathbf{x} \rangle - \epsilon| \right]$$

where $\mathsf{sgn}(t) = \frac{t}{\|t\|}$ for $t \neq 0$ and $\mathsf{sgn}(0) = 0$. The following result, whose full statement and proof is presented in Appendix G.4, applies Theorem 4 to show that the average iterate of clipped SGD attains near-subgaussian rates for heavy tailed LAD regression. To the best of our knowledge, this is the first known streaming estimator for this problem.

**Corollary 4** (Heavy Tailed LAD Regression). *Under the stochastic subgradient oracle described above, realized using $N \gtrsim \ln(\ln(d))$ i.i.d samples from P, the average iterate of Algorithm 1, when run under the parameter settings of Theorem 4 satisfies the following with probability at least $1 - \delta$:*

$$F(\hat{\theta}_N) - F(\theta^*) \lesssim D_1 \sqrt{\frac{\mathsf{Tr}(\Sigma) + \sqrt{\|\Sigma\|_2 \mathsf{Tr}(\Sigma)} \ln\left(\ln(N)/\delta\right)}{N}} + o_N(\Sigma, D_1)$$

*where $o_N$ denotes terms that are lower order in N (explicated in Appendix G.4)*

## 6 Improved Martingale Concentration via Iterative Refinement

Our results are based on the following concentration result for $\mathbb{R}^d$ valued martingales. The proof appears in Appendix F. Suppose $M_t$ for $t = 0, \ldots, T$ is an $\mathbb{R}^d$ valued martingale such that $M_0 = 0$ almost surely, the difference sequence $\mathbf{v}_t := M_t - M_{t-1}$ is such that $\|\mathbf{v}_t\| \leq \Gamma$ and $\mathbb{E}[\mathbf{v}_t \mathbf{v}_t^T | \mathcal{F}_{t-1}] = \Sigma_t$ almost surely for every $t = 1, \ldots, T$ for some $\Gamma > 0$. Assume that there exist deterministic sequences $p_1, \ldots, p_T$ and $q_1, \ldots, q_T$ such that $\mathsf{Tr}(\Sigma_t) \leq q_t$ and $\|\Sigma_t\| \leq p_t$ almost surely.

**Theorem 5.** *Let $\bar{q} := \frac{1}{T} \sum_{t=1}^{T} q_t$ and $\bar{p} := \frac{1}{T} \sum_{t=1}^{T} p_t$. Then, for any $\delta \in (0, \frac{1}{2})$:*

$$\mathbb{P}(\sup_{t \leq T} \|M_t\| \geq g(T, \delta)\sqrt{T}) \leq \delta$$

*Where $g(T, \delta) = C_M \left[ \sqrt{\bar{q}} + \frac{\bar{p}\sqrt{T}}{\Gamma} + \frac{\Gamma}{\sqrt{T}} \log(\frac{K}{\delta}) \right]$ and $K = \ln\ln((\frac{\sqrt{\bar{q}T}}{\Gamma} + 1)\log(d + 1)) + C_M$ for some universal constant $C_M$*

To prove this result, we first use Freedman's inequality [20, 51] to obtain a coarse-grained $g_0$ such that $\mathbb{P}(\sup_t \|M_t\| > g_0\sqrt{T}) \leq \delta$. We then iteratively refine this inequality via a PAC Bayesian [8, 11, 12] argument to show that $\mathbb{P}(\sup_t \|M_t\| > g_{k+1}\sqrt{T} \mid \mathcal{B}_k) \leq \delta$, where $\mathcal{B}_k = \{\sup_t \|M_t\| \leq g_k\sqrt{T}\}$ and $g_{k+1}^2 \lesssim \mathsf{Tr}(\Sigma) + g_k\sqrt{\|\Sigma_2\| \log(1/\delta)}$. This iterative refinement strategy, proved in Theorem 14 is one of the main technical contributions of our work, which could be of independent interest. We arrive at Theorem 5 after $K \approx \log\log(T \log d)$ refinement steps.

**Remark** Theorem 5 is used to control the influence of the fluctuations introduced by clipped SGD. To this end, let $\mathbf{v}_t$ be the centered version of $\text{clip}_\Gamma(\mathbf{g}_t)$, ensuring $\|\mathbf{v}_t\| \leq 2\Gamma$ almost surely. Suppose $\Sigma_t = \Sigma$ for some fixed $\Sigma$ and let $\Gamma = \sqrt{\|\Sigma\|T/\log(\frac{K}{\delta})}$. Then, with probability $1 - \delta$: $\sup_{t \leq T} \|M_t\| \lesssim \sqrt{T\text{Tr}(\Sigma) + T\|\Sigma\|\log(\frac{K}{\delta})}$. This is sharper than the $\sup_{t \leq T} \|M_t\| \lesssim \sqrt{T\text{Tr}(\Sigma)\log(\frac{d}{\delta})}$ guarantee implied by the Matrix Freedman inequality [51, Corollary 1.6].

## 7 Proof Sketch

We sketch our proof technique for the case of smooth convex functions considered in 3. We consider the SGD iterations $\mathbf{x}_1, \ldots, \mathbf{x}_T$ with clipped stochastic gradient at time $t$ denoted by $\text{clip}_\Gamma(\mathbf{g}_t) = \nabla F(\mathbf{x}_t) + \mathbf{v}_t + \mathbf{b}_t$. Here, $\mathbf{v}_t$ is the zero mean 'variance' such that $\mathbb{E}[\mathbf{v}_t|\mathbf{x}_t] = 0$ and $\|\mathbf{v}_t\| \leq 2\Gamma$ almost surely. $\mathbf{b}_t$ is the non-zero mean 'bias' which arises due to clipping. Using the usual analysis of SGD for convex functions (see for instance [31]), we consider:

$$\|\mathbf{x}_{t+1} - \mathbf{x}^*\|^2 \leq \|\mathbf{x}_t - \mathbf{x}^*\|^2 - 2\eta_t[F(\mathbf{x}_t) - F(\mathbf{x}^*)] - 2\eta_t\langle\mathbf{v}_t + \mathbf{b}_t, \mathbf{x}_t - \mathbf{x}^*\rangle + \eta_t^2\|\nabla F(\mathbf{x}_t) + \mathbf{v}_t + \mathbf{b}_t\|^2$$

Considering constant step-sizes, we sum the inequalities for each $t$ to conclude:

$$\frac{1}{T}\sum_{t=1}^{T} F(\mathbf{x}_t) - F(\mathbf{x}^*) \leq \frac{1}{2\eta T}\|\mathbf{x}_1 - \mathbf{x}^\star\|^2 + \frac{1}{T}\sum_{t=1}^{T}\langle\mathbf{v}_t + \mathbf{b}_t, \mathbf{x}_t - \mathbf{x}^*\rangle$$
$$+ \frac{3\eta}{2T}\sum_t[\|\nabla F(\mathbf{x}_t)\|^2 + \|\mathbf{v}_t\|^2 + \|\mathbf{b}_t\|^2] \tag{3}$$

The 'random' terms to bound compared to gradient descent here are $\sum_t\langle\mathbf{v}_t + \mathbf{b}_t, \mathbf{x}_t - \mathbf{x}^*\rangle$ and $\sum_t\|\mathbf{v}_t\|^2 + \|\mathbf{b}_t\|^2$ Lemma 13 shows that $\|\mathbf{x}_t - \mathbf{x}^*\| \leq 2\|\mathbf{x}_1 - \mathbf{x}^*\|$ with high probability. Under this event, we bound $\sum_t\langle\mathbf{v}_t, \mathbf{x}_t - \mathbf{x}^*\rangle$ using the standard Freeman's inequality and $\|\nabla F(\mathbf{x}_t)\|^2$ by using smoothness and the fact that $\nabla F(\mathbf{x}^\star) = 0$. The bias of the estimator $\|\mathbf{b}_t\|$ is bound using arguments similar to [8] (see Lemma 4). The main improvement of our method is given by our method of bounding $\frac{1}{T}\sum_t\|\mathbf{v}_t\|^2$. We show by an application of Theorem 5 that $\frac{1}{T}\sum_t\|\mathbf{v}_t\|^2 \lesssim \text{Tr}(\Sigma) + \sqrt{\text{Tr}(\Sigma)\|\Sigma\|_2}\log(\frac{\log T}{\delta})$ with probability at-least $1 - \delta$ whenever the clipping factor $\Gamma$ is appropriately chosen. Choosing the step size $\eta$ appropriately gives us the result in Theorem 3.

## 8 Conclusion and Limitations

Our work obtained nearly subgaussian rates for heavy-tailed SCO using clipped SGD by developing a fine-grained iterative refinement strategy for martingale concentration. As corollaries, we obtained state-of-the-art streaming estimators for various heavy tailed statistical problems. We note Clipped-SGD is widely used to optimize neural networks with highly nonconvex landscapes, which is currently outside the scope of our work. Nevertheless, we believe our techniques could be useful for providing sharp high-probability guarantees for non-convex losses. Our bounds are currently of the form $\sqrt{\frac{d + \sqrt{d}\ln(\ln(T)/\delta)}{T}}$, which is suboptimal compared to the tight subgaussian rate of $\sqrt{\frac{d + \ln(1/\delta)}{T}}$. Further research is required to understand if it is possible to obtain truly subgaussian rates with clipped mean type estimators. Another notable suboptimality of our result is the $\ln(\ln(T)/\delta)$ dependence on the confidence level (as opposed to the typical $\ln(1/\delta)$ scaling). However, this is not a major drawback as our results continue to significantly outperform prior works unless $T \gg e^{\exp(\sqrt{d}-1)\ln(1/\delta)}$ (which is an impractical regime). This drawback arises due to the $\ln(\ln(T))$ iterations of our iterative refinement technique and we believe it can be removed via more sophisticated martingale concentration arguments. Our work lays the foundation for several interesting avenues for future work including the analysis of heavy tailed statistical estimation under bounded $p^{\text{th}}$ moment assumptions (for $p < 2$) and the development of parameter free statistical estimators that do not require knowledge of problem-dependent parameter such as $\|\Sigma\|, \delta$ etc. (or their respective upper bounds). Deriving anytime valid guarantees for clipped SGD using our techniques is also an interesting future direction.

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

# Contents

## A  Preliminaries

In this section, we collect some preliminary concentration results which will be used in the future sections. For the following lemma, we refer to Exercise 2.8.5 in [54].

**Lemma 1.** *Suppose $X$ is a real valued random variable such that $|X| \leq \Gamma$ almost surely, $\mathbb{E}X = 0$ and $\mathbb{E}X^2 = \nu$. Then, for any $\lambda \in \mathbb{R}$ such that $|\lambda| \leq \frac{1}{2\Gamma}$, the following holds:*

$$\mathbb{E}\exp(\lambda X) \leq \exp(\lambda^2 \nu)$$

Consider a $\mathbb{R}^d$ valued martingale $(M_t)_{t=0}^T$ with respect to the filtration $(\mathcal{F}_t)_{t=0}^T$ such that $M_0 = 0$ almost surely. We consider the martingale difference sequence $\mathbf{v}_t := M_t - M_{t-1}$ for $t \geq 1$. Clearly, we must have:

$$M_t = \sum_{s=1}^{t} \mathbf{v}_s$$

**Definition 1.** *We say that the martingale $M_t$ satisfies $(g, T, \delta)$ uniform concentration if:*

$$\mathbb{P}(\sup_{0 \leq t \leq T} \|M_t\| > g\sqrt{T}) \leq \delta$$

Assume that for fixed $\Gamma > 0$ and $\Sigma \in \mathbb{R}^{d \times d}$ that $\|\mathbf{v}_s\| \leq \Gamma$ almost surely and $\mathbb{E}[\mathbf{v}_t \mathbf{v}_t^\mathsf{T}|\mathcal{F}_{t-1}] =: \Sigma_t$. Suppose $\mathsf{Tr}(\Sigma_t) \leq q_t$ and $\|\Sigma_t\|_2 \leq p_t$ almost surely for some non-random constants $p_t, q_t$. We state a high dimensional version of Freedman's inequality [20, 51] below which follows from From Corollary 1.3 of [51], we have

**Theorem 6.** *Suppose $M_t$ satisfies the assumptions above. Let $\bar{q} := \frac{1}{T}\sum_{s=1}^T q_t$ the following is true:*

$$\mathbb{P}(\sup_{0 \leq t \leq T} \|M_t\| > \alpha) \leq (d+1)\exp(-\frac{\alpha^2/2}{\bar{q}T + \frac{\Gamma\alpha}{3}})$$

*That is, for any $\delta > 0$, the martingale $(M_t)_{t \leq T}$ obeys $(g_0(\delta), T, \delta)$ uniform concentration, where $g_0(\delta) = \frac{2\Gamma}{3\sqrt{T}}\log(\frac{d+1}{\delta}) + \sqrt{2\bar{q}\log(\frac{d+1}{\delta})}$*

The following inequality is a corollary of Theorem 6.

**Lemma 2.** *Let $g_t \in \mathbb{R}^d$ be $\mathcal{F}_{t-1}$ measurable. Then for some constant $c_1 > 0$, we have:*

$$\mathbb{P}(\cup_{t=1}^T \{|\sum_{s=1}^t \langle g_s, \mathbf{v}_s\rangle| \geq \alpha\} \cap_{s \leq t} \{\|g_s\| \leq A_s\}) \leq 2\exp(-\frac{\alpha^2}{\Gamma A\alpha + c_1 \sum_{t=1}^T p_t A_t^2}) \tag{4}$$

*Where $A = \sup_{1 \leq t \leq T} A_t$*

In addition, we also use the following scalar version of Freedman's inequality

**Lemma 3** (Freedman's Inequality). *Let $h_1, h_2, \ldots, h_T$ be a $\mathcal{F}_t$ adapted martingale difference sequence such that $\mathbb{E}[h_t|\mathcal{F}_{t-1}] = 0$, $\mathbb{E}[h_t^2|\mathcal{F}_{t-1}] = \sigma_t^2$ and $\|h_t\| \leq \tau$. Then, for any $\delta \in (0,1)$, the following holds with probability at least $1 - \delta$:*

$$\sum_{s=1}^t h_s \leq 2\sqrt{\ln(1/\delta)\sum_{s=1}^t \sigma_s^2} + 2\tau\ln(1/\delta)$$

The following lemma, which bounds the moments of a clipped random vector, is crucial to our analysis of the bias and variance of the clipped stochastic gradient.

**Lemma 4** (Moments of a Clipped Random Vector). *Let $\mathbf{z} \in \mathbb{R}^d$ be a random vector sampled from the distribution $P$ with mean $\mathbf{m}$ and covariance matrix $\mathbf{S}$. For any $\Gamma > 0$, let $\tilde{\mathbf{z}} = \mathsf{clip}_\Gamma(\mathbf{z})$, and let $\tilde{\mathbf{m}}$ and $\tilde{\mathbf{S}}$ denote the mean and covariance of $\tilde{\mathbf{z}}$ respectively, i.e., $\tilde{\mathbf{m}} = \mathbb{E}[\tilde{\mathbf{z}}]$ and $\tilde{\mathbf{S}} = \mathbb{E}\left[(\tilde{\mathbf{z}} - \tilde{\mathbf{m}})(\tilde{\mathbf{z}} - \tilde{\mathbf{m}})^T\right]$. Then, the following hold:*

$$\|\tilde{\mathbf{m}} - \mathbf{m}\| \leq \frac{\sqrt{\|\mathbf{S}\|_2}}{\Gamma}\left(\|\mathbf{m}\| + \sqrt{\mathsf{Tr}(\mathbf{S})}\right) + \frac{\|\mathbf{m}\|}{\Gamma^2}\left(\|\mathbf{m}\|^2 + \mathsf{Tr}(\mathbf{S})\right)$$

$$\|\tilde{\mathbf{S}}\|_2 \leq \|\mathbf{S}\|_2 + \frac{\|\mathbf{m}\|^2}{\Gamma^2}\left(\|\mathbf{m}\|^2 + \mathsf{Tr}(\mathbf{S})\right)$$

$$\mathsf{Tr}(\tilde{\mathbf{S}}) \leq \mathsf{Tr}(\mathbf{S})$$

*Proof.* The proof of this lemma uses arguments similar to that of Catoni and Giulini [8]. We first note that for any $\mathbf{x} \in \mathbb{R}^d$

$$\mathsf{clip}_\Gamma(\mathbf{x}) = \mathbf{x} \cdot \frac{\min\{1, \Gamma^{-1}\|\mathbf{x}\|\}}{\Gamma^{-1}\|\mathbf{x}\|}$$

Following the proof of Proposition 2.1 of Catoni and Giulini [8], we observe that for any $t > 0$:

$$0 \le 1 - \frac{\min\{1, t\}}{t} \le \inf_{p \ge 1} \frac{p^p t^p}{(p+1)^{p+1}}$$

Define $\theta(\mathbf{x}) = \frac{\min\{1, \Gamma^{-1}\|\mathbf{x}\|\}}{\Gamma^{-1}\|\mathbf{x}\|}$ $\forall \mathbf{x} \in \mathbb{R}^d$. Note that $\mathsf{clip}_\Gamma(\mathbf{x}) = \theta(\mathbf{x}) \cdot \mathbf{x}$. From the above inequality, we note that:

$$0 \le 1 - \theta(\mathbf{x}) \le \inf_{p \ge 1} \frac{p^p}{(p+1)^{p+1}} \cdot \frac{\|\mathbf{x}\|^p}{\Gamma^p} \tag{5}$$

Consider any unit vector $\mathbf{e} \in \mathbb{R}^d$. Then,

$$
\begin{aligned}
\langle \mathbf{e}, \mathbf{m} - \tilde{\mathbf{m}} \rangle &= \mathbb{E}\left[ \langle \mathbf{e}, \mathbf{z} - \tilde{\mathbf{z}} \rangle \right] \\
&= \mathbb{E}\left[ \langle \mathbf{e}, \mathbf{z} - \theta(\mathbf{z})\mathbf{z} \rangle \right] \\
&= \mathbb{E}\left[ (1 - \theta(\mathbf{z})) \langle \mathbf{e}, \mathbf{z} - \mathbf{m} \rangle \right] + \langle \mathbf{e}, \mathbf{m} \rangle \mathbb{E}\left[ (1 - \theta(\mathbf{z})) \right] \\
&\le \mathbb{E}\left[ (1 - \theta(\mathbf{z})) | \langle \mathbf{e}, \mathbf{z} - \mathbf{m} \rangle | \right] + \|\mathbf{m}\| \mathbb{E}\left[ (1 - \theta(\mathbf{z})) \right] \\
&\le \mathbb{E}\left[ \inf_{p \ge 1} \frac{p^p}{(p+1)^{p+1}} \cdot \frac{\|\mathbf{z}\|^p | \langle \mathbf{e}, \mathbf{z} - \mathbf{m} \rangle |}{\Gamma^p} \right] + \|\mathbf{m}\| \mathbb{E}\left[ \inf_{p \ge 1} \frac{p^p}{(p+1)^{p+1}} \cdot \frac{\|\mathbf{z}\|^p}{\Gamma^p} \right]
\end{aligned}
$$

where the second step uses the definition of $\theta(\mathbf{z})$ and the last step uses equation (5). Now, substituting $p = 1$ and $p = 2$ in the first and second terms of the RHS respectively, we obtain the following:

$$
\begin{aligned}
\langle \mathbf{e}, \mathbf{m} - \tilde{\mathbf{m}} \rangle &\le \frac{1}{\Gamma} \mathbb{E}\left[ \|\mathbf{z}\| \langle \mathbf{e}, \mathbf{z} - \mathbf{m} \rangle \right] + \frac{\|\mathbf{m}\|}{\Gamma^2} \mathbb{E}[\|\mathbf{z}\|^2] \\
&\le \frac{1}{\Gamma} \sqrt{\mathbb{E}[\|\mathbf{x}\|^2]} \sqrt{\mathbb{E}[\langle \mathbf{e}, \mathbf{z} - \mathbf{m} \rangle^2]} + \frac{\|\mathbf{m}\|}{\Gamma^2} \mathbb{E}[\|\mathbf{z}\|^2] \\
&\le \frac{\sqrt{\|\mathbf{S}\|}}{\Gamma} \cdot \sqrt{\|\mathbf{m}\|^2 + \mathsf{Tr}(\mathbf{S})} + \frac{\|\mathbf{m}\|}{\Gamma^2} \left( \|\mathbf{m}\|^2 + \mathsf{Tr}(\mathbf{S}) \right) \\
&\le \frac{\sqrt{\|\mathbf{S}\|_2}}{\Gamma} \left( \|\mathbf{m}\| + \sqrt{\mathsf{Tr}(\mathbf{S})} \right) + \frac{\|\mathbf{m}\|}{\Gamma^2} \left( \|\mathbf{m}\|^2 + \mathsf{Tr}(\mathbf{S}) \right)
\end{aligned}
$$

where the second step uses the Cauchy Schwarz inequality and the last step uses the subadditivity of the square root. It follows that:

$$
\begin{aligned}
\|\tilde{\mathbf{m}} - \mathbf{m}\| &= \sup_{\|\mathbf{e}\|=1} \langle \mathbf{e}, \mathbf{m} - \tilde{\mathbf{m}} \rangle \\
&\le \frac{\sqrt{\|\mathbf{S}\|_2}}{\Gamma} \left( \|\mathbf{m}\| + \sqrt{\mathsf{Tr}(\mathbf{S})} \right) + \frac{\|\mathbf{m}\|}{\Gamma^2} \left( \|\mathbf{m}\|^2 + \mathsf{Tr}(\mathbf{S}) \right)
\end{aligned}
$$

To bound $\|\tilde{\mathbf{S}}\|$, we first note that for any $\mathbf{x} \in \mathbb{R}^d$, $0 \le \theta(\mathbf{x}) \le 1$. As before, let $\mathbf{e} \in \mathbb{R}^d$ denote an arbitrary unit vector. We note that $\mathbb{E}[\langle \mathbf{e}, \tilde{\mathbf{z}} - \mathbf{m} \rangle^2] = \mathbb{E}[\langle \mathbf{e}, \tilde{\mathbf{z}} - \tilde{\mathbf{m}} \rangle^2] + \mathbb{E}[\langle \mathbf{e}, \mathbf{m} - \tilde{\mathbf{m}} \rangle^2] \ge \mathbb{E}[\langle \mathbf{e}, \tilde{\mathbf{z}} - \tilde{\mathbf{m}} \rangle^2]$. Hence, it follows that,

$$
\begin{aligned}
\mathbb{E}\left[ \langle \mathbf{e}, \tilde{\mathbf{z}} - \tilde{\mathbf{m}} \rangle^2 \right] &\le \mathbb{E}\left[ \langle \mathbf{e}, \tilde{\mathbf{z}} - \mathbf{m} \rangle^2 \right] \\
&\le \mathbb{E}\left[ (\theta(\mathbf{z}) \langle \mathbf{e}, \mathbf{z} \rangle - \langle \mathbf{e}, \mathbf{m} \rangle)^2 \right] \\
&= \mathbb{E}\left[ (\theta(\mathbf{z}) \langle \mathbf{e}, \mathbf{z} - \mathbf{m} \rangle - (1 - \theta(\mathbf{z})) \langle \mathbf{e}, \mathbf{m} \rangle)^2 \right] \\
&\le \mathbb{E}\left[ \theta(\mathbf{z}) \langle \mathbf{e}, \mathbf{z} - \mathbf{m} \rangle^2 \right] + \langle \mathbf{e}, \mathbf{m} \rangle^2 \mathbb{E}\left[ (1 - \theta(\mathbf{z})) \right] \\
&\le \mathbb{E}\left[ \langle \mathbf{e}, \mathbf{z} - \mathbf{m} \rangle^2 \right] + \|\mathbf{m}\|^2 \mathbb{E}\left[ \inf_{p \ge 1} \frac{p^p}{(p+1)^{p+1}} \frac{\|\mathbf{z}\|^p}{\Gamma^p} \right] \\
&\le \|\mathbf{S}\|_2 + \frac{\|\mathbf{m}\|^2 \mathbb{E}[\|\mathbf{z}\|^2]}{\Gamma^2} \\
&\le \|\mathbf{S}\|_2 + \frac{\|\mathbf{m}\|^2}{\Gamma^2} \left( \|\mathbf{m}\|^2 + \mathsf{Tr}(\mathbf{S}) \right)
\end{aligned}
$$

where the fourth step uses Jensen's inequality by noting that $0 \leq \theta(\mathbf{z}) \leq 1$ and the fifth step uses equation (5).

Finally, To upper bound $\mathsf{Tr}(\tilde{\mathbf{S}})$, we note that $\mathsf{clip}_\Gamma$ is a contractive mapping as it is the projection operator onto a convex set (namely the ball of radius $\Gamma$ in $\mathbb{R}^d$ centered at the origin). To this end,

$$\mathsf{Tr}(\tilde{\mathbf{S}}) = \mathbb{E}\left[\|\tilde{\mathbf{z}} - \tilde{\mathbf{m}}\|^2\right] = \frac{1}{2}\mathbb{E}_{\mathbf{z}_1, \mathbf{z}_2 \overset{i.i.d.}{\sim} P}\left[\|\mathsf{clip}_\Gamma(\mathbf{z}_1) - \mathsf{clip}_\Gamma(\mathbf{z}_2)\|^2\right]$$
$$\leq \frac{1}{2}\mathbb{E}_{\mathbf{z}_1, \mathbf{z}_2 \overset{i.i.d.}{\sim} P}\left[\|\mathbf{z}_1 - \mathbf{z}_2\|^2\right] = \mathsf{Tr}(\mathbf{S})$$

$\square$

The following result, which is a corollary of Theorem 5, is vital for controlling the error introduced due to the variance of the stochastic gradients, and is one of the major components of our analysis. The proof of this result is presented in Appendix F.3

**Corollary 5** (PAC Bayesian Inequality for Quadratic Variation). *Let $\mathbf{v}_1, \ldots, \mathbf{v}_T$ be an $\mathbb{R}^d$ valued martingale difference sequence adapted to the filtration $\mathcal{F}_1, \ldots, \mathcal{F}_T$ satisfying $\mathbb{E}[\mathbf{v}_s | \mathcal{F}_s] = 0, \mathbb{E}[\mathbf{v}_s \mathbf{v}_s^T | \mathcal{F}_s] = \Sigma_s$ and $\|\mathbf{v}_s\| \leq \tau$ almost surely. Let $\mathsf{UP}(t) := \min(T, 2^{\lceil \log_2 t \rceil})$. Suppose $\|\Sigma_s\|_2 \leq p_s$ and $\mathsf{Tr}(\Sigma_s) \leq q_s$ for some fixed sequences $p_1, \ldots, p_T$ and $q_1, \ldots, q_T$. Then, there exists a universal constant $C_{\mathsf{lower}}$ such that whenever $T > C_{\mathsf{lower}} \log((1 + \frac{\sqrt{\bar{q}T}}{\Gamma}) \log(d+1))$ such that the following inequality holds with probability at least $1 - \delta$, for any $\delta \in (0, \frac{1}{2})$:*

$$\sum_{s=1}^{t} \|\mathbf{v}_s\|^2 \leq C_M \sum_{s=1}^{\mathsf{UP}(t)} q_s + C_M \tau^2 \ln(\tfrac{\ln(T)}{\delta})^2 + \frac{C_M t}{\tau^2} \sum_{s=1}^{\mathsf{UP}(t)} p_s^2 \quad \forall t \in [T]$$

*where $C_M > 0$ is an absolute numerical constant.*

# B   Analysis for Smooth Strongly Convex Functions

Let $d_{\mathsf{eff}} = \frac{\mathsf{Tr}(\Sigma)}{\|\Sigma\|_2}$ and let $K = 4\max\{8, C_M, \ln(T)\}$. For $t \geq 1$, define the filtration $\mathcal{F}_t = \sigma(\mathbf{x}_1, \mathbf{g}_s | 1 \leq s \leq t)$ and $\mathcal{F}_0 = \sigma(\mathbf{x}_1)$. Furthermore, let $\nabla F(\mathbf{x}_t) = \mathsf{clip}_\Gamma(\mathbf{g}_t) + \mathbf{b}_t + \mathbf{v}_t$ where $\mathbf{b}_t = \nabla F(\mathbf{x}_t) - \mathbb{E}[\mathsf{clip}_\Gamma(\mathbf{g}_t) | \mathcal{F}_{t-1}]$ and $\mathbf{v}_t = \mathbb{E}[\mathsf{clip}_\Gamma(\mathbf{g}_t) | \mathcal{F}_{t-1}] - \mathsf{clip}_\Gamma(\mathbf{g}_t)$. We note that $\mathbb{E}[\mathbf{v}_t | \mathcal{F}_{t-1}] = 0$ and

$$\|\mathbf{v}_t\| \leq \|\mathsf{clip}_\Gamma(\mathbf{g}_t)\| - \|\mathbb{E}[\mathsf{clip}_\Gamma(\mathbf{g}_t) | \mathcal{F}_{t-1}]\|$$
$$\leq \|\mathsf{clip}_\Gamma(\mathbf{g}_t)\| - \mathbb{E}[\|\mathsf{clip}_\Gamma(\mathbf{g}_t)\| | \mathcal{F}_{t-1}] \leq 2\Gamma$$

where the first step follows from the triangle inequality, the second step uses Jensen's inequality and the last step uses the definition of $\mathsf{clip}_\Gamma$. Hence $\mathbf{v}_t$ is an $\mathcal{F}$ adapted almost surely bounded martingale difference sequence. Now, let $D_t = \|\mathbf{x}_t - \mathbf{x}^*\|$ where $\mathbf{x}^*$ is the unique minimizer of $F$ (guaranteed by strong convexity). Let $\eta_t = \frac{A}{t+\gamma}$ where $A \geq 1$ is a numerical constant and $\gamma \geq A\kappa + A - 1$ is a constant depending on $\kappa, d$ and $\ln(1/\delta)$ which we shall specify later. Note that our choice of $\gamma$ ensures that $\eta_t \leq \frac{1}{L+\mu}$ for $t \in [1:T]$ We prove the following recurrence for $D_t$ by using the smoothness and strong convexity properties of $F$ and by exploiting the choice of the step-size.

**Lemma 5** (Recurrence for $D_t$). *The following holds for every $t \in [1:T]$*

$$D_{t+1}^2 \leq \left(\frac{\gamma+1}{t+\gamma}\right)^{2A} D_1^2 + \frac{A2^{2A+1}}{\mu} \sum_{s=1}^{t} \frac{(s+\gamma-1)^{2A-1}}{(t+\gamma)^{2A}} \langle \mathbf{b}_s, \mathbf{x}_s - \mathbf{x}^* \rangle$$
$$+ \frac{A^2 4^{A+1}}{\mu^2} \sum_{s=1}^{t} \|\mathbf{b}_s\|^2 \frac{(s+\gamma)^{2A-2}}{(t+\gamma)^{2A}} + \frac{A2^{2A+1}}{\mu} \sum_{s=1}^{t} \frac{(s+\gamma)^{2A-1}}{(t+\gamma)^{2A}} \langle \mathbf{v}_s, \mathbf{x}_s - \mathbf{x}^* \rangle$$
$$+ \frac{A^2 4^{A+1}}{\mu^2} \sum_{s=1}^{t} \|\mathbf{v}_s\|^2 \frac{(s+\gamma)^{2A-2}}{(t+\gamma)^{2A}}$$

Now define $R_{T,\delta}$ as follows:

$$R_{T,\delta} = (\gamma + 1)^2 D_1^2 + \frac{(T+\gamma)\|\Sigma\|_2}{\mu^2}\left(d_{\text{eff}} + \sqrt{d_{\text{eff}}}\ln(K/\delta)\right)$$

It is easy to see that $\Gamma = \frac{\mu\sqrt{R_{T,\delta}}}{\ln(K/\delta)}$. In our proof of Theorem 1, we shall establish that the following holds with probability at least $1 - \delta$:

$$D_t^2 \leq \frac{CR_{T,\delta}}{(t+\gamma-1)^2} \; \forall\, t \in [1:T+1]$$

where $C > 0$ is an absolute numerical constant to be chosen later. To this end, we define the event $E_t$ and the random variables $\mathbf{d}_t, \tilde{\mathbf{b}}_t, \tilde{\mathbf{v}}_t$ as follows for $t \in [1:T+1]$:

$$E_t = \left\{D_t^2 \leq \frac{CR_{T,\delta}}{(t+\gamma-1)^2}\right\}$$
$$\mathbf{d}_t = (\mathbf{x}_t - \mathbf{x}^*)\mathbb{1}\{E_t\}$$
$$\tilde{\mathbf{b}}_t = \mathbf{b}_t\mathbb{1}\{E_t\}$$
$$\tilde{\mathbf{v}}_t = \mathbf{v}_t\mathbb{1}\{E_t\}$$

We note that since $\mathbf{x}_t$ is $\mathcal{F}_{t-1}$ measurable, so are $\mathbb{1}\{E_t\}, D_t, \mathbf{d}_t, \mathbf{b}_t$ and $\tilde{\mathbf{b}}_t$. Furthermore, $\mathbb{E}[\tilde{\mathbf{v}}_t|\mathcal{F}_{t-1}] = \mathbb{E}[\mathbf{v}_t|\mathcal{F}_{t-1}]\mathbb{1}\{E_t\} = 0$.

We use the following Lemma to control the bias vector $\tilde{\mathbf{b}}_t$

**Lemma 6** (Bias Control). *The following holds almost surely for every $t \in [1:T]$:*

$$\|\tilde{\mathbf{b}}_t\| \leq \mu\sqrt{R_{T,\delta}}\left(\frac{1}{T+\gamma} + \frac{\kappa\ln(1/\delta)\sqrt{C}}{(t+\gamma-1)\sqrt{d(T+\gamma)}} + \frac{\kappa^3 C^{3/2}\ln(1/\delta)^2}{(t+\gamma-1)^3} + \frac{\kappa\sqrt{C}\ln(1/\delta)^2}{(t+\gamma-1)(T+\gamma)}\right)$$

We use the following lemma to control the variance vector $\tilde{\mathbf{v}}_t$. The proof of this lemma, which uses Freedman's inequality and the PAC Bayesian martingale concentration inequality of Corollary 6.

**Lemma 7** (Variance Control). *The following holds with probability at least $1 - \delta$ uniformly for every $t \in [T]$ whenever $A \geq 3$ and $\gamma \geq 4\max\{\kappa^{4/3}C^{2/3}\ln(\ln(T)/\delta), \kappa\sqrt{C}\ln(\ln(T)/\delta)^{3/2}\}$:*

$$\sum_{s=1}^{t} \frac{(s+\gamma)^{2A-1}}{(t+\gamma)^{2A-2}}\langle\tilde{\mathbf{v}}_s, \mathbf{d}_s\rangle \leq 27\mu R_{T,\delta}\sqrt{C}$$

$$\sum_{s=1}^{t} \left(\frac{s+\gamma}{t+\gamma}\right)^{2A-2}\|\tilde{\mathbf{v}}_s\|^2 \leq C_M\mu^2 R_{T,\delta}\left(6 + 3\cdot 2^{4A-13} + 3\cdot 2^{4A-17}\right)$$

*where $C_M$ is the absolute numerical constant defined in Corollary 5.*

Equipped with this bound on the bias and the variance, we now present the complete proof as follows:

## B.1   Proof of Theorem 1

*Proof.* Let $A \geq 3$, $\gamma \geq 4\max\{\kappa^{4/3}C^{2/3}\ln(\ln(T)/\delta), \kappa\sqrt{C}\ln(\ln(T)/\delta)^{3/2}\}$. Now, let $E$ denote the following event

$$E = \{\sum_{s=1}^{t}\frac{(s+\gamma)^{2A-1}}{(t+\gamma)^{2A-2}}\langle\tilde{\mathbf{v}}_s, \mathbf{d}_s\rangle \leq 27\mu R_{T,\delta}\sqrt{C} \; \forall\, t \in [T]$$

$$\sum_{s=1}^{t}\left(\frac{s+\gamma}{t+\gamma}\right)^{2A-2}\|\tilde{\mathbf{v}}_s\|^2 \leq C_M\mu^2 R_{T,\delta}\left(6 + 3\cdot 2^{4A-13} + 3\cdot 2^{4A-17}\right) \; \forall t \in [T]\}$$

Note that by Lemma 7, $\mathbb{P}(E) \geq 1 - \delta$. We now claim that $\mathbb{P}\left(\cap_{t=1}^{T+1} E_t|E\right) = 1$, i.e., conditioned on the event $E$, the following holds almost surely for every $t \in [1:T+1]$

$$D_t^2 \leq \frac{CR_{T,\delta}}{(t+\gamma-1)^2} \; \forall\, t \in [1:T+1]$$

We prove the above claim by induction. Note that the claim is trivially true for $t = 1$ as $R_{T,\delta} \geq (\gamma + 1)^2 D_1^2$. Now, consider any $t \in [1 : T]$ and suppose the claim holds for some $1 \leq s \leq t$.

Recall that by Lemma 5

$$(t+\gamma)^2 D_{t+1}^2 \leq \frac{(\gamma+1)^{2A}}{(t+\gamma)^{2A-2}} D_1^2 + \frac{A2^{2A+1}}{\mu} \sum_{s=1}^{t} \frac{(s+\gamma-1)^{2A-1}}{(t+\gamma)^{2A-2}} \langle \mathbf{b}_s, \mathbf{x}_s - \mathbf{x}^* \rangle$$

$$+ \frac{A^2 4^{A+1}}{\mu^2} \sum_{s=1}^{t} \|\mathbf{b}_s\|^2 \frac{(s+\gamma)^{2A-2}}{(t+\gamma)^{2A-2}} + \frac{A2^{2A+1}}{\mu} \sum_{s=1}^{t} \frac{(s+\gamma)^{2A-1}}{(t+\gamma)^{2A-2}} \langle \mathbf{v}_s, \mathbf{x}_s - \mathbf{x}^* \rangle$$

$$+ \frac{A^2 4^{A+1}}{\mu^2} \sum_{s=1}^{t} \|\mathbf{v}_s\|^2 \frac{(s+\gamma)^{2A-2}}{(t+\gamma)^{2A-2}}$$

Under the induction hypothesis, $\mathbb{1}\{E_s\} = 1 \; \forall s \in [t]$. Hence, Under the induction hypothesis, $\mathbb{1}\left\{ D_s^2 \leq \frac{CR_{T,\delta}}{(s+\gamma-1)(s+\gamma-2)} \right\} = 1$ and thus, $\mathbf{d}_s = \mathbf{x}_s - \mathbf{x}^*, \mathbf{b}_s = \tilde{\mathbf{b}}_s, \mathbf{v}_s = \tilde{\mathbf{v}}_s \; \forall \; 1 \leq s \leq t$. Substituting this transformation into the above inequality, we obtain the following:

$$(t+\gamma)^2 D_{t+1}^2 \leq \underbrace{\frac{(\gamma+1)^{2A}}{(t+\gamma)^{2A-2}} D_1^2}_{\textcircled{1}} + \underbrace{\frac{A2^{2A+1}}{\mu} \sum_{s=1}^{t} \frac{(s+\gamma)^{2A-1}}{(t+\gamma)^{2A-2}} \langle \tilde{\mathbf{v}}_s, \mathbf{d}_s \rangle}_{\textcircled{2}}$$

$$+ \underbrace{\frac{A^2 4^{A+1}}{\mu^2} \sum_{s=1}^{t} \|\tilde{\mathbf{v}}_s\|^2 \frac{(s+\gamma)^{2A-2}}{(t+\gamma)^{2A-2}}}_{\textcircled{3}} + \underbrace{\sum_{s=1}^{t} \frac{(s+\gamma)^{2A-1}}{(t+\gamma)^{2A-2}} \langle \tilde{\mathbf{b}}_s, \mathbf{d}_s \rangle}_{\textcircled{4}}$$

$$+ \underbrace{\frac{A^2 4^{A+1}}{\mu^2} \sum_{s=1}^{t} \|\tilde{\mathbf{b}}_s\|^2 \frac{(s+\gamma)^{2A-2}}{(t+\gamma)^{2A-2}}}_{\textcircled{5}} \tag{6}$$

We now bound each of the terms in the RHS as follows.

**Bounding $\textcircled{1}$** Since $A \geq 1$ and $t \geq 1$,

$$\textcircled{1} = \frac{(\gamma+1)^{2A}}{(t+\gamma)^{2A-2}} D_1^2 \leq (\gamma+1)^2 D_1^2 \leq R_{T,\delta}$$

**Bounding $\textcircled{2}$** Since $\gamma$ and $A$ satisfy the conditions of Lemma 7 and we have conditioned on the event $E$, it follows that:

$$\frac{A2^{2A+1}}{\mu} \sum_{s=1}^{t} \frac{(s+\gamma)^{2A-1}}{(t+\gamma)^{2A-2}} \langle \tilde{\mathbf{v}}_s, \mathbf{d}_s \rangle \leq 27 A 2^{2A+1} R_{T,\delta} \sqrt{C}$$

**Bounding $\textcircled{3}$** Since $\gamma$ and $A$ satisfy the conditions of Lemma 7 and we have conditioned on the event $E$, it follows that:

$$\frac{A^2 4^{A+1}}{\mu^2} \sum_{s=1}^{t} \left( \frac{s+\gamma}{t+\gamma} \right)^{2A-2} \|\tilde{\mathbf{v}}_s\|^2 \leq C_M 2^{2A+2} \left( 6 + 3 \cdot 2^{4A-13} + 3 \cdot 2^{4A-17} \right) R_{T,\delta}$$

Before controlling terms $\textcircled{4}$ and $\textcircled{5}$, we note that the following holds for every $s \in [t]$ by Lemma 6

$$\|b_s\| \leq \mu \sqrt{R_{T,\delta}} (B_1 + B_2 + B_3 + B_4)$$

where $B_1, \ldots, B_4$ are defined as:

$$B_1 = \frac{1}{T + \gamma}$$

$$B_2 = \frac{\kappa \ln(K/\delta)\sqrt{C}}{(s + \gamma - 1)\sqrt{d(T + \gamma)}}$$

$$B_3 = \frac{\kappa^3 C^{3/2} \ln(K/\delta)^2}{(s + \gamma - 1)^3}$$

$$B_4 = \frac{\kappa \ln(K/\delta)^2 \sqrt{C}}{(s + \gamma - 1)(T + \gamma)}$$

**Bounding ④** Since $\mathbb{1}\{E_s\} = 1$

$$\|\mathbf{d}_s\| \leq \frac{\sqrt{CR_{T,\delta}}}{s + \gamma - 1} \leq \frac{2\sqrt{CR_{T,\delta}}}{s + \gamma}$$

Hence,

$$\frac{A2^{2A+1}}{\mu} \sum_{s=1}^{t} \left\langle \tilde{\mathbf{b}}_s, \mathbf{d}_s \right\rangle \frac{(s + \gamma)^{2A-1}}{(t + \gamma)^{2A-2}} \leq A2^{2A+2} R_{T,\delta}\sqrt{C} \sum_{s=1}^{t} \left(\frac{s + \gamma}{t + \gamma}\right)^{2A-2} (B_1 + B_2 + B_3 + B_4)$$

We now control the first term

$$\sum_{s=1}^{t} \left(\frac{s + \gamma}{t + \gamma}\right)^{2A-2} B_1 = \frac{1}{T + \gamma} \sum_{s=1}^{t} \left(\frac{s + \gamma}{t + \gamma}\right)^{2A-2}$$

$$\leq \frac{t}{T + \gamma} \leq 1$$

where the first inequality follows from the fact that $A \geq 1$ and $s \leq t$. We now bound the second term

$$\sum_{s=1}^{t} \left(\frac{s + \gamma}{t + \gamma}\right)^{2A-2} B_2 \leq \frac{\kappa\sqrt{C}\ln(K/\delta)}{\sqrt{d(T + \gamma)}} \left[\sum_{s=1}^{t} \left(\frac{s + \gamma}{t + \gamma}\right)^{2A-2} \frac{1}{s + \gamma - 1}\right]$$

Setting $A \geq 3/2$ and using the fact that $s + \gamma \geq 2$, it follows that

$$\sum_{s=1}^{t} \left(\frac{s + \gamma}{t + \gamma}\right)^{2A-2} B_2 \leq \frac{2\kappa\sqrt{C}\ln(K/\delta)}{\sqrt{d(T + \gamma)}} \sum_{s=1}^{t} \frac{(s + \gamma)^{2A-3}}{(t + \gamma)^{2A-2}}$$

$$\leq \frac{2\kappa\sqrt{C}\ln(K/\delta)}{\sqrt{d(T + \gamma)}} \leq 2$$

where the last inequality follows by setting $\gamma \geq \frac{C\kappa^2}{d} \cdot \ln(K/\delta)^2$

To control the third term, we set $A \geq 5/2$ and proceed as follows:

$$\sum_{s=1}^{t} \left(\frac{s + \gamma}{t + \gamma}\right)^{2A-2} B_3 \leq \kappa^3 C^{3/2} \ln(K/\delta)^2 \sum_{s=1}^{t} \frac{(s + \gamma)^{2A-5}}{(t + \gamma)^{2A-2}}$$

$$\leq \frac{\kappa^3 C^{3/2} \ln(K/\delta)^2}{(t + \gamma)^2}$$

$$\leq \frac{\kappa^3 C^{3/2} \ln(K/\delta)^2}{(\gamma + 1)^2} \leq 1$$

where the last inequality follows by setting $\gamma \geq \kappa^{3/2} C^{3/4} \ln(K/\delta)$.

To bound the last term,

$$\sum_{s=1}^{t} \left(\frac{s + \gamma}{t + \gamma}\right)^{2A-2} B_4 \leq \frac{\kappa C^{1/2} \ln(K/\delta)^2}{T + \gamma} \sum_{s=1}^{t} \frac{(s + \gamma)^{2A-3}}{(t + \gamma)^{2A-2}}$$

$$\leq \frac{\kappa C^{1/2} \ln(1/\delta)^2}{\gamma + 1} \leq 1$$

where the second inequality uses the fact that $A \geq 3/2$ and the last inequality follows by setting $\gamma \geq \kappa C^{1/2} \ln(K/\delta)^2$. Putting it all together, it follows that

$$④ \leq 5A4^{A+1} R_{T,\delta} \sqrt{C}$$

by setting $\gamma$ as follows

$$\gamma \geq \max \left\{ \frac{\kappa^2 C}{d} \cdot \ln(K/\delta)^2, \kappa^{3/2} C^{3/4} \ln(1/\delta), \kappa C^{1/2} \ln(K/\delta)^2 \right\}$$

**Bounding ⑤**   By Lemma 6 and Jensen's inequality

$$\|\tilde{\mathbf{b}}_s\|^2 \leq 4\mu^2 R_{T,\delta} \left( B_1^2 + B_2^2 + B_3^2 + B_4^2 \right)$$

It follows that

$$\frac{A^2 2^{2A+2}}{\mu^2} \sum_{s=1}^{t} \|\tilde{\mathbf{b}}_s\|^2 \left( \frac{s+\gamma}{t+\gamma} \right)^{2A-2} \leq A^2 4^{A+2} R_{T,\delta} \sum_{s=1}^{t} \left( \frac{s+\gamma}{t+\gamma} \right)^{2A-2} \left( B_1^2 + B_2^2 + B_3^2 + B_4^2 \right)$$

The first term is controlled as follows using the fact that $A \geq 1$

$$\sum_{s=1}^{t} \left( \frac{s+\gamma}{t+\gamma} \right)^{2A-2} B_1^2 = \sum_{s=1}^{t} \frac{1}{(T+\gamma)^2} \leq 1$$

The second term is controlled as

$$\sum_{s=1}^{t} \left( \frac{s+\gamma}{t+\gamma} \right)^{2A-2} B_2^2 = \frac{4\kappa^2 C \ln(K/\delta)^2}{d(T+\gamma)} \sum_{s=1}^{t} \frac{(s+\gamma)^{2A-4}}{(t+\gamma)^{2A-2}}$$

$$\leq \frac{4\kappa^2 C \ln(K/\delta)^2}{d(t+\gamma)(T+\gamma)} \leq 1$$

where the last inequality follows because $\gamma \geq \kappa \sqrt{\frac{C}{d}} \ln(K/\delta)$
For controlling the third term, we set $A \geq 4$ to obtain

$$\sum_{s=1}^{t} \left( \frac{s+\gamma}{t+\gamma} \right)^{2A-2} B_3^2 = \kappa^6 C^3 \ln(K/\delta)^4 \sum_{s=1}^{t} \frac{(s+\gamma)^{2A-8}}{(t+\gamma)^{2A-2}}$$

$$\leq \frac{\kappa^6 C^3 \ln(K/\delta)^4}{(\gamma+1)^5} \leq 1$$

where the last inequality uses the fact that $\gamma \geq \kappa^{6/5} C^{3/5} \ln(K/\delta)^{4/5}$ To control the fourth term, we use the fact that $A \geq 2$ to obtain

$$\sum_{s=1}^{t} \left( \frac{s+\gamma}{t+\gamma} \right)^{2A-2} B_4^2 = \frac{\kappa^2 C \ln(K/\delta)^4}{(T+\gamma)^2} \sum_{s=1}^{t} \frac{(s+\gamma)^{2A-4}}{(t+\gamma)^{2A-2}}$$

$$\leq \frac{\kappa^2 C \ln(K/\delta)^4}{(\gamma+1)^3} \leq 1$$

where the last inequality uses the fact that $\gamma \geq \kappa^{2/3} C^{1/3} \ln(K/\delta)^{4/3}$ From the obtained bounds, we conclude that $⑤ \leq A^2 4^{A+3} R_{T,\delta}$.

Hence, setting $A = 4$ and $\gamma = 4C \max\{ \frac{\|\Sigma\|_2 \kappa^2 \ln(\ln(T)/\delta)^2}{\mathrm{Tr}(\Sigma)}, \kappa^{3/2} \ln(\ln(T)/\delta), \kappa \ln(\ln(T)/\delta)^2 \}$, we obtain the following

$$(t+\gamma)^2 D_{t+1}^2 \leq ① + ② + ③ + ④ + ⑤$$

$$\leq R_{T,\delta} \left[ 1 + C_M 2^{2A+2} \left( 6 + 3 \cdot 2^{4A-13} + 3 \cdot 2^{4A-17} \right) + A^2 4^{A+3} + \sqrt{C} \left( 27 A 2^{2A+1} + 5 A 4^{A+1} \right) \right]$$

$$\leq R_{T,\delta} \left( 262145 + 524288 C_M + 75776 \sqrt{C} \right)$$

$$\leq C R_{T,\delta}$$

where the last inequality is obtained by setting $C = \left(\sqrt{262145 + 524288C_M} + 75776\right)^2$. It follows that

$$D_{t+1}^2 \le \frac{CR_{T,\delta}}{(t+\gamma)^2}$$

Thus, we have proved by induction that conditioned on $E$, $D_t^2 \le \frac{CR_{T,\delta}}{(t+\gamma)^2}$ for every $t \in [T+1]$. In particular, the following holds with probability at least $1 - \delta$:

$$D_{T+1}^2 \le C\left(\frac{\gamma+1}{T+\gamma}\right)^2 D_1^2 + \frac{C\|\Sigma\|_2 \left(d_{\text{eff}} + \sqrt{d_{\text{eff}}} \ln(K/\delta)\right)}{\mu^2(T+\gamma)}$$

$$\lesssim \left(\frac{\gamma+1}{T+\gamma}\right)^2 D_1^2 + \frac{\text{Tr}(\Sigma) + \sqrt{\|\Sigma\|_2 \text{Tr}(\Sigma)} \ln(\ln(T)/\delta)}{\mu^2(T+\gamma)}$$

$\square$

## B.2 Proof of Lemma 5

Let $\epsilon_t = \mathbf{b}_t + \mathbf{v}_t$

$$D_{t+1}^2 = \|\Pi_\mathcal{X}(\mathbf{x}_t - \eta_t \nabla F(\mathbf{x}_t) + \eta_t \epsilon_t) - \mathbf{x}^*\|^2$$
$$\le \|\mathbf{x}_t - \eta_t \nabla F(\mathbf{x}_t) + \eta_t \epsilon_t\|^2$$
$$\le D_t^2 - 2\eta_t \langle \nabla F(\mathbf{x}_t), \mathbf{x}_t - \mathbf{x}^* \rangle + 2\eta_t \langle \epsilon_t, \mathbf{x}_t - \mathbf{x}^* \rangle + 2\eta_t^2 \|\nabla F(\mathbf{x}_t)\|^2 + 2\eta_t^2 \|\epsilon_t\|^2$$

By the coercivity lemma in Bubeck [6] ,

$$\|\nabla F(\mathbf{x}_t)\|^2 \le (L+\mu) \langle \nabla F(\mathbf{x}_t), \mathbf{x}_t - \mathbf{x}^* \rangle - L\mu D_t^2$$

It follows that,

$$D_{t+1}^2 \le (1 - 2\eta_t^2 L\mu)D_t^2 - 2\eta_t[1 - \eta_t(L+\mu)] \langle \nabla F(\mathbf{x}_t), \mathbf{x}_t - \mathbf{x}^* \rangle + 2\eta_t \langle \epsilon_t, \mathbf{x}_t - \mathbf{x}^* \rangle + 2\eta_t^2 \|\epsilon_t\|^2$$
$$\le (1 - 2\eta_t^2 L\mu)D_t^2 - 2\eta_t[1 - \eta_t(L+\mu)]\mu D_t^2 + 2\eta_t \langle \epsilon_t, \mathbf{x}_t - \mathbf{x}^* \rangle + 2\eta_t^2 \|\epsilon_t\|^2$$
$$\le (1 - 2\eta_t\mu - 2\eta_t^2\mu^2)D_t^2 + 2\eta_t \langle \epsilon_t, \mathbf{x}_t - \mathbf{x}^* \rangle + 2\eta_t^2 \|\epsilon_t\|^2$$
$$\le (1 - 2\eta_t\mu)D_t^2 + 2\eta_t \langle \epsilon_t, \mathbf{x}_t - \mathbf{x}^* \rangle + 2\eta_t^2 \|\epsilon_t\|^2$$

where the second inequality follows from the strong monotonicity property of $\nabla F(\mathbf{x})$ and the fact that $\eta_t \le \frac{1}{L+\mu}$ since $\gamma \ge A\kappa + A - 1$. Now, substituting $\eta_t = \frac{A}{\mu(t+\gamma)}$,

$$D_{t+1}^2 \le \left(1 - \frac{2A}{t+\gamma}\right)D_t^2 + \frac{2A}{\mu(t+\gamma)} \langle \epsilon_t, \mathbf{x}_t - \mathbf{x}^* \rangle + \frac{2A^2\|\epsilon_t\|^2}{\mu^2(t+\gamma)^2} \quad (7)$$

Since $1 - t \le e^{-t} \; \forall \, t \in \mathbb{R}$, we note that $\forall s < t$:

$$\prod_{j=s+1}^{t} \left(1 - \frac{2A}{j+\gamma}\right) \le \exp\left(-\sum_{j=s+1}^{t} \frac{2A}{j+\gamma}\right)$$
$$\le \exp\left(-2A \int_{s+1}^{t+1} \frac{du}{u+\gamma}\right)$$
$$\le \exp\left(-2A \ln\left(\frac{t+1+\gamma}{s+1+\gamma}\right)\right)$$
$$= \left(\frac{s+1+\gamma}{t+1+\gamma}\right)^{2A}$$
$$\le 2^{2A} \left(\frac{s+\gamma}{t+\gamma}\right)^{2A}$$

Using the above bound to unroll the recurence (7), we obtain:

$$D_{t+1}^2 \leq \left[ \prod_{j=1}^t \left( 1 - \frac{2A}{j+\gamma} \right) \right] D_1^2 + \frac{2A}{\mu} \sum_{s=1}^t \frac{\langle \epsilon_s, \mathbf{x}_s - \mathbf{x}^* \rangle}{(s+\gamma)} \left[ \prod_{j=s+1}^t \left( 1 - \frac{2A}{j+\gamma} \right) \right]$$

$$+ \frac{2A^2}{\mu^2} \sum_{s=1}^t \frac{\|\epsilon_s\|^2}{(s+\gamma)^2} \left[ \prod_{j=s+1}^t \left( 1 - \frac{2A}{j+\gamma} \right) \right]$$

$$\leq \left( \frac{\gamma+1}{t+\gamma} \right)^{2A} D_1^2 + \frac{A2^{2A+1}}{\mu} \sum_{s=1}^t \frac{(s+\gamma)^{2A-1}}{(t+\gamma)^{2A}} \langle \epsilon_s, \mathbf{x}_s - \mathbf{x}^* \rangle + \frac{A^2 2^{2A+1}}{\mu^2} \sum_{s=1}^t \|\epsilon_s\|^2 \frac{(s+\gamma)^{2A-2}}{(t+\gamma)^{2A}}$$

Expanding $\epsilon_s = \mathbf{b}_s + \mathbf{v}_s$ and using Young's inequality, we conclude that the following holds for every $t \in [1:T]$

$$D_{t+1}^2 \leq \left( \frac{\gamma+1}{t+\gamma} \right)^{2A} D_1^2 + \frac{A2^{2A+1}}{\mu} \sum_{s=1}^t \frac{(s+\gamma-1)^{2A-1}}{(t+\gamma)^{2A}} \langle \mathbf{b}_s, \mathbf{x}_s - \mathbf{x}^* \rangle$$

$$+ \frac{A^2 4^{A+1}}{\mu^2} \sum_{s=1}^t \|\mathbf{b}_s\|^2 \frac{(s+\gamma)^{2A-2}}{(t+\gamma)^{2A}} + \frac{A2^{2A+1}}{\mu} \sum_{s=1}^t \frac{(s+\gamma)^{2A-1}}{(t+\gamma)^{2A}} \langle \mathbf{v}_s, \mathbf{x}_s - \mathbf{x}^* \rangle$$

$$+ \frac{A^2 4^{A+1}}{\mu^2} \sum_{s=1}^t \|\mathbf{v}_s\|^2 \frac{(s+\gamma)^{2A-2}}{(t+\gamma)^{2A}}$$

### B.3   Proof of Lemma 6

Note that by definition of $E_t$

$$\|\nabla F(\mathbf{x}_t)\| \mathbb{1}\{E_t\} \leq LD_t \mathbb{1}\{E_t\}$$

$$\leq L \frac{\sqrt{CR_{T,\delta}}}{(t+\gamma-1)}$$

Recall that $\Gamma = \frac{\mu\sqrt{R_{T,\delta}}}{\ln(K/\delta)}$ i.e. $\sqrt{R_{T,\delta}} = \frac{\gamma \ln(K/\delta)}{\mu}$. Substituting this into the above inequality gives us:

$$\|\nabla F(\mathbf{x}_t)\| \mathbb{1}\{E_t\} \leq \frac{\kappa \Gamma \ln(K/\delta)\sqrt{C}}{t+\gamma-1} \tag{8}$$

We recall that $\mathbf{b}_t = \nabla F(\mathbf{x}_t) - \mathbb{E}[\mathrm{clip}_\Gamma(\mathbf{g}_t)|\mathcal{F}_{t-1}] = \mathbb{E}[\mathbf{g}_t|\mathcal{F}_{t-1}] - \mathbb{E}[\mathrm{clip}_\Gamma(\mathbf{g}_t)|\mathcal{F}_{t-1}]$. Since $\mathrm{Cov}[\mathbf{g}_t|\mathcal{F}_{t-1}] \preceq \Sigma$ by Assumption Bdd. 2nd Moment, we obtain the following bound on $\|\mathbf{b}_t\|$ by an application of Lemma 4

$$\|\mathbf{b}_t\| \leq \frac{\|\Sigma\|_2 \sqrt{d_{\mathrm{eff}}}}{\Gamma} + \frac{\|\nabla F(\mathbf{x}_t)\|\sqrt{\|\Sigma\|_2}}{\Gamma} + \frac{\|\nabla F(\mathbf{x}_t)\|^3}{\Gamma^2} + \frac{\|\Sigma\|_2 d_{\mathrm{eff}} \|\nabla F(\mathbf{x}_t)\|}{\Gamma^2}$$

Since $\tilde{\mathbf{b}}_t = \mathbf{b}_t \mathbb{1}\{E_t\}$, it follows that

$$\|\tilde{\mathbf{b}}_t\| \leq \underbrace{\frac{\|\Sigma\|_2 \sqrt{d_{\mathrm{eff}}}}{\Gamma}}_{\text{Ⓐ}} + \underbrace{\frac{\|\nabla F(\mathbf{x}_t)\| \mathbb{1}\{E_t\}\sqrt{\|\Sigma\|_2}}{\Gamma}}_{\text{Ⓑ}} + \underbrace{\frac{\|\nabla F(\mathbf{x}_t)\|^3 \mathbb{1}\{E_t\}}{\Gamma^2}}_{\text{Ⓒ}} + \underbrace{\frac{\|\Sigma\|_2 d_{\mathrm{eff}} \|\nabla F(\mathbf{x}_t)\| \mathbb{1}\{E_t\}}{\Gamma^2}}_{\text{Ⓓ}}$$

**Bounding Ⓐ**   By definition of $\Gamma$,

$$\frac{\|\Sigma\|_2 \sqrt{d_{\mathrm{eff}}}}{\Gamma} = \frac{\|\Sigma\|_2 \sqrt{d_{\mathrm{eff}}} \ln(K/\delta)}{\mu\sqrt{R_{T,\delta}}}$$

$$\leq \frac{(T+\gamma)\|\Sigma\|_2 \sqrt{d_{\mathrm{eff}}} \ln(K/\delta)}{\mu T \sqrt{R_{T,\delta}}}$$

$$\leq \frac{\mu\sqrt{R_{T,\delta}}}{(T+\gamma)}$$

Hence Ⓐ $\leq \frac{\mu\sqrt{R_{T,\delta}}}{T+\gamma}$

**Bounding Ⓑ** Since $R_{T,\delta} \geq \frac{\|\Sigma\|_2 d_{\text{eff}}(T+\gamma)}{\mu^2} \geq \frac{\|\Sigma\|_2 T}{\mu^2}$, $\sqrt{\|\Sigma\|_2} \leq \frac{\mu\sqrt{R_{T,\delta}}}{\sqrt{d(T+\gamma)}}$. Substituting this into equation (8),

$$\frac{\|\nabla F(\mathbf{x}_t)\| \mathbb{1}\{E_t\} \sqrt{\|\Sigma\|_2}}{\Gamma} \leq \frac{\kappa\sqrt{C}\ln(K/\delta)}{t+\gamma-1} \cdot \frac{\mu\sqrt{R_{T,\delta}}}{\sqrt{d(T+\gamma)}}$$

Hence, Ⓑ $\leq \mu\sqrt{R_{T,\delta}} \cdot \frac{\kappa\ln(1/\delta)\sqrt{C}}{(s+\gamma)\sqrt{d(T+\gamma)}}$

**Bounding Ⓒ** From equation (8),

$$\frac{\|\nabla F(\mathbf{x}_t)\|^3}{\Gamma^2} \leq \frac{\kappa^3 C^{3/2}\Gamma\ln(1/\delta)^3}{(t+\gamma-1)^3}$$

$$\leq \mu\sqrt{R_{T,\delta}} \cdot \frac{\kappa^3 C^{3/2}\ln(1/\delta)^2}{(t+\gamma-1)^3}$$

Hence, Ⓒ $\leq \mu\sqrt{R_{T,\delta}} \cdot \frac{\kappa^3 C^{3/2}\ln(1/\delta)^2}{(t+\gamma-1)^3}$

**Bounding Ⓓ** Recall that,

$$\|\Sigma\|_2 d_{\text{eff}} \leq \frac{\mu^2 R_{T,\delta}}{T+\gamma}$$

$$\frac{\|\nabla F(\mathbf{x}_t)\| \mathbb{1}\{E_t\}}{\Gamma} \leq \frac{\kappa\ln(K/\delta)\sqrt{C}}{(t+\gamma-1)}$$

$$\Gamma = \frac{\mu\sqrt{R_{T,\delta}}}{\ln(K/\delta)}$$

It follows that

$$Ⓓ = \frac{\|\Sigma\|_2 d_{\text{eff}}\|\nabla F(\mathbf{x}_t)\| \mathbb{1}\{E_t\}}{\Gamma^2} \leq \mu\sqrt{R_{T,\delta}} \cdot \frac{\kappa\ln(K/\delta)^2\sqrt{C}}{(t+\gamma-1)(T+\gamma)}$$

Hence,

$$\|\tilde{\mathbf{b}}_t\| \leq \mu\sqrt{R_{T,\delta}}\left(\frac{1}{T+\gamma} + \frac{\kappa\ln(1/\delta)\sqrt{C}}{(t+\gamma-1)\sqrt{d(T+\gamma)}} + \frac{\kappa^3 C^{3/2}\ln(1/\delta)^2}{(t+\gamma-1)^3} + \frac{\kappa\sqrt{C}\ln(1/\delta)^2}{(t+\gamma-1)(T+\gamma)}\right)$$

### B.4 Proof of Lemma 7

For any $s \in [T]$, we recall that $\mathbf{v}_s = \mathbb{E}\left[\text{clip}_\Gamma(\mathbf{g}_s)|\mathcal{F}_{s-1}\right] - \text{clip}_\Gamma(\mathbf{g}_s)$. Since $\mathbb{E}[\mathbf{g}_s|\mathcal{F}_{s-1}] = \nabla F(\mathbf{x}_s)$ and $\text{Cov}[\mathbf{g}_s|\mathcal{F}_{s-1}] \preceq \Sigma$, we obtain the following from Lemma 4

$$\|\mathbb{E}\left[\mathbf{v}_s\mathbf{v}_s^T|\mathcal{F}_{s-1}\right]\|_2 = \|\text{Cov}\left[\text{clip}_\Gamma(\mathbf{g}_s)|\mathcal{F}_{s-1}\right]\| \leq \|\Sigma\|_2 + \frac{\|\nabla F(\mathbf{x}_s)\|^4}{\Gamma^2} + \frac{\|\nabla F(\mathbf{x}_s)\|^2 \text{Tr}(\Sigma)}{\Gamma^2}$$

$$\text{Tr}\left(\mathbb{E}\left[\mathbf{v}_s\mathbf{v}_s^T|\mathcal{F}_{s-1}\right]\right) = \text{Tr}\left(\text{Cov}\left[\text{clip}_\Gamma(\mathbf{g}_s)|\mathcal{F}_{s-1}\right]\right) \leq \text{Tr}(\Sigma)$$

For $s \in [1:T]$ define $\mathbb{E}[\tilde{\mathbf{v}}_s\tilde{\mathbf{v}}_s^T|\mathcal{F}_{s-1}] = \tilde{\Sigma}_s$. Since $\mathbb{1}\{E_s\}$ is $\mathcal{F}_{s-1}$-measurable and $\tilde{\mathbf{v}}_s = \mathbf{v}_s\mathbb{1}\{E_s\}$, it follows that $\tilde{\Sigma}_s = \mathbb{E}\left[\mathbf{v}_s\mathbf{v}_s^T|\mathcal{F}_s\right]\mathbb{1}\{E_s\}$. Hence, we conclude the following from the above inequality

$$\|\tilde{\Sigma}_s\|_2 \leq \|\Sigma\|_2 + \frac{\|\nabla F(\mathbf{x}_s)\|^4 \mathbb{1}\{E_s\}}{\Gamma^2} + \frac{\|\nabla F(\mathbf{x}_s)\|^2 \text{Tr}(\Sigma)\mathbb{1}\{E_s\}}{\Gamma^2}$$

$$\text{Tr}(\tilde{\Sigma}_s) \leq \text{Tr}(\Sigma) \tag{9}$$

Now, for $s \in [t]$, we define $h_s$ as follows:

$$h_s = \langle \tilde{\mathbf{v}}_s, \mathbf{d}_s \rangle \frac{(s+\gamma)^{2A-1}}{(t+\gamma)^{2A-2}}$$

Note that $\mathbb{E}[h_s|\mathcal{F}_{s-1}] = \langle \mathbb{E}[\tilde{\mathbf{v}}_s|\mathcal{F}_{s-1}], \mathbf{d}_s\rangle \frac{(s+\gamma)^{2A-1}}{(t+\gamma)^{2A-2}} = 0$. Furthermore, since $\|\tilde{\mathbf{v}}_s\| \le \|\mathbf{v}_s\| \le 2\Gamma$ and $\|\mathbf{d}_s\| \le \frac{\sqrt{CR_{T,\delta}}}{s+\gamma-1}$

$$
\begin{aligned}
|h_s| &\le 2\Gamma \cdot \frac{\sqrt{CR_{T,\delta}}}{s+\gamma-1} \cdot \frac{(s+\gamma)^{2A-1}}{(t+\gamma)^{2A-2}} \\
&\le 4\Gamma\sqrt{CR_{T,\delta}} \left(\frac{s+\gamma}{t+\gamma}\right)^{2A-2} \\
&\le \frac{4\mu R_{T,\delta}\sqrt{C}}{\ln(K/\delta)}
\end{aligned}
\tag{10}
$$

For $s \in [t]$, define $\sigma_s^2 = \mathbb{E}[h_s^2|\mathcal{F}_{s-1}]$. It follows that,

$$
\begin{aligned}
\sigma_s^2 &= \frac{(s+\gamma)^{4A-2}}{(t+\gamma)^{4A-4}} \mathbf{v}_s^T \tilde{\Sigma}_s \mathbf{v}_s \\
&\le \frac{(s+\gamma)^{4A-2}}{(t+\gamma)^{4A-4}} \|\mathbf{v}_s\|^2 \|\tilde{\Sigma}_s\|_2 \\
&\le 4CR_{T,\delta} \cdot \left(\frac{s+\gamma}{t+\gamma}\right)^{4A-4} \|\tilde{\Sigma}_s\|_2 \\
&\le 4CR_{T,\delta} \left(\frac{s+\gamma}{t+\gamma}\right)^{4A-4} \left(\|\Sigma\|_2 + \frac{\|\nabla F(\mathbf{x}_s)\|^4}{\Gamma^2} + \frac{\|\nabla F(\mathbf{x}_s)\|^2 \|\Sigma\|_2 d_{\text{eff}}}{\Gamma^2}\right)
\end{aligned}
$$

where the last inequality follows from equation (9) and the fact that $d_{\text{eff}} = \text{Tr}(\Sigma)/\|\Sigma\|_2$. We now use the above inequality to control $\sum_{s=1}^t \sigma_s^2 \ln(K/\delta)$ as follows:

$$
\begin{aligned}
\sum_{s=1}^t \sigma_s^2 \ln(K/\delta) &\le 4CR_{T,\delta}\ln(K/\delta)\sum_{s=1}^t \left(\frac{s+\gamma}{t+\gamma}\right)^{4A-4} \|\Sigma\|_2 \\
&\quad + 4CR_{T,\delta}\ln(K/\delta)\sum_{s=1}^t \left(\frac{s+\gamma}{t+\gamma}\right)^{4A-4} \frac{\|\nabla F(\mathbf{x}_s)\|^4}{\Gamma^2} \\
&\quad + 4CR_{T,\delta}\ln(K/\delta)\sum_{s=1}^t \left(\frac{s+\gamma}{t+\gamma}\right)^{4A-4} \frac{\|\nabla F(\mathbf{x}_s)\|^2 \|\Sigma\|_2 d_{\text{eff}}}{\Gamma^2}
\end{aligned}
\tag{11}
$$

We now control each of the three terms in the above inequality as follows

$$
\begin{aligned}
4CR_{T,\delta}\ln(K/\delta)\sum_{s=1}^t \left(\frac{s+\gamma}{t+\gamma}\right)^{4A-4} \|\Sigma\|_2 &\le 4CR_{T,\delta}\ln(K/\delta)\|\Sigma\|_2 t \\
&\le 4CtR_{T,\delta} \cdot \frac{\mu^2 R_{T,\delta}}{(T+\gamma)\sqrt{d_{\text{eff}}}} \\
&\le 4\mu^2 CR_{T,\delta}^2
\end{aligned}
$$

Before controlling the remaining two terms, we recall from (8) in the proof of Lemma **??** that

$$
\begin{aligned}
\|\nabla F(\mathbf{x}_s)\|\mathbb{1}\{E_s\} &\le \frac{\kappa\Gamma\ln(K/\delta)\sqrt{C}}{s+\gamma-1} \\
&\le \frac{2\kappa\Gamma\ln(K/\delta)\sqrt{C}}{s+\gamma}
\end{aligned}
$$

where $\Gamma = \frac{\mu\sqrt{R_{T,\delta}}}{\ln(K/\delta)}$. It follows that

$$
\begin{aligned}
\frac{\|\nabla F(\mathbf{x}_s)\|^4}{\Gamma^2} &\le \frac{16\kappa^4 C^2\Gamma^2\ln(K/\delta)^4}{(s+\gamma)^4} \\
&= \mu^2 R_{T,\delta} \cdot \frac{16\kappa^4 C^2\ln(K/\delta)^2}{(s+\gamma)^4}
\end{aligned}
$$

Thus, we can control the second term in equation (11) as follows

$$4CR_{T,\delta}\ln(K/\delta)\sum_{s=1}^{t}\left(\frac{s+\gamma}{t+\gamma}\right)^{4A-4}\frac{\|\nabla F(\mathbf{x}_s)\|^4}{\Gamma^2} \leq 64\mu^2 CR_{T,\delta}^2\cdot\kappa^4 C^2\ln(K/\delta)^3\sum_{s=1}^{t}\frac{(s+\gamma)^{4A-8}}{(t+\gamma)^{4A-4}}$$

$$\leq 64\mu^2 CR_{T,\delta}^2\cdot\frac{\kappa^4 C^2\ln(K/\delta)^3}{(t+\gamma)^3}$$

$$\leq 64\mu^2 CR_{T,\delta}^2$$

where the second inequality follows by setting $A \geq 2$ and the last inequality follows by setting $\gamma \geq \kappa^{4/3}C^{2/3}\ln(K/\delta)$.

To control the third term in (11), we note that by equation (8) and the definition of $R_{T,\delta}$

$$\frac{\|\nabla F(\mathbf{x}_s)\|^2\|\Sigma\|_2 d_{\text{eff}}}{\Gamma^2} \leq 4\mu^2 R_{T,\delta}\cdot\frac{\kappa^2 C\ln(K/\delta)^2}{(T+\gamma)(s+\gamma)^2}$$

It follows that

$$4CR_{T,\delta}\ln(K/\delta)\sum_{s=1}^{t}\left(\frac{s+\gamma}{t+\gamma}\right)^{4A-4}\frac{\|\nabla F(\mathbf{x}_s)\|^2\|\Sigma\|_2 d_{\text{eff}}}{\Gamma^2} \leq 16\mu^2 CR_{T,\delta}^2\cdot\frac{\kappa^2 C\ln(K/\delta)^3}{T+\gamma}\sum_{s=1}^{t}\frac{(s+\gamma)^{4A-6}}{(t+\gamma)^{4A-4}}$$

$$\leq 16\mu^2 CR_{T,\delta}^2\cdot\frac{\kappa^2 C\ln(K/\delta)^3}{(T+\gamma)(t+\gamma)}$$

$$\leq 16\mu^2 CR_{T,\delta}^2$$

where the second inequality follows by setting $A \geq 3/2$ and the last inequality follows by setting $\gamma \geq \kappa\sqrt{C}\ln(K/\delta)^{3/2}$. Substituting the above bounds into equation (11), we note that

$$\sum_{s=1}^{t}\sigma_s^2\ln(K/\delta) \leq 84\mu^2 CR_{T,\delta}$$

Thus, by Freedman's inequality (Lemma 3), we conclude that the following holds with probability at least $1 - \delta/2$ uniformly for every $t \in [T]$:

$$\sum_{s=1}^{t}\frac{(s+\gamma)^{2A-1}}{(t+\gamma)^{2A-2}}\langle\tilde{\mathbf{v}}_s,\mathbf{d}_s\rangle = \sum_{s=1}^{t}h_s \leq 2\sqrt{\sum_{s=1}^{t}\sigma_s^2\ln(K/\delta)} + 8\mu R_{T,\delta}\sqrt{C} \leq 27R_{T,\delta}\sqrt{C} \quad (12)$$

To prove the second inequality of this lemma, we define $\mathbf{z}_s = \tilde{\mathbf{v}}_s\cdot\left(\frac{s+\gamma}{t+\gamma}\right)^{A-1}$ for $s \in [t]$. Note that $\mathbb{E}[\mathbf{z}_s|\mathcal{F}_{s-1}] = 0$ and $\|\mathbf{z}_s\| \leq \|\tilde{\mathbf{v}}_s\| \leq 2\Gamma$. Define the PSD matrices $\mathbf{G}_s = \mathbb{E}[\mathbf{z}_s\mathbf{z}_s^T|\mathcal{F}_{s-1}] = \left(\frac{s+\gamma}{t+\gamma}\right)^{2A-2}\tilde{\Sigma}_s$. Recalling that $\text{Tr}(\tilde{\Sigma}_s) \leq \text{Tr}(\Sigma)$ and the bound obtained on $\|\tilde{\Sigma}_s|_2$ in equation (9), we infer the following:

$$\text{Tr}(\mathbf{G}_s) \leq \left(\frac{s+\gamma}{t+\gamma}\right)^{2A-2}\text{Tr}(\Sigma)$$

$$\|\mathbf{G}_s\|_2 \leq \left(\frac{s+\gamma}{t+\gamma}\right)^{2A-2}\|\Sigma\|_2 + \left(\frac{s+\gamma}{t+\gamma}\right)^{2A-2}\frac{\|\nabla F(\mathbf{x}_s)\|^4\mathbb{1}\{E_s\}}{\Gamma^2}$$

$$+ \left(\frac{s+\gamma}{t+\gamma}\right)^{2A-2}\frac{\|\nabla F(\mathbf{x}_s)\|^2\text{Tr}(\Sigma)\mathbb{1}\{E_s\}}{\Gamma^2}$$

Substituting (8) into the bound for $\|\mathbf{G}_s\|_2$, we obtain the following

$$\text{Tr}(\mathbf{G}_s) \leq q_s = \left(\frac{s+\gamma}{t+\gamma}\right)^{2A-2}\text{Tr}(\Sigma)$$

$$\|\mathbf{G}_s\|_2 \leq p_s = \left(\frac{s+\gamma}{t+\gamma}\right)^{2A-2}\|\Sigma\|_2 + \frac{(s+\gamma)^{2A-6}}{(t+\gamma)^{2A-2}}\cdot 16\kappa^4 C^2\ln(K/\delta)^2\mu^2 R_{T,\delta}$$

$$+ \frac{(s+\gamma)^{2A-4}}{(t+\gamma)^{2A-2}}\cdot 4\kappa^2 C\ln(K/\delta)^2\|\Sigma\|_2 d_{\text{eff}} \quad (13)$$

By Cauchy Schwarz Inequality,

$$p_s^2 \le 3 \left( \frac{s+\gamma}{t+\gamma} \right)^{4A-4} \|\Sigma\|_2^2 + 3 \cdot \frac{(s+\gamma)^{4A-12}}{(t+\gamma)^{4A-4}} \cdot 256\kappa^8 C^4 \ln(K/\delta)^4 \mu^4 R_{T,\delta}^2$$
$$+ 3 \cdot \frac{(s+\gamma)^{4A-8}}{(t+\gamma)^{4A-4}} \cdot 16\kappa^4 C^2 \ln(K/\delta)^4 \|\Sigma\|_2^2 d_{\text{eff}}^2 \tag{14}$$

Since $T \gtrsim \ln(\ln(d))$, $K = \ln(\ln(T))$ and $q_s \le \text{Tr}(\Sigma) \ \forall s \in [T]$, our choice of $\Gamma$ ensures that the conditions of Corollary 5 are satisfied. Hence, by Corollary 5, we conclude that the following holds with probability $1 - \delta/2$ uniformly for all $t \in [T]$

$$\sum_{s=1}^{t} \|\mathbf{z}_s\|^2 \le 4C_M \Gamma^2 \ln(K/\delta) + C_M \sum_{s=1}^{\text{UP}(t)} q_s + \frac{C_M t}{4\Gamma^2} \sum_{s=1}^{\text{UP}(t)} p_s^2$$

Simplifying the above using equations (13), (14) and the definition of $\Gamma$, we obtain the following inequality which holds with probability at least $1 - \delta/2$ uniformly for every $t \in [T]$:

$$\sum_{s=1}^{t} \|\mathbf{z}_s\|^2 \le 4C_M \mu^2 R_{T,\delta} + C_M \sum_{s=1}^{\text{UP}(t)} \left( \frac{s+\gamma}{t+\gamma} \right)^{2A-2} \text{Tr}(\Sigma) + \frac{3C_M}{4} \sum_{s=1}^{\text{UP}(t)} \left( \frac{s+\gamma}{t+\gamma} \right)^{4A-4} \frac{t\ln(K/\delta)^2 \|\Sigma\|^2}{\mu^2 R_{T,\delta}}$$

$$+ \frac{3C_M}{4} \sum_{s=1}^{\text{UP}(t)} \frac{(s+\gamma)^{4A-12}}{(t+\gamma)^{4A-4}} \cdot 256 t\kappa^8 C^4 \ln(K/\delta)^6 \mu^2 R_{T,\delta}$$

$$+ \frac{3C_M}{4} \sum_{s=1}^{\text{UP}(t)} \frac{(s+\gamma)^{4A-8}}{(t+\gamma)^{4A-4}} \frac{16 t\kappa^4 C^2 \ln(K/\delta)^6 \text{Tr}(\Sigma)^2}{\mu^2 R_{T,\delta}} \tag{15}$$

We now simplify each term in the above inequality by using the fact that $\text{UP}(t) \le \min\{T, 2t\}$. To this end, the second term is simplified as follows by using the fact that $A \ge 1$

$$\sum_{s=1}^{\text{UP}(t)} \left( \frac{s+\gamma}{t+\gamma} \right)^{4A-4} \text{Tr}(\Sigma) \le \text{UP}(t)\text{Tr}(\Sigma) \le \mu^2 R_{T,\delta}$$

We now control the third term as follows using the definition of $R_{T,\delta}$ and the fact that $A \ge 1$:

$$\sum_{s=1}^{\text{UP}(t)} \left( \frac{s+\gamma}{t+\gamma} \right)^{4A-4} \frac{t\ln(K/\delta)^2 \|\Sigma\|^2}{\mu^2 R_{T,\delta}} \le \mu^2 R_{T,\delta} \cdot \frac{t\text{UP}(t)}{d(T+\gamma)^2}$$

$$\le \mu^2 R_{T,\delta}$$

To control the fourth term, we use the fact that $A \ge 3$ and note that for $s \le 2t$, $(s+\gamma) \le 2(t+\gamma)$

$$\sum_{s=1}^{\text{UP}(t)} \frac{(s+\gamma)^{4A-12}}{(t+\gamma)^{4A-4}} \cdot 256 t\kappa^8 C^4 \ln(K/\delta)^6 \mu^2 R_{T,\delta} \le \mu^2 R_{T,\delta} 2^8 \kappa^8 C^4 \ln(K/\delta)^6 \sum_{s=1}^{2t} \frac{(s+\gamma)^{4A-12}}{(t+\gamma)^{4A-4}}$$

$$\le \mu^2 R_{T,\delta} \cdot \frac{t^2 2^{4A-3} \kappa^8 C^4 \ln(K/\delta)^6}{(t+\gamma)^8}$$

$$\le \mu^2 R_{T,\delta} \cdot \frac{2^{4A-3} \kappa^8 C^4 \ln(K/\delta)^6}{(t+\gamma)^6}$$

$$\le \mu^2 R_{T,\delta} 2^{4A-15}$$

where the last inequality follows by setting $\gamma \ge 4\kappa^{4/3} C^{4/3} \ln(K/\delta)$

We control the last term by a similar argument

$$\sum_{s=1}^{\text{UP}(t)} \frac{(s+\gamma)^{4A-8}}{(t+\gamma)^{4A-4}} \frac{16 t\kappa^4 C^2 \ln(K/\delta)^6 \text{Tr}(\Sigma)^2}{\mu^2 R_{T,\delta}} \le \mu^2 R_{T,\delta} \cdot \frac{t}{(T+\gamma)^2} \cdot 2^4 \kappa^4 C^2 \ln(K/\delta)^6 \sum_{s=1}^{2t} \frac{(s+\gamma)^{4A-8}}{(t+\gamma)^{4A-4}}$$

$$\le \frac{t^2}{(T+\gamma)^2 (t+\gamma)^4} \cdot 2^{4A-3} \kappa^4 C^2 \ln(K/\delta)^6$$

$$\le 2^{4A-11} \mu^2 R_{T,\delta}$$

where the last inequality follows by setting $\gamma \geq 4\kappa\sqrt{C}\ln(K/\delta)^{3/2}$. Substituting the obtained bounds into equation (15), we conclude that the following holds with probability at least $1 - \delta/2$ uniformly for every $t \in [T]$:

$$\sum_{s=1}^{t} \left(\frac{s+\gamma}{t+\gamma}\right)^{2A-2} \|\tilde{\mathbf{v}}_s\|^2 = \sum_{s=1}^{t} \|\mathbf{z}_s\|^2 \leq C_M \mu^2 R_{T,\delta} \left(6 + 3 \cdot 2^{4A-13} + 3 \cdot 2^{4A-17}\right)$$

The proof is completed via a union bound.

## C  Analysis for Smooth Strongly Convex Functions Under Quadratic Growth Noise Model

Following a convention similar to that of Section B, let $K = 4\max\{8, C_M, \ln(T)\}$. For $t \geq 1$, define the filtration $\mathcal{F}_t = \sigma\left(\mathbf{x}_1, \mathbf{g}_s | 1 \leq s \leq t\right)$ and $\mathcal{F}_0 = \sigma(\mathbf{x}_1)$. Furthermore, let $\nabla F(\mathbf{x}_t) = \mathsf{clip}_\Gamma(\mathbf{g}_t) + \mathbf{b}_t + \mathbf{v}_t$ where $\mathbf{b}_t = \nabla F(\mathbf{x}_t) - \mathbb{E}[\mathsf{clip}_\Gamma(\mathbf{g}_t)|\mathcal{F}_{t-1}]$ and $\mathbf{v}_t = \mathbb{E}[\mathsf{clip}_\Gamma(\mathbf{g}_t)|\mathcal{F}_{t-1}] - \mathsf{clip}_\Gamma(\mathbf{g}_t)$. As beforem, we note that $\mathbb{E}[\mathbf{v}_t|\mathcal{F}_{t-1}] = 0$ and $\|\mathbf{v}_t\| \leq 2\Gamma$. Hence $\mathbf{v}_t$ is an $\mathcal{F}$ adapted almost surely bounded martingale difference sequence. Now, let $D_t = \|\mathbf{x}_t - \mathbf{x}^*\|$ where $\mathbf{x}^*$ is the unique minimizer of $F$ (guaranteed by strong convexity). We also define $\Sigma_t = \Sigma(\mathbf{x}_t)$ and note that $\|\Sigma_t\| \leq \alpha D_t^2 + \beta$ and $\mathsf{Tr}(\Sigma_t) \leq d_{\mathsf{eff}}\left(\alpha D_t^2 + \beta\right)$. Furthermore $\Sigma_t$ is $\mathcal{F}_{t-1}$ measurable. Let $\eta_t = \frac{A}{t+\gamma}$ where $A \geq 1$ is a numerical constant and $\gamma \geq A\kappa + A - 1$ is a constant depending on $\kappa, d$ and $\ln(1/\delta)$ which we shall specify later. Note that our choice of $\gamma$ ensures that $\eta_t \leq \frac{1}{L+\mu}$ for $t \in [1:T]$ An application of Lemma 5 shows that $D_t$ satisfies the following for every $t \in [1:T]$

$$D_{t+1}^2 \leq \left(\frac{\gamma+1}{t+\gamma}\right)^{2A} D_1^2 + \frac{A2^{2A+1}}{\mu} \sum_{s=1}^{t} \frac{(s+\gamma-1)^{2A-1}}{(t+\gamma)^{2A}} \langle \mathbf{b}_s, \mathbf{x}_s - \mathbf{x}^* \rangle$$

$$+ \frac{A^2 4^{A+1}}{\mu^2} \sum_{s=1}^{t} \|\mathbf{b}_s\|^2 \frac{(s+\gamma)^{2A-2}}{(t+\gamma)^{2A}} + \frac{A2^{2A+1}}{\mu} \sum_{s=1}^{t} \frac{(s+\gamma)^{2A-1}}{(t+\gamma)^{2A}} \langle \mathbf{v}_s, \mathbf{x}_s - \mathbf{x}^* \rangle$$

$$+ \frac{A^2 4^{A+1}}{\mu^2} \sum_{s=1}^{t} \|\mathbf{v}_s\|^2 \frac{(s+\gamma)^{2A-2}}{(t+\gamma)^{2A}}$$

We now define $R_{T,\delta}$ as follows:

$$R_{T,\delta} = (\gamma+1)^2 D_1^2 + \frac{(T+\gamma)\beta}{\mu^2}\left(d_{\mathsf{eff}} + \sqrt{d_{\mathsf{eff}}}\ln(K/\delta)\right)$$

It is easy to see that $\Gamma = \frac{\mu\sqrt{R_{T,\delta}}}{\ln(K/\delta)}$. In our proof of Theorem 1, we shall establish that the following holds with probability at least $1 - \delta$:

$$D_t^2 \leq \frac{CR_{T,\delta}}{(t+\gamma-1)^2} \ \forall\, t \in [1:T+1]$$

where $C > 0$ is an absolute numerical constant to be chosen later. To this end, we define the event $E_t$ and the $\mathcal{F}_t$ measurable random variables $\mathbf{d}_t, \tilde{\mathbf{b}}_t, \tilde{\mathbf{v}}_t$ as follows for $t \in [1:T+1]$:

$$E_t = \left\{D_t^2 \leq \frac{CR_{T,\delta}}{(t+\gamma-1)^2}\right\}$$

$$\mathbf{d}_t = (\mathbf{x}_t - \mathbf{x}^*)\mathbb{1}\{E_t\}$$

$$\tilde{\mathbf{b}}_t = \mathbf{b}_t\mathbb{1}\{E_t\}$$

$$\tilde{\mathbf{v}}_t = \mathbf{v}_t\mathbb{1}\{E_t\}$$

We use the following Lemma to control the bias vector $\tilde{\mathbf{b}}_t$

**Lemma 8** (Bias Control). *The following holds almost surely for every $t \in [1:T]$:*

$$\|\tilde{\mathbf{b}}_t\| \leq \mu\sqrt{R_{T,\delta}}\sum_{j=1}^{7} B_j$$

*where $B_1, \ldots, B_7$ are defined as follows:*

$$B_1 = \frac{1}{T+\gamma},$$

$$B_2 = \frac{4\alpha C\sqrt{d}\ln(\ln(T)/\delta)}{\mu^2(s+\gamma)^2},$$

$$B_3 = \frac{2\kappa\sqrt{C}\ln(\ln(T)/\delta)}{(s+\gamma)\sqrt{d(T+\gamma)}},$$

$$B_4 = \frac{4\kappa C\ln(\ln(T)/\delta)\sqrt{\alpha}}{\mu(s+\gamma)^2},$$

$$B_5 = \frac{8\kappa^3 C^{3/2}\ln(\ln(T)/\delta)^2}{(s+\gamma)^3},$$

$$B_6 = \frac{2\kappa\sqrt{C}\ln(\ln(T)/\delta)^2}{(s+\gamma)(T+\gamma)},$$

$$B_7 = \frac{8\alpha\kappa d\ln(\ln(T)/\delta)^2 C^{3/2}}{\mu^2(s+\gamma)^3}$$

We use the following lemma to control the variance vector $\tilde{\mathbf{v}}_t$. The proof of this lemma, which uses Freedman's inequality and the PAC Bayesian martingale concentration inequality of Corollary 6.

**Lemma 9** (Variance Control). *The following holds with probability at least $1-\delta$ uniformly for every $t \in [T]$ for $A \geq 3$ and $\gamma \geq 4C\max\{\frac{\alpha d_{\text{eff}}}{\mu^2}, \frac{\alpha\ln(K/\delta)}{\mu^2}, \kappa^{4/3}\ln(K/\delta), \kappa\ln(K/\delta)^{3/2}, \frac{\kappa^{2/3}d_{\text{eff}}^{1/3}\alpha^{1/3}}{\mu^{2/3}}\ln(K/\delta)\}$*

$$\sum_{s=1}^{t} \frac{(s+\gamma)^{2A-1}}{(t+\gamma)^{2A-2}} \langle \tilde{\mathbf{v}}_s, \mathbf{d}_s \rangle \lesssim 34 \cdot \mu R_{T,\delta}\sqrt{C}$$

$$\sum_{s=1}^{t} \left(\frac{s+\gamma}{t+\gamma}\right)^{2A-2} \|\tilde{\mathbf{v}}_s\|^2 \lesssim C_M \left(2^{4A-3}\frac{25}{4} + 5 \cdot 2^{4A-11} + 5 \cdot 2^{4A-16} + 5 \cdot 2^{4A-13}\right)\mu^2 R_{T,\delta}$$

*where $C_M$ is the absolute numerical constant defined in Corollary 5.*

Equipped with this bound on the bias and the variance, we now present the complete proof as follows:

### C.1 Proof of Theorem 2

*Proof.* Let $A \geq 3$, $\gamma \geq 4C\max\{\frac{\alpha d_{\text{eff}}}{\mu^2}, \frac{\alpha\ln(K/\delta)}{\mu^2}, \kappa^{4/3}\ln(K/\delta), \kappa\ln(K/\delta)^{3/2}, \frac{\kappa^{2/3}d_{\text{eff}}^{1/3}\alpha^{1/3}}{\mu^{2/3}}\ln(K/\delta)\}$. Now, let $E$ denote the following event

$$E = \{\sum_{s=1}^{t} \frac{(s+\gamma)^{2A-1}}{(t+\gamma)^{2A-2}} \langle \tilde{\mathbf{v}}_s, \mathbf{d}_s \rangle \leq 34 \cdot \mu R_{T,\delta}\sqrt{C} \ \forall t \in [T]$$

$$\sum_{s=1}^{t} \left(\frac{s+\gamma}{t+\gamma}\right)^{2A-2} \|\tilde{\mathbf{v}}_s\|^2 \leq 53 \cdot C_M\mu^2 R_{T,\delta} \ \forall t \in [T]\}$$

Note that by Lemma 9, $\mathbb{P}(E) \geq 1-\delta$. We now claim that $\mathbb{P}\left(\bigcap_{t=1}^{T+1} E_t | E\right) = 1$, i.e., conditioned on the event $E$, the following holds almost surely for every $t \in [1:T+1]$

$$D_t^2 \leq \frac{CR_{T,\delta}}{(t+\gamma-1)^2} \ \forall t \in [1:T+1]$$

We prove the above claim by induction. Note that the claim is trivially true for $t=1$ as $R_{T,\delta} \geq (\gamma+1)^2 D_1^2$. Now, consider any $t \in [1:T]$ and suppose the claim holds for some $1 \leq s \leq t$.

Recall that by Lemma 5

$$(t+\gamma)^2 D_{t+1}^2 \le \frac{(\gamma+1)^{2A}}{(t+\gamma)^{2A-2}} D_1^2 + \frac{A 2^{2A+1}}{\mu} \sum_{s=1}^{t} \frac{(s+\gamma-1)^{2A-1}}{(t+\gamma)^{2A-2}} \langle \mathbf{b}_s, \mathbf{x}_s - \mathbf{x}^* \rangle$$

$$+ \frac{A^2 4^{A+1}}{\mu^2} \sum_{s=1}^{t} \|\mathbf{b}_s\|^2 \frac{(s+\gamma)^{2A-2}}{(t+\gamma)^{2A-2}} + \frac{A 2^{2A+1}}{\mu} \sum_{s=1}^{t} \frac{(s+\gamma)^{2A-1}}{(t+\gamma)^{2A-2}} \langle \mathbf{v}_s, \mathbf{x}_s - \mathbf{x}^* \rangle$$

$$+ \frac{A^2 4^{A+1}}{\mu^2} \sum_{s=1}^{t} \|\mathbf{v}_s\|^2 \frac{(s+\gamma)^{2A-2}}{(t+\gamma)^{2A-2}}$$

Under the induction hypothesis, $\mathbb{1}\{E_s\} = 1 \ \forall s \in [t]$. Hence, Under the induction hypothesis, $\mathbb{1}\left\{D_s^2 \le \frac{C R_{T,\delta}}{(s+\gamma-1)(s+\gamma-2)}\right\} = 1$ and thus, $\mathbf{d}_s = \mathbf{x}_s - \mathbf{x}^*, \mathbf{b}_s = \tilde{\mathbf{b}}_s, \mathbf{v}_s = \tilde{\mathbf{v}}_s \ \forall \ 1 \le s \le t$. Substituting this transformation into the above inequality, we obtain the following:

$$(t+\gamma)^2 D_{t+1}^2 \le \underbrace{\frac{(\gamma+1)^{2A}}{(t+\gamma)^{2A-2}} D_1^2}_{\textcircled{1}} + \underbrace{\frac{A 2^{2A+1}}{\mu} \sum_{s=1}^{t} \frac{(s+\gamma)^{2A-1}}{(t+\gamma)^{2A-2}} \langle \tilde{\mathbf{v}}_s, \mathbf{d}_s \rangle}_{\textcircled{2}}$$

$$+ \underbrace{\frac{A^2 4^{A+1}}{\mu^2} \sum_{s=1}^{t} \|\tilde{\mathbf{v}}_s\|^2 \frac{(s+\gamma)^{2A-2}}{(t+\gamma)^{2A-2}}}_{\textcircled{3}} + \underbrace{\sum_{s=1}^{t} \frac{(s+\gamma)^{2A-1}}{(t+\gamma)^{2A-2}} \left\langle \tilde{\mathbf{b}}_s, \mathbf{d}_s \right\rangle}_{\textcircled{4}}$$

$$+ \underbrace{\frac{A^2 4^{A+1}}{\mu^2} \sum_{s=1}^{t} \|\tilde{\mathbf{b}}_s\|^2 \frac{(s+\gamma)^{2A-2}}{(t+\gamma)^{2A-2}}}_{\textcircled{5}} \qquad (16)$$

We now bound each of the terms in the RHS as follows.

**Bounding ①** Since $A \ge 1$ and $t \ge 1$,

$$\textcircled{1} = \frac{(\gamma+1)^{2A}}{(t+\gamma)^{2A-2}} D_1^2 \le (\gamma+1)^2 D_1^2 \le R_{T,\delta}$$

**Bounding ②** Since $\gamma$ and $A$ satisfy the conditions of Lemma 7 and we have conditioned on the event $E$, it follows that:

$$\frac{A 2^{2A+1}}{\mu} \sum_{s=1}^{t} \frac{(s+\gamma)^{2A-1}}{(t+\gamma)^{2A-2}} \langle \tilde{\mathbf{v}}_s, \mathbf{d}_s \rangle \le 17 A 4^{A+1} R_{T,\delta} \sqrt{C}$$

**Bounding ③** Since $\gamma$ and $A$ satisfy the conditions of Lemma 7 and we have conditioned on the event $E$, it follows that:

$$\frac{A^2 4^{A+1}}{\mu^2} \sum_{s=1}^{t} \left(\frac{s+\gamma}{t+\gamma}\right)^{2A-2} \|\tilde{\mathbf{v}}_s\|^2 \le A^2 2^{2A+2} C_M \left(2^{4A-3} \frac{25}{4} + 5 \cdot 2^{4A-11} + 5 \cdot 2^{4A-16} + 5 \cdot 2^{4A-13}\right) R_{T,\delta}$$

**Bounding ④** Since $\mathbb{1}\{E_s\} = 1$

$$\|\mathbf{d}_s\| \le \frac{\sqrt{C R_{T,\delta}}}{s+\gamma-1} \le \frac{2\sqrt{C R_{T,\delta}}}{s+\gamma}$$

Hence, by Lemma 8

$$\frac{A 2^{2A+1}}{\mu} \sum_{s=1}^{t} \left\langle \tilde{\mathbf{b}}_s, \mathbf{d}_s \right\rangle \frac{(s+\gamma)^{2A-1}}{(t+\gamma)^{2A-2}} \le A 2^{2A+2} R_{T,\delta} \sqrt{C} \sum_{s=1}^{t} \left(\frac{s+\gamma}{t+\gamma}\right)^{2A-2} \sum_{j=1}^{7} B_j$$

We now control the first term

$$\sum_{s=1}^{t} \left(\frac{s+\gamma}{t+\gamma}\right)^{2A-2} B_1 = \frac{1}{T+\gamma} \sum_{s=1}^{t} \left(\frac{s+\gamma}{t+\gamma}\right)^{2A-2}$$

$$\leq \frac{t}{T+\gamma} \leq 1$$

where the first inequality follows from the fact that $A \geq 1$ and $s \leq t$.

We now control the second term

$$\sum_{s=1}^{t} \left(\frac{s+\gamma}{t+\gamma}\right)^{2A-2} B_2 \leq \frac{4\alpha C \sqrt{d} \ln(K/\delta)}{\mu^2} \sum_{s=1}^{t} \frac{(s+\gamma)^{2A-4}}{(t+\gamma)^{2A-2}}$$

$$\leq \frac{4\alpha C \sqrt{d} \ln(K/\delta)}{\mu^2(t+\gamma)} \leq 1$$

where the first inequality follows from the fact that $A \geq 2$ and $s \leq t$ and the second inequality follows by setting $\gamma \geq \frac{4\alpha C \sqrt{d} \ln(K/\delta)}{\mu^2}$.

We now bound the third term as follows:

$$\sum_{s=1}^{t} \left(\frac{s+\gamma}{t+\gamma}\right)^{2A-2} B_3 \leq \frac{2\kappa\sqrt{C} \ln(K/\delta)}{\sqrt{d}(T+\gamma)} \sum_{s=1}^{t} \frac{(s+\gamma)^{2A-3}}{(t+\gamma)^{2A-2}}$$

$$\leq \frac{2\kappa\sqrt{C} \ln(K/\delta)}{\sqrt{d}(T+\gamma)} \leq 1$$

where we use the fact that $A \geq 2$ and set $\gamma \geq \frac{4\kappa^2 \ln(K/\delta)^2}{d}$.

We now bound the fourth term as follows:

$$\sum_{s=1}^{t} \left(\frac{s+\gamma}{t+\gamma}\right)^{2A-2} B_4 \leq \frac{4\kappa C \ln(K/\delta)\sqrt{\alpha}}{\mu} \sum_{s=1}^{t} \frac{(s+\gamma)^{2A-4}}{(t+\gamma)^{2A-2}}$$

$$\leq \frac{4\kappa C \ln(K/\delta)\sqrt{\alpha}}{\mu(t+\gamma)} \leq 1$$

where $A \geq 2$ and $\gamma \geq \frac{4\kappa C \ln(K/\delta)\sqrt{\alpha}}{\mu}$

We now bound the fifth term as follows

$$\sum_{s=1}^{t} \left(\frac{s+\gamma}{t+\gamma}\right)^{2} B_5 \leq 8\kappa^3 C^{3/2} \ln(K/\delta)^2 \sum_{s=1}^{t} \frac{(s+\gamma)^{2A-5}}{(t+\gamma)^{[2A-2]}}$$

$$\leq \frac{8\kappa^3 C^{3/2} \ln(K/\delta)^2}{(t+\gamma)^2} \leq 1$$

where $A \geq 3$ and $\gamma \geq 4\kappa^{3/2} C^{3/4} \ln(K/\delta)$.

We now bound the sixth term as follows

$$\sum_{s=1}^{t} \left(\frac{s+\gamma}{t+\gamma}\right)^{2A-2} B_6 \leq \frac{2\kappa\sqrt{C} \ln(K/\delta)^2}{T+\gamma} \sum_{s=1}^{t} \frac{(s+\gamma)^{2A-3}}{(t+\gamma)^{2A-2}}$$

$$\leq \frac{2\kappa\sqrt{C} \ln(K/\delta)^2}{T+\gamma} \leq 1$$

where $A \geq 3$ and $\gamma \geq 2\kappa\sqrt{C} \ln(K/\delta)^2$

Finally, we control the seventh term as follows

$$\sum_{s=1}^{t} \left(\frac{s+\gamma}{t+\gamma}\right)^{2A-2} B_7 \leq \frac{8\alpha\kappa d \ln(K/\delta)^2 C^{3/2}}{\mu^2} \sum_{s=1}^{t} \frac{(s+\gamma)^{2A-5}}{(t+\gamma)^{2A-2}}$$

$$\leq \frac{8\alpha\kappa d \ln(K/\delta)^2 C^{3/2}}{\mu^2(t+\gamma)^2} \leq 1$$

where $A \geq 3$ and $\gamma \geq \frac{4\sqrt{\alpha\kappa d}\ln(K/\delta)C^{3/4}}{\mu}$. Putting it all together, it follows that

$$④ \leq 7A4^{A+1}R_{T,\delta}\sqrt{C}$$

by setting $\gamma$ as follows

$$\gamma \geq 4C\max\left\{\frac{\alpha\sqrt{d}\ln(K/\delta)}{\mu^2}, \frac{\kappa^2\ln(K/\delta)^2}{d}, \frac{\kappa\sqrt{\alpha}\ln(K/\delta)}{\mu}, \kappa^{3/2}\ln(K/\delta), \kappa\ln(K/\delta)^2, \frac{\sqrt{\kappa\alpha d}\ln(K/\delta)}{\mu}\right\}$$

**Bounding ⑤**   By Lemma 8 and Jensen's inequality

$$\|\tilde{\mathbf{b}}_s\|^2 \leq 7\mu^2 R_{T,\delta}\sum_{j=1}^{7}B_j^2$$

It follows that

$$\frac{A^2 2^{2A+2}}{\mu^2}\sum_{s=1}^{t}\|\tilde{\mathbf{b}}_s\|^2\left(\frac{s+\gamma}{t+\gamma}\right)^{2A-2} \leq 7A^2 2^{2A+2}R_{T,\delta}\sum_{s=1}^{t}\left(\frac{s+\gamma}{t+\gamma}\right)^{2A-2}\sum_{j=1}^{7}B_j^2$$

The first term is controlled as follows using the fact that $A \geq 1$

$$\sum_{s=1}^{t}\left(\frac{s+\gamma}{t+\gamma}\right)^{2A-2}B_1^2 = \sum_{s=1}^{t}\frac{1}{(T+\gamma)^2} \leq 1$$

The second term is controlled as

$$\sum_{s=1}^{t}\left(\frac{s+\gamma}{t+\gamma}\right)^{2A-2}B_2^2 \leq \frac{16\alpha^2 C^2 d\ln(K/\delta)^2}{\mu^4}\sum_{s=1}^{t}\frac{(s+\gamma)^{2A-6}}{(t+\gamma)^{2A-2}}$$

$$\leq \frac{16\alpha^2 C^2 d\ln(K/\delta)^2}{\mu^4(t+\gamma)^3} \leq 1$$

where $A \geq 3$ and $\gamma \geq \frac{2^{4/3}\alpha^{2/3}C^{2/3}d^{1/3}\ln(K/\delta)^{2/3}}{\mu^{4/3}}$.

The third term is controlled as

$$\sum_{s=1}^{t}\left(\frac{s+\gamma}{t+\gamma}\right)^{2A-2}B_3^2 = \frac{4\kappa^2 C\ln(K/\delta)^2}{d(T+\gamma)}\sum_{s=1}^{t}\frac{(s+\gamma)^{2A-4}}{(t+\gamma)^{2A-2}}$$

$$\leq \frac{4\kappa^2 C\ln(K/\delta)^2}{d(t+\gamma)(T+\gamma)} \leq 1$$

where the last inequality follows because $\gamma \geq \kappa\sqrt{\frac{C}{d}}\ln(K/\delta)$

The fourth term is controlled as

$$\sum_{s=1}^{t}\left(\frac{s+\gamma}{t+\gamma}\right)^{2A-2}B_4^2 \leq \frac{16\kappa^2 C^2\ln(K/\delta)^2\alpha}{\mu^2}\sum_{s=1}^{t}\frac{(s+\gamma)^{2A-6}}{(t+\gamma)^{2A-2}}$$

where $A \geq 3$ and $\gamma \geq \frac{2^{4/3}\kappa^{2/3}C^{2/3}\ln(K/\delta)^{2/3}\alpha^{1/3}}{\mu^{2/3}}$ For controlling the fifth term, we set $A \geq 4$ to obtain

$$\sum_{s=1}^{t}\left(\frac{s+\gamma}{t+\gamma}\right)^{2A-2}B_5^2 = \kappa^6 C^3\ln(K/\delta)^4\sum_{s=1}^{t}\frac{(s+\gamma)^{2A-8}}{(t+\gamma)^{2A-2}}$$

$$\leq \frac{\kappa^6 C^3\ln(K/\delta)^4}{(\gamma+1)^5} \leq 1$$

where the last inequality uses the fact that $\gamma \geq \kappa^{6/5}C^{3/5}\ln(K/\delta)^{4/5}$

To control the sixth term, we use the fact that $A \geq 2$ to obtain

$$\sum_{s=1}^{t} \left(\frac{s+\gamma}{t+\gamma}\right)^{2A-2} B_6^2 = \frac{\kappa^2 C \ln(K/\delta)^4}{(T+\gamma)^2} \sum_{s=1}^{t} \frac{(s+\gamma)^{2A-4}}{(t+\gamma)^{2A-2}}$$

$$\leq \frac{\kappa^2 C \ln(K/\delta)^4}{(\gamma+1)^3} \leq 1$$

where the last inequality uses the fact that $\gamma \geq \kappa^{2/3} C^{1/3} \ln(K/\delta)^{4/3}$

To control the seventh term, we set $A \geq 4$ to obtain the following:

$$\sum_{s=1}^{t} \left(\frac{s+\gamma}{t+\gamma}\right)^{2A-2} B_6^2 = \frac{64\alpha^2\kappa^2 d \ln(K/\delta)^4 C^3}{\mu^4} \sum_{s=1}^{t} \frac{(s+\gamma)^{2A-8}}{(t+\gamma)^{2A-2}}$$

$$\leq \frac{64\alpha^2\kappa^2 d \ln(K/\delta)^4 C^3}{\mu^4(t+\gamma)^5} \leq 1$$

where $\gamma \geq \frac{2^{6/5}\alpha^{2/5}\kappa^{2/5}d^{1/5}\ln(K/\delta)^{4/5}C^{3/5}}{\mu^{4/5}}$ From the obtained bounds, we conclude that $\circledS \leq 49A^2 4^{A+1} R_{T,\delta}$.

Now, we set $A = 4$ and $\gamma$ as follows:

$$\gamma = \max\left\{\frac{\alpha d}{\mu^2}, \frac{\alpha\sqrt{d}\ln(K/\delta)}{\mu^2}, \frac{\kappa\sqrt{\alpha}\ln(K/\delta)}{\mu^2}, \frac{\sqrt{\kappa\alpha d}\ln(K/\delta)}{\mu}, \frac{\kappa^{2/3}d^{1/3}\alpha^{1/3}\ln(K/\delta)}{\mu^{2/3}}, \kappa^{3/2}\ln(K/\delta), \kappa\ln(K/\delta)^2, \frac{\kappa^2\ln(K/\delta)}{d}\right\}$$

Under this setting of $A$ and $\gamma$, we obtain the following

$$(t+\gamma)^2 D_{t+1}^2 \leq \circled{1} + \circled{2} + \circled{3} + \circled{4} + \circled{5}$$

$$\leq R_{T,\delta}\left[1 + A^2 2^{2A+2} C_M \left(2^{4A-3}\frac{25}{4} + 5 \cdot 2^{4A-11} + 5 \cdot 2^{4A-16} + 5 \cdot 2^{4A-13}\right)\right.$$

$$\left. + 49A^2 4^{A+1} + 24A 4^{A+1}\sqrt{\tilde{C}}\right]$$

$$\leq R_{T,\delta}\left(802817 + 6946816C_M + 98304\sqrt{\tilde{C}}\right)$$

$$\leq CR_{T,\delta}$$

where the second inequality holds due to our choice of $A$ and $\gamma$ and the last inequality is obtained by setting $C = \left(\sqrt{802817 + 6946816C_M} + 98304\right)^2$. It follows that

$$D_{t+1}^2 \leq \frac{CR_{T,\delta}}{(t+\gamma)^2}$$

Thus, we have proved by induction that conditioned on $E$, $D_t^2 \leq \frac{CR_{T,\delta}}{(t+\gamma-1)^2}$ for every $t \in [T+1]$. In particular, the following holds with probability at least $1 - \delta$:

$$D_{T+1}^2 \leq C\left(\frac{\gamma+1}{T+\gamma}\right)^2 D_1^2 + \frac{C\beta\left(d_{\text{eff}} + \sqrt{d_{\text{eff}}}\ln(K/\delta)\right)}{\mu^2(T+\gamma)}$$

$\square$

## C.2 Proof of Lemma 8

Following the same steps as in that of the proof of Lemma 6, we use Lemma 4 and the fact that $\text{Cov}[\mathbf{g}_t|\mathcal{F}_{t-1}] = \Sigma_t$ to obtain:

$$\|\tilde{\mathbf{b}}_s\| \leq \underbrace{\frac{\|\Sigma_s\|\sqrt{d_{\text{eff}}}\mathbb{1}\{E_s\}}{\Gamma}}_{\circledA} + \underbrace{\frac{\|\nabla F(\mathbf{x}_s)\|\sqrt{\|\Sigma_s\|}\mathbb{1}\{E_s\}}{\Gamma}}_{\circledB} + \underbrace{\frac{\|\nabla F(\mathbf{x}_s)\|^3\mathbb{1}\{E_s\}}{\Gamma^2}}_{\circledC} + \underbrace{\frac{\|\Sigma_s\|d_{\text{eff}}\|\nabla F(\mathbf{x}_s)\|\mathbb{1}\{E_s\}}{\Gamma^2}}_{\circledD}$$

**Bounding Ⓐ** Note that by Assumption QG 2$^{\text{nd}}$ Moment

$$\|\Sigma_s\|_2 \mathbb{1}\{E_s\} \le (\beta + \alpha D_s^2)\mathbb{1}\{E_s\}$$
$$\le \beta + \frac{4\alpha C R_{T,\delta}}{(s+\gamma)^2}$$

It follows that

$$\frac{\|\Sigma_s\|_2\sqrt{d}\,\mathbb{1}\{E_s\}}{\Gamma} \le \frac{\beta\sqrt{d_{\text{eff}}}\ln(K/\delta)}{\mu\sqrt{R_{T,\delta}}} + \frac{4\alpha C\ln(K/\delta)\sqrt{R_{T,\delta}d_{\text{eff}}}}{\mu(s+\gamma)^2}$$

Since $\beta\sqrt{d_{\text{eff}}}\ln(K/\delta) \le \frac{\mu^2 R_{T,\delta}}{T+\gamma}$, we obtain

$$Ⓐ = \frac{\|\Sigma\|_s\sqrt{d}\,\mathbb{1}\{E_s\}}{\Gamma} \le \mu\sqrt{R_{T,\delta}}\left(\frac{1}{T+\gamma} + \frac{4\alpha C\ln(K/\delta)\sqrt{R_{T,\delta}d_{\text{eff}}}}{\mu^2(s+\gamma)^2}\right)$$

**Bounding Ⓑ** Note that by equation (8),

$$\frac{\|\nabla F(\mathbf{x}_s)\|\mathbb{1}\{E_s\}}{\Gamma} \le \frac{2\kappa\sqrt{C}\ln(K/\delta)}{s+\gamma}$$

Furthermore, by Assumption QG 2$^{\text{nd}}$ Moment and the definition of $E_s$

$$\sqrt{\|\Sigma_s\|_2}\mathbb{1}\{E_s\} \le \sqrt{\beta} + \frac{2\sqrt{\alpha C R_{T,\delta}}}{s+\gamma}$$

Recalling that $\beta \le \frac{\mu^2 R_{T,\delta}}{d_{\text{eff}}(T+\gamma)}$,

$$\frac{\|\nabla F(\mathbf{x}_s)\|\sqrt{\|\Sigma_s\|_2}\mathbb{1}\{E_s\}}{\Gamma} \le \frac{2\kappa\sqrt{C}\ln(K/\delta)\mu\sqrt{R_{T,\delta}}}{(s+\gamma)\sqrt{d_{\text{eff}}(T+\gamma)}} + \frac{4\kappa C\ln(K/\delta)\sqrt{\alpha R_{T,\delta}}}{(s+\gamma)^2}$$
$$\le \mu\sqrt{R_{T,\delta}}\left(\frac{2\kappa\sqrt{C}\ln(K/\delta)}{(s+\gamma)\sqrt{d_{\text{eff}}(T+\gamma)}} + \frac{4\kappa C\ln(K/\delta)\sqrt{\alpha}}{\mu(s+\gamma)^2}\right)$$

**Bounding Ⓒ** By equation (8),

$$\frac{\|\nabla F(\mathbf{x}_s)\|^3\mathbb{1}\{E_s\}}{\Gamma^2} \le \mu\sqrt{R_{T,\delta}}\cdot\frac{8\kappa^3 C^{3/2}\ln(K/\delta)^2}{(s+\gamma)^3}$$

**Bounding Ⓓ** Since $\beta d \le \frac{\mu^2 R_{T,\delta}}{T+\gamma}$, it follows tat

$$\frac{\|\nabla F(\mathbf{x}_s)\|\|\Sigma_s\|_2 d_{\text{eff}}\mathbb{1}\{E_s\}}{\Gamma^2} \le \frac{2\kappa\sqrt{C}\ln(K/\delta)^2}{\mu\sqrt{R_{T,\delta}}(s+\gamma)}\left(\beta d + \frac{4\alpha C R_{T,\delta}d}{(s+\gamma)^2}\right)$$
$$\le \frac{2\kappa\sqrt{C}\ln(K/\delta)^2\mu\sqrt{R_{T,\delta}}}{(s+\gamma)(T+\gamma)} + \frac{8\alpha\kappa d\ln(K/\delta)^2 C^{3/2}\sqrt{R_{T,\delta}}}{\mu(s+\gamma)^3}$$
$$\le \mu\sqrt{R_{T,\delta}}\left(\frac{2\kappa\sqrt{C}\ln(K/\delta)^2}{(s+\gamma)(T+\gamma)} + \frac{8\alpha\kappa d\ln(K/\delta)^2 C^{3/2}}{\mu^2(s+\gamma)^3}\right)$$

Hence, we conclude that

$$\|\tilde{\mathbf{b}}_t\| \le Ⓐ + Ⓑ + Ⓒ + Ⓓ \le \mu\sqrt{R_{T,\delta}}\sum_{j=1}^{7}B_j$$

where $B_1, \ldots, B_7$ are defined as follows:

$$B_1 = \frac{1}{T + \gamma},$$

$$B_2 = \frac{4\alpha C \sqrt{d} \ln(\ln(T)/\delta)}{\mu^2 (s + \gamma)^2},$$

$$B_3 = \frac{2\kappa \sqrt{C} \ln(\ln(T)/\delta)}{(s + \gamma)\sqrt{d(T + \gamma)}},$$

$$B_4 = \frac{4\kappa C \ln(\ln(T)/\delta)\sqrt{\alpha}}{\mu(s + \gamma)^2},$$

$$B_5 = \frac{8\kappa^3 C^{3/2} \ln(\ln(T)/\delta)^2}{(s + \gamma)^3},$$

$$B_6 = \frac{2\kappa \sqrt{C} \ln(\ln(T)/\delta)^2}{(s + \gamma)(T + \gamma)},$$

$$B_7 = \frac{8\alpha\kappa d \ln(\ln(T)/\delta)^2 C^{3/2}}{\mu^2 (s + \gamma)^3}$$

### C.3 Proof of Lemma 9

As before, for $s \in [1 : T]$ define $\mathbb{E}[\tilde{\mathbf{v}}_s \tilde{\mathbf{v}}_s^T | \mathcal{F}_{s-1}] = \tilde{\Sigma}_s$. Following the same steps as in that of the proof of Lemma 7, we use Lemma 4 and the fact that $\mathsf{Cov}[\mathbf{g}_t | \mathcal{F}_{t-1}] = \Sigma_t$ to obtain:

$$\|\tilde{\Sigma}_s\|_2 \le \|\Sigma_s\|_2 \mathbb{1}\{E_s\} + \frac{\|\nabla F(\mathbf{x}_s)\|^4 \mathbb{1}\{E_s\}}{\Gamma^2} + \frac{\|\nabla F(\mathbf{x}_s)\|^2 \mathsf{Tr}(\Sigma_s) \mathbb{1}\{E_s\}}{\Gamma^2}$$

$$\le \mathbb{1}\{E_t\} \left(\beta + \alpha D_s^2\right) + \frac{\|\nabla F(\mathbf{x}_s)\|^4 \mathbb{1}\{E_s\}}{\Gamma^2} + \frac{\mathbb{1}\{E_s\} \|\nabla F(\mathbf{x}_s)\|^2 d_{\text{eff}}}{\Gamma^2} \left(\beta + \alpha D_s^2\right)$$

$$\le \beta + \frac{4\alpha C R_{T,\delta}}{(s + \gamma)^2} + \frac{\|\nabla F(\mathbf{x}_s)\|^4 \mathbb{1}\{E_s\}}{\Gamma^2} + \frac{\|\nabla F(\mathbf{x}_t)\|^2 d_{\text{eff}} \mathbb{1}\{E_s\}}{\Gamma^2} \left(\beta + \frac{4\alpha C R_{T,\delta}}{(s + \gamma)^2}\right) \tag{17}$$

where the second inequality follows from Assumption QG 2$^{\text{nd}}$ Moment and the second inequality follows by definition of $E_s$

Furthermore, since $\mathsf{clip}_\Gamma$ is a convex projection, the following holds:

$$\mathsf{Tr}(\tilde{\Sigma}_s) \le \mathsf{Tr}(\Sigma_s) \mathbb{1}\{E_s\}$$

$$\le d_{\text{eff}} \left(\beta + \alpha D_s^2\right) \mathbb{1}\{E_s\}$$

$$\le \beta d_{\text{eff}} + \frac{4\alpha d C R_{T,\delta}}{(s + \gamma)^2} \tag{18}$$

Now, for $s \in [t]$, we define $h_s$ as follows:

$$h_s = \langle \tilde{\mathbf{v}}_s, \mathbf{d}_s \rangle \frac{(s + \gamma)^{2A-1}}{(t + \gamma)^{2A-2}}$$

Note that $\mathbb{E}[h_s | \mathcal{F}_{s-1}] = 0$. Furthermore, since $\|\tilde{\mathbf{v}}_s\| \le 2\Gamma$ and $\|\mathbf{d}_s\| \le \frac{\sqrt{C R_{T,\delta}}}{s + \gamma - 1}$

$$|h_s| \le 2\Gamma \cdot \frac{\sqrt{C R_{T,\delta}}}{s + \gamma - 1} \cdot \frac{(s + \gamma)^{2A-1}}{(t + \gamma)^{2A-2}}$$

$$\le 4\Gamma \sqrt{C R_{T,\delta}} \left(\frac{s + \gamma}{t + \gamma}\right)^{2A-2}$$

$$\le \frac{4\mu R_{T,\delta}\sqrt{C}}{\ln(K/\delta)} \tag{19}$$

For $s \in [t]$, define $\sigma_s^2 = \mathbb{E}[h_s^2 | \mathcal{F}_{s-1}]$. It follows that,

$$\sigma_s^2 = \frac{(s+\gamma)^{4A-2}}{(t+\gamma)^{4A-4}} \mathbf{v}_s^T \tilde{\Sigma}_s \mathbf{v}_s$$

$$\leq \frac{(s+\gamma)^{4A-2}}{(t+\gamma)^{4A-4}} \|\mathbf{v}_s\|^2 \|\tilde{\Sigma}_s\|_2$$

$$\leq 4CR_{T,\delta} \cdot \left(\frac{s+\gamma}{t+\gamma}\right)^{4A-4} \|\tilde{\Sigma}_s\|_2$$

$$\leq 4CR_{T,\delta} \left(\frac{s+\gamma}{t+\gamma}\right)^{4A-4} \left[\beta + \frac{4\alpha CR_{T,\delta}}{(s+\gamma)^2} + \frac{\|\nabla F(\mathbf{x}_s)\|^4 \mathbb{1}\{E_s\}}{\Gamma^2} + \frac{\|\nabla F(\mathbf{x}_t)\|^2 d_{\text{eff}} \mathbb{1}\{E_s\}}{\Gamma^2} \left(\beta + \frac{4\alpha CR_{T,\delta}}{(s+\gamma)^2}\right)\right]$$

where the last inequality follows from equation (9) and the fact that $d_{\text{eff}} = \text{Tr}(\Sigma)/\|\Sigma\|_2$. We now use the above inequality to control $\sum_{s=1}^t \sigma_s^2 \ln(K/\delta)$ as follows:

$$\sum_{s=1}^t \sigma_s^2 \ln(K/\delta) \leq 4CR_{T,\delta} \ln(K/\delta) \sum_{s=1}^t \left(\frac{s+\gamma}{t+\gamma}\right)^{4A-4} \beta$$

$$+ 4CR_{T,\delta} \ln(K/\delta) \sum_{s=1}^t \frac{(s+\gamma)^{4A-6}}{(t+\gamma)^{4A-4}} 4\alpha CR_{T,\delta}$$

$$+ 4CR_{T,\delta} \ln(K/\delta) \sum_{s=1}^t \left(\frac{s+\gamma}{t+\gamma}\right)^{4A-4} \frac{\|\nabla F(\mathbf{x}_s)\|^4 \mathbb{1}\{E_s\}}{\Gamma^2}$$

$$+ 4CR_{T,\delta} \ln(K/\delta) \sum_{s=1}^t \left(\frac{s+\gamma}{t+\gamma}\right)^{4A-4} \frac{\|\nabla F(\mathbf{x}_s)\|^2 \mathbb{1}\{E_s\} \beta d_{\text{eff}}}{\Gamma^2}$$

$$+ 4CR_{T,\delta} \ln(K/\delta) \sum_{s=1}^t \frac{(s+\gamma)^{4A-6}}{(t+\gamma)^{4A-4}} \frac{4\|\nabla F(\mathbf{x}_s)\|^2 \mathbb{1}\{E_s\} \alpha d CR_{T,\delta}}{\Gamma^2} \quad (20)$$

We now control each of the five terms in the above inequality as follows

$$4CR_{T,\delta} \ln(K/\delta) \sum_{s=1}^t \left(\frac{s+\gamma}{t+\gamma}\right)^{4A-4} \beta \leq 4CR_{T,\delta} \ln(K/\delta) \beta t$$

$$\leq 4CtR_{T,\delta} \cdot \frac{\mu^2 R_{T,\delta}}{(T+\gamma)\sqrt{d_{\text{eff}}}}$$

$$\leq 4\mu^2 CR_{T,\delta}^2$$

To control the second term,

$$4CR_{T,\delta} \ln(K/\delta) \sum_{s=1}^t \frac{(s+\gamma)^{4A-6}}{(t+\gamma)^{4A-4}} 4\alpha CR_{T,\delta} \leq 16CR_{T,\delta}^2 \mu^2 \frac{\alpha C \ln(K/\delta)}{\mu^2 (t+\gamma)}$$

$$\leq 16CR_{T,\delta}^2 \mu^2$$

where the second inequality follows by setting $A \geq 3/2$ and the last inequality follows by setting $\gamma \geq \frac{\alpha C \ln(K/\delta)}{\mu^2}$ Before controlling the remaining terms, we recall from (8) in the proof of Lemma 6 that

$$\|\nabla F(\mathbf{x}_s)\| \mathbb{1}\{E_s\} \leq \frac{\kappa \Gamma \ln(K/\delta)\sqrt{C}}{s+\gamma-1}$$

$$\leq \frac{2\kappa \Gamma \ln(K/\delta)\sqrt{C}}{s+\gamma}$$

where $\Gamma = \frac{\mu\sqrt{R_{T,\delta}}}{\ln(K/\delta)}$. It follows that

$$\frac{\|\nabla F(\mathbf{x}_s)\|^4 \mathbb{1}\{E_s\}}{\Gamma^2} \leq \frac{16\kappa^4 C^2 \Gamma^2 \ln(K/\delta)^4}{(s+\gamma)^4}$$

$$= \mu^2 R_{T,\delta} \cdot \frac{16\kappa^4 C^2 \ln(K/\delta)^2}{(s+\gamma)^4}$$

Thus, we can control the third term in equation (20) as follows

$$4CR_{T,\delta}\ln(K/\delta)\sum_{s=1}^{t}\left(\frac{s+\gamma}{t+\gamma}\right)^{4A-4}\frac{\|\nabla F(\mathbf{x}_s)\|^4\mathbb{1}\{E_s\}}{\Gamma^2} \le 64\mu^2 CR_{T,\delta}^2\cdot\kappa^4 C^2\ln(K/\delta)^3\sum_{s=1}^{t}\frac{(s+\gamma)^{4A-8}}{(t+\gamma)^{4A-4}}$$

$$\le 64\mu^2 CR_{T,\delta}^2\cdot\frac{\kappa^4 C^2\ln(K/\delta)^3}{(t+\gamma)^3}$$

$$\le 64\mu^2 CR_{T,\delta}^2$$

where the second inequality follows by setting $A \ge 2$ and the last inequality follows by setting $\gamma \ge \kappa^{4/3}C^{2/3}\ln(K/\delta)$.

To control the fourth term in (20), we note that by equation (8) and the definition of $R_{T,\delta}$

$$\frac{\|\nabla F(\mathbf{x}_s)\|^2 d_{\mathrm{eff}}\beta\mathbb{1}\{E_s\}}{\Gamma^2}\le 4\mu^2 R_{T,\delta}\cdot\frac{\kappa^2 C\ln(K/\delta)^2}{(T+\gamma)(s+\gamma)^2}$$

It follows that

$$4CR_{T,\delta}\ln(K/\delta)\sum_{s=1}^{t}\left(\frac{s+\gamma}{t+\gamma}\right)^{4A-4}\frac{\|\nabla F(\mathbf{x}_s)\|^2\beta d_{\mathrm{eff}}}{\Gamma^2}\le 16\mu^2 CR_{T,\delta}^2\cdot\frac{\kappa^2 C\ln(K/\delta)^3}{T+\gamma}\sum_{s=1}^{t}\frac{(s+\gamma)^{4A-6}}{(t+\gamma)^{4A-4}}$$

$$\le 16\mu^2 CR_{T,\delta}^2\cdot\frac{\kappa^2 C\ln(K/\delta)^3}{(T+\gamma)(t+\gamma)}$$

$$\le 16\mu^2 CR_{T,\delta}^2$$

where the second inequality follows by setting $A \ge 3/2$ and the last inequality follows by setting $\gamma \ge \kappa\sqrt{C}\ln(K/\delta)^{3/2}$.

To control the fifth term in equation (20), we proceed as follows:

$$4CR_{T,\delta}\ln(K/\delta)\sum_{s=1}^{t}\frac{(s+\gamma)^{4A-6}}{(t+\gamma)^{4A-4}}\frac{4\|\nabla F(\mathbf{x}_s)\|^2\mathbb{1}\{E_s\}\alpha dCR_{T,\delta}}{\Gamma^2}$$

$$\le 64\mu^2 CR_{T,\delta}^2\frac{\alpha d\kappa^2 C^2\ln(K/\delta)^3}{\mu^2}\sum_{s=1}^{t}\frac{(s+\gamma)^{4A-8}}{(t+\gamma)^{4A-4}}$$

$$\le 64\mu^2 CR_{T,\delta}^2\cdot\frac{\alpha d\kappa^2 C^2\ln(K/\delta)^3}{\mu^2(t+\gamma)^3}$$

$$\le 64\mu^2 CR_{T,\delta}^2$$

where the second inequality follows by setting $A \ge 2$ and the last inequality follows by setting $\gamma \ge \frac{\alpha^{1/3}d_{\mathrm{eff}}^{1/3}\kappa^{2/3}C^{2/3}\ln(K/\delta)}{\mu^{2/3}}$ Substituting the above bounds into equation (20), we note that

$$\sum_{s=1}^{t}\sigma_s^2\ln(K/\delta)\le 164\mu^2 CR_{T,\delta}$$

Thus, by Freedman's inequality (Lemma 3), we conclude that the following holds with probability at least $1 - \delta/2$ uniformly for every $t \in [T]$:

$$\sum_{s=1}^{t}\frac{(s+\gamma)^{2A-1}}{(t+\gamma)^{2A-2}}\langle\tilde{\mathbf{v}}_s,\mathbf{d}_s\rangle = \sum_{s=1}^{t}h_s \le 2\sqrt{\sum_{s=1}^{t}\sigma_s^2\ln(K/\delta)} + 8\mu R_{T,\delta}\sqrt{C}\le 34R_{T,\delta}\sqrt{C} \quad (21)$$

To prove the second inequality of this lemma, we define $\mathbf{z}_s = \tilde{\mathbf{v}}_s\cdot\left(\frac{s+\gamma}{t+\gamma}\right)^{A-1}$ for $s \in [t]$. Note that $\mathbb{E}[\mathbf{z}_s|\mathcal{F}_{s-1}] = 0$ and $\|\mathbf{z}_s\| \le \|\tilde{\mathbf{v}}_s\| \le 2\Gamma$. Define the PSD matrices $\mathbf{G}_s = \mathbb{E}[\mathbf{z}_s\mathbf{z}_s^T|\mathcal{F}_{s-1}] = \left(\frac{s+\gamma}{t+\gamma}\right)^{2A-2}\tilde{\Sigma}_s$. Recalling the bounds obtained on $\|\tilde{\Sigma}_s\|_2$ and $\mathsf{Tr}(\tilde{\Sigma}_s)$ in equations (17) and (18), we

infer the following:

$$\mathsf{Tr}(\mathbf{G}_s) \le \left(\frac{s+\gamma}{t+\gamma}\right)^{2A-2} \mathsf{Tr}(\Sigma_s)\mathbb{1}\{E_s\}$$

$$\le \left(\frac{s+\gamma}{t+\gamma}\right)^{2A-2} \beta d_{\mathsf{eff}} + \frac{(s+\gamma)^{2A-4}}{(t+\gamma)^{2A-2}} 4\alpha d_{\mathsf{eff}} C R_{T,\delta}$$

$$\|\mathbf{G}_s\|_2 = \left(\frac{s+\gamma}{t+\gamma}\right)^{2A-2} \|\tilde{\Sigma}_s\|_2$$

$$\le \left(\frac{s+\gamma}{t+\gamma}\right)^{2A-2} \beta + \frac{(s+\gamma)^{2A-4}}{(t+\gamma)^{2A-2}} 4\alpha C R_{T,\delta} + \left(\frac{s+\gamma}{t+\gamma}\right)^{2A-2} \frac{\|\nabla F(\mathbf{x}_s)\|^4 \mathbb{1}\{E_s\}}{\Gamma^2}$$

$$+ \left(\frac{s+\gamma}{t+\gamma}\right)^{2A-2} \frac{\|\nabla F(\mathbf{x}_s)\|^2 \mathbb{1}\{E_s\} \beta d_{\mathsf{eff}}}{\Gamma^2} + \frac{(s+\gamma)^{2A-4}}{(t+\gamma)^{2A-2}} \frac{\|\nabla F(\mathbf{x}_s)\|^2 \mathbb{1}\{E_s\} 4\alpha d_{\mathsf{eff}} C R_{T,\delta}}{\Gamma^2}$$

Substituting equation (8) into the bound for $\|\mathbf{G}_s\|_2$, we obtain the following

$$\mathsf{Tr}(\mathbf{G}_s) \le q_s = \left(\frac{s+\gamma}{t+\gamma}\right)^{2A-2} \beta d_{\mathsf{eff}} + \frac{(s+\gamma)^{2A-4}}{(t+\gamma)^{2A-2}} 4\alpha d_{\mathsf{eff}} C R_{T,\delta}$$

$$\|\mathbf{G}_s\|_2 \le p_s = \left(\frac{s+\gamma}{t+\gamma}\right)^{2A-2} \beta + \frac{(s+\gamma)^{2A-4}}{(t+\gamma)^{2A-2}} \cdot 4\alpha C R_{T,\delta} + \frac{(s+\gamma)^{2A-6}}{(t+\gamma)^{2A-2}} \cdot 16\kappa^4 C^2 \ln(K/\delta)^2 \mu^2 R_{T,\delta}$$

$$\le \frac{(s+\gamma)^{2A-4}}{(t+\gamma)^{2A-2}} \cdot 4\beta d_{\mathsf{eff}} \kappa^2 C \ln(K/\delta)^2 + \frac{(s+\gamma)^{2A-6}}{(t+\gamma)^{2A-2}} \cdot 16\alpha d_{\mathsf{eff}} R_{T,\delta} \kappa^2 C^2 \ln(K/\delta)^2 \quad (22)$$

By Cauchy Schwarz inequality,

$$p_s^2 \le 5\left(\frac{s+\gamma}{t+\gamma}\right)^{4A-4} \beta^2 + 5 \cdot \frac{(s+\gamma)^{4A-8}}{(t+\gamma)^{4A-4}} 16\alpha^2 C^2 R_{T,\delta}^2 + 5 \cdot \frac{(s+\gamma)^{4A-12}}{(t+\gamma)^{4A-4}} \cdot 256\kappa^8 C^4 \ln(K/\delta)^4 \mu^4 R_{T,\delta}^2$$

$$+ 5 \cdot \frac{(s+\gamma)^{4A-8}}{(t+\gamma)^{4A-4}} \cdot 16\beta^2 d_{\mathsf{eff}}^2 \kappa^4 C^2 \ln(K/\delta)^4 + 5 \cdot \frac{(s+\gamma)^{4A-12}}{(t+\gamma)^{4A-4}} \cdot 256\alpha^2 d_{\mathsf{eff}}^2 R_{T,\delta}^2 \kappa^4 C^4 \ln(K/\delta)^4$$

$$(23)$$

Since $T \gtrsim \ln(\ln(d))$, $K = \ln(\ln(T))$, our choice of $\Gamma$ and the definition of $R_{T,\delta}$ ensures that the conditions of Corollary 5 are satisfied. Hence, by Corollary 5, we conclude that the following holds with probability $1 - \delta/2$ uniformly for all $t \in [T]$

$$\sum_{s=1}^{t} \|\mathbf{z}_s\|^2 \le 4C_M \Gamma^2 \ln(K/\delta)^2 + C_M \sum_{s=1}^{\mathsf{UP}(t)} Q_s + \frac{C_M t}{4\Gamma^2} \sum_{s=1}^{t} P_s^2$$

Simplyfing the above using equations (22), (23) and the definition of $\Gamma$, we obtain the following:

$$\sum_{s=1}^{t} \|\mathbf{z}_s\|^2 \le 4C_M \mu^2 R_{T,\delta} + C_M \sum_{s=1}^{\mathsf{UP}(t)} \left(\frac{s+\gamma}{t+\gamma}\right)^{2A-2} \beta d_{\mathsf{eff}} + C_M \sum_{s=1}^{\mathsf{UP}(t)} \frac{(s+\gamma)^{2A-4}}{(t+\gamma)^{2A-2}} \cdot 4\alpha d_{\mathsf{eff}} C R_{T,\delta}$$

$$+ \frac{5C_M}{4} \sum_{s=1}^{\mathsf{UP}(t)} \left(\frac{s+\gamma}{t+\gamma}\right)^{4A-4} \frac{\beta^2 t \ln(K/\delta)^2}{\mu^2 R_{T,\delta}} + \frac{5C_M}{4} \sum_{s=1}^{\mathsf{UP}(t)} \frac{(s+\gamma)^{4A-8}}{(t+\gamma)^{4A-4}} \cdot \frac{16\alpha^2 C^2 R_{T,\delta} t \ln(K/\delta)^2}{\mu^2}$$

$$+ \frac{5C_M}{4} \sum_{s=1}^{\mathsf{UP}(t)} \frac{(s+\gamma)^{4A-12}}{(t+\gamma)^{4A-4}} \cdot 256\kappa^8 C^4 \ln(K/\delta)^6 t \mu^2 R_{T,\delta}$$

$$+ \frac{5C_M}{4} \sum_{s=1}^{\mathsf{UP}(t)} \frac{(s+\gamma)^{4A-8}}{(t+\gamma)^{4A-4}} \cdot \frac{\beta^2 d_{\mathsf{eff}}^2 t}{\mu^2 R_{T,\delta}} \cdot 16\kappa^4 C^2 \ln(K/\delta)^6$$

$$+ \frac{5C_M}{4} \sum_{s=1}^{\mathsf{UP}(t)} \frac{(s+\gamma)^{4A-12}}{(t+\gamma)^{4A-4}} \cdot \frac{256\kappa^4 C^4 \alpha^2 d_{\mathsf{eff}}^2 \ln(K/\delta)^6 R_{T,\delta}}{\mu^2} \quad (24)$$

We now simplify each term in the above inequality by using the fact that $\mathsf{UP}(t) \leq \min\{T, 2t\}$. To this end, the second term is simplified as follows by using $A \geq 1$

$$\sum_{s=1}^{\mathsf{UP}(t)} \left(\frac{s+\gamma}{t+\gamma}\right)^{4A-4} \beta d_{\text{eff}} \leq \mathsf{UP}(t)\beta d_{\text{eff}} \leq \mu^2 R_{T,\delta}$$

We now control the third term by noting that for $s \leq 2t$, $s + \gamma \leq 2t + \gamma \leq 2(t+\gamma)$:

$$\sum_{s=1}^{t} \frac{(s+\gamma)^{2A-4}}{(t+\gamma)^{2A-2}} \cdot 4\alpha d_{\text{eff}} C R_{T,\delta} \leq \mu^2 R_{T,\delta} \cdot \frac{2^{2A-2}\alpha d_{\text{eff}}}{\mu^2} \sum_{s=1}^{2t} \frac{1}{(t+\gamma)^2}$$

$$\leq 2^{2A-1}\mu^2 R_{T,\delta} \cdot \frac{\alpha d t}{\mu^2(t+\gamma)^2}$$

$$\leq 2^{2A-3}\mu^2 R_{T,\delta}$$

where the last inquality follows by setting $\gamma \geq \frac{4\alpha C d_{\text{eff}}}{\mu^2}$.

We now control the fourth term as follows:

$$\sum_{s=1}^{\mathsf{UP}(t)} \left(\frac{s+\gamma}{t+\gamma}\right)^{4A-4} \frac{\beta^2 t \ln(K/\delta)^2}{\mu^2 R_{T,\delta}} \leq \mu^2 R_{T,\delta} \cdot \frac{\mathsf{UP}(t)}{d(T+\gamma)^2} \leq \mu^2 R_{T,\delta}$$

We now control the fifth term as follows:

$$\frac{16\alpha^2 C^2 R_{T,\delta} t \ln(K/\delta)^2}{\mu^2} \sum_{s=1}^{\mathsf{UP}(t)} \frac{(s+\gamma)^{4A-8}}{(t+\gamma)^{4A-4}} \leq \mu^2 R_{T,\delta} \cdot \frac{2^{4A-4}\alpha^2 C^2 \ln(K/\delta)^2 t}{\mu^4} \sum_{s=1}^{2t} \frac{1}{(t+\gamma)^4}$$

$$\leq \mu^2 R_{T,\delta} \cdot \frac{\alpha^2 C^2 \ln(K/\delta)^2 2^{4A-5}}{\mu^4(t+\gamma)^2}$$

$$\leq 2^{4A-9}\mu^2 R_{T,\delta}$$

where the last inequality uses $\gamma \geq \frac{4\alpha C \ln(K/\delta)}{\mu^2}$

We now simplify the sixth term as follows:

$$\sum_{s=1}^{\mathsf{UP}(t)} \frac{(s+\gamma)^{4A-12}}{(t+\gamma)^{4A-4}} \cdot 256\mu^2 R_{T,\delta}\kappa^8 C^4 \ln(K/\delta)^6 t \leq \mu^2 R_{T,\delta} t \cdot 2^{4A-4}\kappa^8 C^4 \ln(K/\delta)^6 \sum_{s=1}^{2t} \frac{1}{(t+\gamma)^8}$$

$$\leq \frac{\mu^2 R_{T,\delta}}{(t+\gamma)^6} \cdot 2^{4A-3}\kappa^8 C^4 \ln(K/\delta)^6$$

$$\leq 2^{4A-15}\mu^2 R_{T,\delta}$$

where the last inequality follows by setting $\gamma \geq 4\kappa^{4/3}C^{2/3}\ln(K/\delta)$.

We control the seventh term as follows:

$$\sum_{s=1}^{\mathsf{UP}(t)} \frac{(s+\gamma)^{4A-8}}{(t+\gamma)^{4A-4}} \cdot \frac{\beta^2 d_{\text{eff}}^2 t}{\mu^2 R_{T,\delta}} \cdot 16\kappa^4 C^2 \ln(K/\delta)^6 \leq \mu^2 R_{T,\delta} \cdot \frac{2^4\kappa^4 C^2 \ln(K/\delta)^6 t}{(T+\gamma)^2} \sum_{s=1}^{2t} \frac{(s+\gamma)^{4A-8}}{(t+\gamma)^{4A-4}}$$

$$\leq \mu^2 R_{T,\delta} \cdot \frac{2^{4A-3}\kappa^4 C^2 \ln(K/\delta)^6}{(t+\gamma)^4}$$

$$\leq 2^{4A-11}\mu^2 R_{T,\delta}$$

where $\gamma \geq 4\kappa\sqrt{C}\ln(K/\delta)^{3/2}$.

We use a similar argument to simplify the final term as follows:

$$\sum_{s=1}^{\mathsf{UP}(t)} \frac{t(s+\gamma)^{4A-12}}{(t+\gamma)^{4A-4}} \cdot \frac{2^8\alpha^2 d_{\text{eff}}^2 R_{T,\delta}\kappa^4 C^4 \ln(K/\delta)^6}{\mu^2} \leq \mu^2 R_{T,\delta} \cdot \frac{2^{4A-3}\alpha^2 d_{\text{eff}}^2 \kappa^4 C^4 \ln(K/\delta)^6}{\mu^4(t+\gamma)^6}$$

$$\leq 2^{4A-15}\mu^2 R_{T,\delta}$$

where $\gamma \geq \frac{4\kappa^{2/3} d_{\text{eff}}^{1/3} C^{2/3} \ln(K/\delta)}{\mu^{2/3}}$. We now set $A \geq 3$ and $\gamma$ as follows:

$$\gamma \geq 4C \max\{\frac{\alpha d_{\text{eff}}}{\mu^2}, \frac{\alpha \ln(K/\delta)}{\mu^2}, \kappa^{4/3} \ln(K/\delta), \kappa \ln(K/\delta)^{3/2}, \frac{\kappa^{2/3} d_{\text{eff}}^{1/3} \alpha^{1/3}}{\mu^{2/3}} \ln(K/\delta)\}$$

Under these parameter settings, we substitute the obtained bounds into equation (15), we conclude that the following holds with probability at least $1 - \delta/2$ uniformly for every $t \in [T]$:

$$\sum_{s=1}^{t} \left(\frac{s+\gamma}{t+\gamma}\right)^{2A-2} \|\tilde{\mathbf{v}}_s\|^2 = \sum_{s=1}^{t} \|\mathbf{z}_s\|^2 \leq C_M \mu^2 R_{T,\delta} \left(2^{4A-3}\frac{25}{4} + 5 \cdot 2^{4A-11} + 5 \cdot 2^{4A-16} + 5 \cdot 2^{4A-13}\right)$$

The proof is completed via a union bound.

# D    Analysis for Smooth Convex Functions

Let $d_{\text{eff}} = \frac{\text{Tr}(\Sigma)}{\|\Sigma\|_2}$. Following a convention similar to that of Section B, let $K = 4\max\{8, C_M, \ln(T)\}$. For $t \geq 1$, define the filtration $\mathcal{F}_t = \sigma(\mathbf{x}_1, \mathbf{g}_s | 1 \leq s \leq t)$ and $\mathcal{F}_0 = \sigma(\mathbf{x}_1)$. Furthermore, let $\nabla F(\mathbf{x}_t) = \text{clip}_\Gamma(\mathbf{g}_t) + \mathbf{b}_t + \mathbf{v}_t$ where $\mathbf{b}_t = \nabla F(\mathbf{x}_t) - \mathbb{E}[\text{clip}_\Gamma(\mathbf{g}_t)|\mathcal{F}_{t-1}]$ and $\mathbf{v}_t = \mathbb{E}[\text{clip}_\Gamma(\mathbf{g}_t)|\mathcal{F}_{t-1}] - \text{clip}_\Gamma(\mathbf{g}_t)$. As beforem, we note that $\mathbb{E}[\mathbf{v}_t|\mathcal{F}_{t-1}] = 0$ and $\|\mathbf{v}_t\| \leq 2\Gamma$. Hence $\mathbf{v}_t$ is an $\mathcal{F}$ adapted almost surely bounded martingale difference sequence. Now, let $D_t = \|\mathbf{x}_t - \mathbf{x}^*\|$ where $\mathbf{x}^*$ is the minimizer of $F$ considered in the statement of Theorem 3. Using the smoothness and convexity properties of $F$, we first prove the following intermediate average iterate guarantee:

**Lemma 10** (Intermediate Average Iterate Guarantee). *The following holds for $\eta \leq 1/2L$*

$$F(\hat{\mathbf{x}}_T) - F(\mathbf{x}^*) \leq \frac{D_1^2}{2\eta T} + \frac{1}{T}\sum_{t=1}^{T} \langle \mathbf{b}_t, \mathbf{x}_t - \mathbf{x}^* \rangle + \frac{1}{T}\sum_{t=1}^{T} \langle \mathbf{v}_t, \mathbf{x}_t - \mathbf{x}^* \rangle + \frac{2\eta}{T}\sum_{t=1}^{T} \|\mathbf{b}_t\|^2 + \frac{2\eta}{T}\sum_{t=1}^{T} \|\mathbf{v}_t\|^2$$

Define the events $E_t$ and the random vectors $\mathbf{d}_t$, $\tilde{\mathbf{b}}_t$ and $\tilde{\mathbf{v}}_t$ as follows for $t \in [T]$:

$$E_t = \{D_t \leq 2D_1\}$$
$$\mathbf{d}_t = (\mathbf{x}_t - \mathbf{x}^*)\mathbb{1}\{E_t\}$$
$$\tilde{\mathbf{b}}_t = \mathbf{b}_t\mathbb{1}\{E_t\}$$
$$\tilde{\mathbf{v}}_t = \mathbf{v}_t\mathbb{1}\{E_t\}$$

We use the following lemma to control the bias

**Lemma 11** (Bias Control). *For every $t \in [T]$, $\|\tilde{\mathbf{b}}_t\| \leq B$ where $B$ is defined as follows:*

$$B = \frac{\|\Sigma\|_2\sqrt{d_{\text{eff}}}}{\Gamma} + \frac{2LD_1\sqrt{\|\Sigma\|_2}}{\Gamma} + \frac{8L^3D_1^3}{\Gamma^2} + \frac{2\|\Sigma\|_2 d_{\text{eff}}LD_1}{\Gamma^2}$$

We use the following lemma to control the varince

**Lemma 12** (Variance Control). *Let $V \geq 0$ be defined as follows:*

$$V = \|\Sigma\|_2 + \frac{16L^4D_1^4}{\Gamma^2} + \frac{4L^2D_1^2\|\Sigma\|_2 d_{\text{eff}}}{\Gamma^2}$$

*Then the following holds with probability at least $1 - \delta$ uniformly for every $t \in [T]$*

$$\sum_{s=1}^{t} \langle \tilde{\mathbf{v}}_t, \mathbf{d}_t \rangle \leq 4D_1\sqrt{Vt\ln(K/\delta)} + 8\Gamma D_1\ln(K/\delta)$$

$$\sum_{s=1}^{t} \|\tilde{\mathbf{v}}_t\|^2 \leq C_M g^2 T$$

*where $C_M$ is a numerical constant and $g^2$ is defined as follows*

$$g^2 = \|\Sigma\|_2 d_{\text{eff}} + \frac{4\Gamma^2 \ln(K/\delta)^2}{T} + \frac{V^2T}{4\Gamma^2}$$

Let $E$ denote the following event

$$E = \{ \sum_{s=1}^{t} \langle \tilde{\mathbf{v}}_s, \mathbf{d}_s \rangle \leq 4D_1 \sqrt{Vt \ln(K/\delta)} + 8\Gamma D_1 \ln(K/\delta) \quad \forall\, t \in [T]$$

$$\sum_{s=1}^{t} \|\tilde{\mathbf{v}}_s\|^2 \leq C_M g^2 T \quad \forall t \in [T]\}$$

We define the constant $A$ as follows:

$$A = \|\Sigma\|_2 \sqrt{d_{\text{eff}}} + LD_1 \sqrt{\|\Sigma\|_2} = \sqrt{\|\Sigma\|_2} \left( \sqrt{\text{Tr}(\Sigma)} + LD_1 \right)$$

We now set the clipping level $\Gamma = \sqrt{\frac{AT}{\ln(K/\delta)}}$. For this choice of $\Gamma$, we now obtain the following bound on $B$:

$$B \leq \frac{\|\Sigma\|_2 \sqrt{d_{\text{eff}}}}{\Gamma} + \frac{2LD_1 \sqrt{\|\Sigma\|_2}}{\Gamma} + \frac{8L^3 D_1^3}{\Gamma^2} + \frac{2\|\Sigma\|_2 d_{\text{eff}} LD_1}{\Gamma^2}$$

$$\leq 2\sqrt{\frac{A \ln(K/\delta)}{T}} + \frac{2LD_1 \ln(K/\delta)}{AT} \left( \|\Sigma\|_2 d_{\text{eff}} + L^2 D_1^2 \right) = B' \tag{25}$$

Similarly, we bound the value of $V$ as follows:

$$V \leq \|\Sigma\|_2 + \frac{16L^4 D^4 \ln(K/\delta)}{AT} + \frac{4L^2 D^2 \|\Sigma\|_2 d \ln(K/\delta)}{AT} = V' \tag{26}$$

Equipped with the above inequality, we then bound the value of $g$ as:

$$g \leq \sqrt{\|\Sigma\|_2 d_{\text{eff}}} + \frac{2\Gamma \ln(K/\delta)}{\sqrt{T}} + \frac{V\sqrt{T}}{2\Gamma}$$

$$\leq \sqrt{\|\Sigma\|_2 d_{\text{eff}}} + 2\sqrt{A \ln(K/\delta)} + \frac{V\sqrt{\ln(K/\delta)}}{2\sqrt{A}}$$

$$\leq \sqrt{\|\Sigma\|_2 d_{\text{eff}}} + 2\sqrt{A \ln(K/\delta)} + \frac{\|\Sigma\|_2 \sqrt{\ln(K/\delta)}}{2\sqrt{A}} + \frac{8L^4 D^4 \ln(K/\delta)^{3/2}}{A^{3/2}T} + \frac{2L^2 D^2 \|\Sigma\|_2 d_{\text{eff}} \ln(K/\delta)^{3/2}}{A^{3/2}T}$$

$$\leq \sqrt{\|\Sigma\|_2 d_{\text{eff}}} + 3\sqrt{A \ln(K/\delta)} + \frac{8L^4 D^4 \ln(K/\delta)^{3/2}}{A^{3/2}T} + \frac{2L^2 D^2 \|\Sigma\|_2 d_{\text{eff}} \ln(K/\delta)^{3/2}}{A^{3/2}T} = g' \tag{27}$$

We prove the following lemma to control the growth of the iterates $D_t$.

**Lemma 13** (Iterate Bound). *Let* $\eta \leq c \min\{1/2L, D_1/B'T, D_1/g'\sqrt{T}\}$ *where* $c = \frac{1}{\sqrt{8C_M + 330}}$. *Then, conditioned on the event* $E$, $D_t \leq 2D_1 \;\forall t \in [T]$.

Equipped with the above lemmas, we now present a proof of the following theorem, which is a formal restatement of Theorem 3

**Theorem 7** (Smooth Convex Objectives). *Let Convexity, L-smoothness and Bdd. $2^{\text{nd}}$ Moment be satisfied. Then, for any* $\delta \in (0, 1/2)$ *and* $T \geq \ln(\ln(d))$, *there exists an* $\eta \in (0, 1/2L]$ *such that the average iterate of Algorithm 1 run for* $T$ *iterations with step-size* $\eta_t = \eta$ *and clipping level* $\Gamma = \sqrt{\frac{T\sqrt{\|\Sigma\|_2}(\sqrt{\text{Tr}(\Sigma)}+LD_1)}{\ln(\ln(T)/\delta)}}$ *satisfies the following with probability at least* $1 - \delta$ :

$$F(\hat{\mathbf{x}}_T) - F(\mathbf{x}^*) \lesssim D_1 \sqrt{\frac{\text{Tr}(\Sigma) + \sqrt{\|\Sigma\|_2}\left(\sqrt{\text{Tr}(\Sigma)} + LD_1\right)\ln(\ln(T)/\delta)}{T}} + \frac{LD_1^2}{T}$$

$$+ \frac{LD_1^2 \ln(\ln(T)/\delta)}{T}\sqrt{\frac{\text{Tr}(\Sigma) + L^2 D_1^2}{\|\Sigma\|_2}} + \frac{L^2 D_1^3 \ln(\ln(T)/\delta)^{3/2}}{T^{3/2}}\left[\frac{\text{Tr}(\Sigma) + L^2 D_1^2}{\|\Sigma\|^3}\right]^{1/4}$$

## D.1 Proof of Theorem 7

We condition on the event $E$ and let $\eta = \frac{c}{2} \min\{\frac{1}{2L}, \frac{D_1}{B'T}, \frac{D_1}{g'\sqrt{T}}\}$ where $c = \frac{1}{\sqrt{8C_M + 330}}$. Note that this choice of $\eta$ satisfies the requirements of Lemma 10 and Lemma 13. By Lemma 10, the following holds:

$$F(\hat{\mathbf{x}}_T) - F(\mathbf{x}^*) \leq \frac{D_1^2}{2\eta T} + \frac{1}{T}\sum_{t=1}^T \langle \mathbf{b}_t, \mathbf{x}_t - \mathbf{x}^* \rangle + \frac{1}{T}\sum_{t=1}^T \langle \mathbf{v}_t, \mathbf{x}_t - \mathbf{x}^* \rangle + \frac{2\eta}{T}\sum_{t=1}^T \|\mathbf{b}_t\|^2 + \frac{2\eta}{T}\sum_{t=1}^T \|\mathbf{v}_t\|^2$$

By Lemma 13, $\mathbb{1}\{E_t\} = 1 \; \forall t \in [T]$. Hence, the following holds.

$$F(\hat{\mathbf{x}}_T) - F(\mathbf{x}^*) \leq \frac{D_1^2}{2\eta T} + \frac{1}{T}\sum_{t=1}^T \langle \tilde{\mathbf{b}}_t, \mathbf{d}_t \rangle + \frac{1}{T}\sum_{t=1}^T \langle \tilde{\mathbf{v}}_t, \mathbf{d}_t \rangle + \frac{2\eta}{T}\sum_{t=1}^T \|\tilde{\mathbf{b}}_t\|^2 + \frac{2\eta}{T}\sum_{t=1}^T \|\tilde{\mathbf{v}}_t\|^2$$

$$\leq \frac{D_1^2}{2\eta T} + 2BD_1 + 2\eta B^2 + 2\eta C_M g^2 + 4D_1\sqrt{\frac{V\ln(K/\delta)}{T}} + \frac{8D_1\Gamma\ln(K/\delta)}{T}$$

$$\leq \frac{D_1^2}{\eta T} + 3\eta B^2 T + 2\eta C_M g^2 + 4D_1\sqrt{\frac{V\ln(K/\delta)}{T}} + 8D_1\sqrt{\frac{A\ln(K/\delta)}{T}}$$

Where the second inequality uses Lemma 11 and the definition of the event $E$ and the third inequality uses $ab \leq a^2 + b^2/4$. For the rest of the proof, we shall use $C$ to denote an absolute numerical constant whose value can differ at every step. By our choice of the step-size

$$\frac{D^2}{\eta T} \leq \frac{CLD_1^2}{T} + CD_1 B' + \frac{CD_1 g'}{\sqrt{T}}$$

$$3\eta B^2 T \leq CD_1 B'$$

$$2\eta C_M g^2 \leq \frac{CD_1 g'}{\sqrt{T}}$$

Hence, conditioned on the event $E$, the following holds:

$$F(\hat{\mathbf{x}}_T) - F(\mathbf{x}^*) \leq \frac{CLD_1^2}{T} + CD_1 B' + CD_1 g'\sqrt{T} + CD_1\sqrt{\frac{V'\ln(K/\delta)}{T}} + CD_1\sqrt{\frac{A\ln(K/\delta)}{T}}$$

Substituting the values of $g'$, $B'$ and $V'$, we obtain the following:

$$F(\hat{\mathbf{x}}_T) - F(\mathbf{x}^*) \leq \frac{CLD_1^2}{T} + CD_1\sqrt{\frac{A\ln(K/\delta)}{T}} + \frac{CLD_1^2\ln(K/\delta)}{T} \cdot \left[\frac{\mathsf{Tr}(\Sigma) + L^2 D_1^2}{A}\right]$$

$$+ \frac{CLD_1^2\ln(K/\delta)}{T} \cdot \sqrt{\frac{\mathsf{Tr}(\Sigma) + L^2 D^2}{A}} + \frac{CL^2 D_1^3\ln(K/\delta)^{3/2}}{T^{3/2}} \cdot \left[\frac{\mathsf{Tr}(\Sigma) + L^2 D_1^2}{A^{3/2}}\right]$$

Substituting the value of $A$, we conclude that the following inequality holds almost surely conditioned on the event $E$

$$F(\hat{\mathbf{x}}_T) - F(\mathbf{x}^*) \lesssim D_1\sqrt{\frac{\mathsf{Tr}(\Sigma) + \sqrt{\|\Sigma\|_2}\left(\sqrt{\mathsf{Tr}(\Sigma)} + LD_1\right)\ln(\ln(T)/\delta)}{T}} + \frac{LD_1^2}{T}$$

$$+ \frac{LD_1^2\ln(\ln(T)/\delta)}{T}\sqrt{\frac{\mathsf{Tr}(\Sigma) + L^2 D_1^2}{\|\Sigma\|_2}} + \frac{L^2 D_1^3\ln(\ln(T)/\delta)^{3/2}}{T^{3/2}}\left[\frac{\mathsf{Tr}(\Sigma) + L^2 D_1^2}{\|\Sigma\|^3}\right]^{1/4}$$

The prooof is completed by observing that $\mathbb{P}(E) \geq 1 - \delta$ by Lemma 12 which implies that the above inequality also holds with probability at least $1 - \delta$

## D.2 Proof of Lemma 10

*Proof.* Since $\Pi_{\mathcal{C}}$ is a contractive operator

$$D_{t+1}^2 = \|\mathbf{x}_{t+1} - \mathbf{x}^*\|^2 \leq D_t^2 - 2\eta\langle\nabla F(\mathbf{x}_t) - \mathbf{b}_t - \mathbf{v}_t, \mathbf{x}_t - \mathbf{x}^*\rangle + \eta^2\|\nabla F(\mathbf{x}_t) - \mathbf{b}_t - \mathbf{v}_t\|$$

$$\leq D_t^2 - 2\eta\langle\nabla F(\mathbf{x}_t), \mathbf{x}_t - \mathbf{x}^*\rangle + 2\eta\langle\mathbf{b}_t, \mathbf{x}_t - \mathbf{x}^*\rangle + 2\eta\langle\mathbf{v}_t, \mathbf{x}_t - \mathbf{x}^*\rangle$$

$$+ 2\eta^2\|\nabla F(\mathbf{x}_t)\| + 4\eta^2\|\mathbf{v}_t\|^2 + 4\eta^2\|\mathbf{b}_t\|^2$$

By the coercivity property,

$$-2\eta \langle \nabla F(\mathbf{x}_t), \mathbf{x}_t - \mathbf{x}^* \rangle \le -2\eta[F(\mathbf{x}_t) - F(\mathbf{x}^*)] - \frac{\eta}{L}\|\nabla F(\mathbf{x}_t)\|^2$$

Substituting this into the recurrence for $D_{t+1}^2$, we obtain the following:

$$
\begin{aligned}
D_{t+1}^2 &\le D_t^2 - 2\eta[F(\mathbf{x}_t) - F(\mathbf{x}^*)] + 2\eta \langle \mathbf{v}_t, \mathbf{x}_t - \mathbf{x}^* \rangle + 2\eta \langle \mathbf{b}_t, \mathbf{x}_t - \mathbf{x}^* \rangle \\
&\quad + \eta(2\eta - 1/L)\|\nabla F(\mathbf{x}_t)\|^2 + 4\eta^2\|\mathbf{v}_t\|^2 + 4\eta^2\|\mathbf{b}_t\|^2 \\
&\le D_t^2 - 2\eta[F(\mathbf{x}_t) - F(\mathbf{x}^*)] + 2\eta \langle \mathbf{v}_t, \mathbf{x}_t - \mathbf{x}^* \rangle + 2\eta \langle \mathbf{b}_t, \mathbf{x}_t - \mathbf{x}^* \rangle + 4\eta^2\|\mathbf{v}_t\|^2 + 4\eta^2\|\mathbf{b}_t\|^2
\end{aligned}
$$

where the last inequality uses the fact that $\eta \le 1/2L$, Rearranging and taking averages on both sides

$$\sum_{t=1}^{T} F(\mathbf{x}_t) - F(\mathbf{x}^*) \le \frac{D_1^2}{2\eta T} + \frac{1}{T}\sum_{t=1}^{T}\langle \mathbf{b}_t, \mathbf{x}_t - \mathbf{x}^* \rangle + \frac{1}{T}\sum_{t=1}^{T}\langle \mathbf{v}_t, \mathbf{x}_t - \mathbf{x}^* \rangle + \frac{2\eta}{T}\sum_{t=1}^{T}\|\mathbf{b}_t\|^2 + \frac{2\eta}{T}\sum_{t=1}^{T}\|\mathbf{v}_t\|^2$$

Using the above inequality and the convexity of $F$, we conclude that

$$
\begin{aligned}
F(\hat{\mathbf{x}}_T) - F(\mathbf{x}^*) &= F\left(\tfrac{1}{T}\sum_{t=1}^{T}\mathbf{x}_t\right) - F(\mathbf{x}^*) \\
&\le \frac{1}{T}\sum_{t=1}^{T} F(\mathbf{x}_t) - F(\mathbf{x}^*) \\
&\le \frac{D_1^2}{2\eta T} + \frac{1}{T}\sum_{t=1}^{T}\langle \mathbf{b}_t, \mathbf{x}_t - \mathbf{x}^* \rangle + \frac{1}{T}\sum_{t=1}^{T}\langle \mathbf{v}_t, \mathbf{x}_t - \mathbf{x}^* \rangle + \frac{2\eta}{T}\sum_{t=1}^{T}\|\mathbf{b}_t\|^2 + \frac{2\eta}{T}\sum_{t=1}^{T}\|\mathbf{v}_t\|^2
\end{aligned}
$$

$\square$

### D.3 Proof of Lemma 11

Note that by definition of $E_t$

$$\|\nabla F(\mathbf{x}_t)\|\mathbb{1}\{E_t\} \le LD_t\mathbb{1}\{E_t\} \le 2LD_1$$

We recall that $\mathbf{b}_t = \mathbb{E}[\mathbf{g}_t|\mathcal{F}_{t-1}] - \mathbb{E}[\text{clip}_\Gamma(\mathbf{g}_t)|\mathcal{F}_{t-1}]$. Since $\text{Cov}[\mathbf{g}_t|\mathcal{F}_{t-1}] \preceq \Sigma$ by Assumption Bdd. $2^{\text{nd}}$ Moment, we obtain the following bound on $\|\mathbf{b}_t\|$ by an application of Lemma 4

$$\|\mathbf{b}_t\| \le \frac{\|\Sigma\|_2\sqrt{d_{\text{eff}}}}{\Gamma} + \frac{\|\nabla F(\mathbf{x}_t)\|\sqrt{\|\Sigma\|_2}}{\Gamma} + \frac{\|\nabla F(\mathbf{x}_t)\|^3}{\Gamma^2} + \frac{\|\Sigma\|_2 d_{\text{eff}}\|\nabla F(\mathbf{x}_t)\|}{\Gamma^2}$$

Since $\tilde{\mathbf{b}}_t = \mathbf{b}_t\mathbb{1}\{E_t\}$, it follows that

$$
\begin{aligned}
\|\mathbf{b}_t\| &\le \frac{\|\Sigma\|_2\sqrt{d_{\text{eff}}}}{\Gamma} + \frac{\|\nabla F(\mathbf{x}_t)\mathbb{1}\{E_t\}\|\sqrt{\|\Sigma\|_2}}{\Gamma} + \frac{\|\nabla F(\mathbf{x}_t)\|^3\mathbb{1}\{E_t\}}{\Gamma^2} + \frac{\|\Sigma\|_2 d_{\text{eff}}\|\nabla F(\mathbf{x}_t)\|\mathbb{1}\{E_t\}}{\Gamma^2} \\
&\le \frac{\|\Sigma\|_2\sqrt{d_{\text{eff}}}}{\Gamma} + \frac{2LD_1\sqrt{\|\Sigma\|_2}}{\Gamma} + \frac{8L^3D_1^3}{\Gamma^2} + \frac{2\|\Sigma\|_2 d_{\text{eff}}LD_1}{\Gamma^2}
\end{aligned}
$$

### D.4 Proof of Lemma 12

For any $s \in [T]$, we recall that $\mathbf{v}_s = \mathbb{E}\left[\text{clip}_\Gamma(\mathbf{g}_s)|\mathcal{F}_{s-1}\right] - \text{clip}_\Gamma(\mathbf{g}_s)$. Since $\mathbb{E}[\mathbf{g}_s|\mathcal{F}_{s-1}] = \nabla F(\mathbf{x}_s)$ and $\text{Cov}[\mathbf{g}_s|\mathcal{F}_{s-1}] \preceq \Sigma$, we obtain the following from Lemma 4

$$\|\mathbb{E}\left[\mathbf{v}_s\mathbf{v}_s^T|\mathcal{F}_{s-1}\right]\|_2 = \|\text{Cov}\left[\text{clip}_\Gamma(\mathbf{g}_s)|\mathcal{F}_{s-1}\right]\| \le \|\Sigma\|_2 + \frac{\|\nabla F(\mathbf{x}_s)\|^4}{\Gamma^2} + \frac{\|\nabla F(\mathbf{x}_s)\|^2\text{Tr}(\Sigma)}{\Gamma^2}$$

$$\text{Tr}\left(\mathbb{E}\left[\mathbf{v}_s\mathbf{v}_s^T|\mathcal{F}_{s-1}\right]\right) = \text{Tr}\left(\text{Cov}\left[\text{clip}_\Gamma(\mathbf{g}_s)|\mathcal{F}_{s-1}\right]\right) \le \text{Tr}(\Sigma)$$

For $s \in [1:T]$ define $\mathbb{E}[\tilde{\mathbf{v}}_s\tilde{\mathbf{v}}_s^T|\mathcal{F}_{s-1}] = \tilde{\Sigma}_s$. Since $\mathbb{1}\{E_s\}$ is $\mathcal{F}_{s-1}$-measurable and $\tilde{\mathbf{v}}_s = \mathbf{v}_s\mathbb{1}\{E_s\}$, it follows that $\tilde{\Sigma}_s = \mathbb{E}\left[\mathbf{v}_s\mathbf{v}_s^T|\mathcal{F}_{s-1}\right]\mathbb{1}\{E_s\}$. Hence, we conclude the following from the above

inequality

$$\|\tilde{\Sigma}_s\|_2 \leq \|\Sigma\|_2 + \frac{\|\nabla F(\mathbf{x}_s)\|^4 \mathbb{1}\{E_s\}}{\Gamma^2} + \frac{\|\nabla F(\mathbf{x}_s)\|^2 \mathsf{Tr}(\Sigma)\mathbb{1}\{E_s\}}{\Gamma^2}$$

$$\leq \|\Sigma\|_2 + \frac{16L^4 D_1^4}{\Gamma^2} + \frac{4L^2 D_1^2 \mathsf{Tr}(\Sigma)}{\Gamma^2} = V$$

$$\mathsf{Tr}(\tilde{\Sigma}_s) \leq \mathsf{Tr}(\Sigma) \tag{28}$$

For $s \in [T]$, define $h_s = \langle \tilde{\mathbf{v}}_s, \mathbf{d}_s \rangle$. We note that

$$|h_s| \leq \|\tilde{\mathbf{v}}_s\| \cdot \|\mathbf{d}_s\| \leq 4\Gamma D_1$$
$$\mathbb{E}[h_s|\mathcal{F}_{s-1}] = \langle \mathbb{E}[\tilde{\mathbf{v}}_s|\mathcal{F}_{s-1}], \mathbf{d}_s \rangle = 0$$
$$\mathbb{E}[h_s^2|\mathcal{F}_{s-1}] = \mathbf{d}_s^T \mathbb{E}[\tilde{\mathbf{v}}_s \tilde{\mathbf{v}}_s^T]\mathbf{d}^s$$
$$= \mathbf{d}_s^T \tilde{\Sigma}_s \mathbf{d}_s$$
$$\leq \|\mathbf{d}_s\|^2 \|\tilde{\Sigma}\| \leq 4D_1^2 V$$

Hence, by Freedman's inequality (Lemma 3), we conclude that the following holds with probability at least $1 - \delta/2$:

$$\sum_{s=1}^{t} \langle \tilde{\mathbf{v}}_t, \mathbf{d}_t \rangle \leq 4D_1\sqrt{Vt\ln(K/\delta)} + 8\Gamma D_1 \ln(K/\delta) \quad \forall\, t \in [T]$$

We now apply Corollary 6 with $p_s = V$, $q_s = \mathsf{Tr}(\Sigma)$ and $\tau = 2\Gamma$ to conclude that the following holds with probability at least $1 - \delta/2$ uniformly for every $t \in [T]$

$$\sum_{s=1}^{t} \|\tilde{\mathbf{v}}_s\|^2 \leq 4C_M\Gamma^2\ln(K/\delta)^2 + C_M\mathsf{UP}(t)\mathsf{Tr}(\Sigma) + \frac{C_M t\mathsf{UP}(t)V^2}{4\Gamma^2}$$

$$\leq 4C_M\Gamma^2\ln(K/\delta)^2 + C_M T\mathsf{Tr}(\Sigma) + \frac{C_M T^2 V^2}{4\Gamma^2}$$

$$\leq C_M T\left(\|\Sigma\|_2 d_{\mathsf{eff}} + \frac{4\Gamma^2\ln(K/\delta)^2}{T} + \frac{V^2 T}{4\Gamma^2}\right) = C_M g^2 T$$

where

$$g^2 = \|\Sigma\|_2 d_{\mathsf{eff}} + \frac{4\Gamma^2\ln(K/\delta)^2}{T} + \frac{V^2 T}{4\Gamma^2}$$

The proof is concluded by a union bound

### D.5 Proof of Lemma 13

We prove the claim via induction. Clearly, the claim is true for $t = 1$. Now, suppose the claim holds for every $s \leq t$ for some $t \in [T]$. Since $\Pi_C$ is a contractive operator

$$D_{t+1}^2 = \|\mathbf{x}_{t+1} - \mathbf{x}^*\|^2 \leq D_t^2 - 2\eta\langle\nabla F(\mathbf{x}_t) - \mathbf{b}_t - \mathbf{v}_t, \mathbf{x}_t - \mathbf{x}^*\rangle + \eta^2\|\nabla F(\mathbf{x}_t) - \mathbf{b}_t - \mathbf{v}_t\|$$

$$\leq D_t^2 - 2\eta\langle\nabla F(\mathbf{x}_t), \mathbf{x}_t - \mathbf{x}^*\rangle + 2\eta\langle\mathbf{b}_t, \mathbf{x}_t - \mathbf{x}^*\rangle + 2\eta\langle\mathbf{v}_t, \mathbf{x}_t - \mathbf{x}^*\rangle$$

$$+ 2\eta^2\|\nabla F(\mathbf{x}_t)\| + 4\eta^2\|\mathbf{v}_t\|^2 + 4\eta^2\|\mathbf{b}_t\|^2$$

By the coercivity property,

$$-2\eta\langle\nabla F(\mathbf{x}_t), \mathbf{x}_t - \mathbf{x}^*\rangle \leq -2\eta[F(\mathbf{x}_t) - F(\mathbf{x}^*)] - \frac{\eta}{L}\|\nabla F(\mathbf{x}_t)\|^2$$

Substituting this into the recurrence for $D_{t+1}^2$, we obtain the following:

$$D_{t+1}^2 \leq D_t^2 - 2\eta[F(\mathbf{x}_t) - F(\mathbf{x}^*)] + 2\eta\langle\mathbf{v}_t, \mathbf{x}_t - \mathbf{x}^*\rangle + 2\eta\langle\mathbf{b}_t, \mathbf{x}_t - \mathbf{x}^*\rangle$$

$$+ \eta(2\eta - 1/L)\|\nabla F(\mathbf{x}_t)\|^2 + 4\eta^2\|\mathbf{v}_t\|^2 + 4\eta^2\|\mathbf{b}_t\|^2$$

$$\leq D_t^2 + 2\eta\langle\mathbf{v}_t, \mathbf{x}_t - \mathbf{x}^*\rangle + 2\eta\langle\mathbf{b}_t, \mathbf{x}_t - \mathbf{x}^*\rangle + 4\eta^2\|\mathbf{v}_t\|^2 + 4\eta^2\|\mathbf{b}_t\|^2$$

where we use the fact that $\eta \leq {}^1/_{2L}$. Now, by the Cauchy Schwarz inequality and the fact that $ab \leq a^2 + {}^{b^2}/_4$ we obtain the following:

$$2\eta \langle \mathbf{b}_t, \mathbf{x}_t - \mathbf{x}^* \rangle \leq \frac{D_t^2}{2T} + \eta^2 T \|\mathbf{b}_t\|^2$$

It follows that

$$D_{t+1}^2 \leq \left(1 + \frac{1}{2T}\right) D_t^2 + 5\eta^2 T \|\mathbf{b}_t\|^2 + 4\eta^2 \|\mathbf{v}_t\|^2 - 2\eta \langle \mathbf{v}_t, \mathbf{x}_t - \mathbf{x}^* \rangle$$

Unrolling the above recursion for $t$ steps and using the fact that $(1 + {}^1/_{2T})^T \leq 2$, we obtain the following:

$$D_{t+1}^2 \leq \left(1 + \frac{1}{2T}\right)^T D_1^2 + \sum_{s=1}^{t} \left(1 + \frac{1}{2T}\right)^{t-s} \left(5\eta^2 T \|\mathbf{b}_s\|^2 + 4\eta^2 \|\mathbf{v}_s\|^2 + 2\eta \langle \mathbf{v}_s, \mathbf{x}_s - \mathbf{x}^* \rangle\right)$$

$$\leq 2D_1^2 + \sum_{s=1}^{t} 10\eta^2 T \|\mathbf{b}_s\|^2 + 8\eta^2 \|\mathbf{v}_s\|^2 - 4\eta \langle \mathbf{v}_s, \mathbf{x}_s - \mathbf{x}^* \rangle$$

By the induction hypothesis, $\mathbb{1}\{E_s\} = 1 \; \forall s \in [t]$. Hence,

$$D_{t+1}^2 \leq 2D_1^2 + 10\eta^2 T \sum_{s=1}^{t} \|\tilde{\mathbf{b}}_s\|^2 + 8\eta^2 \sum_{s=1}^{t} \|\tilde{\mathbf{v}}_s\|^2 - 4\eta \sum_{s=1}^{t} \langle \tilde{\mathbf{v}}_s, \mathbf{d}_s \rangle$$

$$\leq 2D_1^2 + 10\eta^2 T^2 B^2 + 8C_M \eta^2 g^2 T + 16\eta D_1 \left[\sqrt{Vt \ln({}^K/_\delta)} + 2\Gamma \ln({}^K/_\delta)\right]$$

$$\leq 3D_1^2 + 10\eta^2 T^2 B^2 + 8C_M \eta^2 g^2 T + 64\eta^2 \left(\sqrt{Vt \ln({}^K/_\delta)} + 2\Gamma \ln({}^K/_\delta)\right)^2$$

$$\leq 3D_1^2 + 10\eta^2 T^2 B^2 + 8C_M \eta^2 g^2 T + 128\eta^2 VT \ln({}^K/_\delta) + 1024\Gamma^2 \ln({}^K/_\delta)^2$$

where the second inequality follows from the Lemma 11 and the fact that we have conditioned on $E$. Note that by definition of $g^2$ and the AM-GM inequality

$$g^2 T \geq 4\Gamma^2 \ln({}^K/_\delta)^2 + \frac{V^2 T^2}{4\Gamma^2} \geq \max\{4\Gamma^2 \ln({}^K/_\delta)^2, 2VT \ln({}^K/_\delta)\}$$

It follows that

$$D_{t+1}^2 \leq 3D_1^2 + 10\eta^2 T^2 B^2 + 8(C_M + 40)\eta^2 g^2 T$$
$$\leq 3D_1^2 + 10c^2 D_1^2 + c^2(8C_M + 320)D_1^2$$
$$\leq 4D_1^2$$

where the second inequality uses the definition of $\eta$ and the fact that $B'$ and $g'$ upper bound $B$ and $G$ respectively by equations (25) and (27) and the last inequality sets $c = \frac{1}{\sqrt{8C_M + 330}}$. Hence, $D_{t+1} \leq 2D_1$ which proves the claim by induction.

## E  Analysis for Lipschitz Convex Functions

Let $d_{\text{eff}} = \frac{\text{Tr}(\Sigma)}{\|\Sigma\|_2}$. Since $\Sigma$ is positive semidefinite, $1 \leq d_{\text{eff}} \leq d$. Moreover, let $\text{clip}_\Gamma(\mathbf{g}_t) = \partial F(\mathbf{x}_t) + \mathbf{b}_t + \mathbf{v}_t$ where $\mathbf{b}_t = \mathbb{E}[\text{clip}_\Gamma(\mathbf{g}_t)|\mathcal{F}_t] - \partial F(\mathbf{x}_t)$ represents the bias due to clipping and $\mathbb{E}[\mathbf{v}_t|\mathcal{F}_t] = 0$. Let $D_t = \|\mathbf{x}_t - \mathbf{x}^*\|$ where $\mathbf{x}^*$ is the minimizer of $F$ considered in the statement of Theorem 3. Using the smoothness and convexity properties of $F$, we first prove the following intermediate average iterate guarantee:

**Lemma 14** (Intermediate Average Iterate Guarantee). *The following holds for any $\eta > 0$*

$$F(\hat{\mathbf{x}}_T) - F(\mathbf{x}^*) \leq \frac{D_1^2}{2\eta T} - \frac{1}{T} \sum_{t=1}^{T} \langle \mathbf{b}_t, \mathbf{x}_t - \mathbf{x}^* \rangle - \frac{1}{T} \sum_{t=1}^{T} \langle \mathbf{v}_t, \mathbf{x}_t - \mathbf{x}^* \rangle$$

$$+ \eta G^2 + \frac{2\eta}{T} \sum_{t=1}^{T} \|\mathbf{b}_t\|^2 + \frac{2\eta}{T} \sum_{t=1}^{T} \|\mathbf{v}_t\|^2$$

Define the events $E_t$ and the random vectors $\mathbf{d}_t$ as follows for $t \in [T]$:
$$E_t = \{D_t \leq 2D_1\}$$
$$\mathbf{d}_t = (\mathbf{x}_t - \mathbf{x}^*)\mathbb{1}\{E_t\}$$

We use the following lemma to control the bias

**Lemma 15** (Bias Control). *For every $t \in [T]$, $\|\mathbf{b}_t\| \leq B$ where $B$ is defined as follows:*
$$B = \frac{\|\Sigma\|_2\sqrt{d_{\text{eff}}}}{\Gamma} + \frac{G\sqrt{\|\Sigma\|_2}}{\Gamma} + \frac{G^3}{\Gamma^2} + \frac{\|\Sigma\|_2 d_{\text{eff}} G}{\Gamma^2}$$

We use the following lemma to control the varince

**Lemma 16** (Variance Control). *Let $V \geq 0$ be defined as follows:*
$$V = \|\Sigma\|_2 + \frac{G^4}{\Gamma^2} + \frac{G^2\|\Sigma\|_2 d_{\text{eff}}}{\Gamma^2}$$
*Then the following holds with probability at least $1 - \delta$ uniformly for every $t \in [T]$*

$$\sum_{s=1}^{t}\langle \mathbf{v}_s, \mathbf{d}_s\rangle \leq 4D_1\sqrt{Vt\ln(K/\delta)} + 8\Gamma D_1\ln(K/\delta)$$

$$\sum_{s=1}^{t}\|\mathbf{v}_s\|^2 \leq C_M g^2 T$$

*where $C_M$ is a numerical constant and $g^2$ is defined as follows*
$$g^2 = \|\Sigma\|_2 d_{\text{eff}} + \frac{4\Gamma^2\ln(K/\delta)^2}{T} + \frac{V^2 T}{4\Gamma^2}$$

Let $E$ denote the following event

$$E = \{\sum_{s=1}^{t}\langle \mathbf{v}_s, \mathbf{d}_s\rangle \leq 4D_1\sqrt{Vt\ln(K/\delta)} + 8\Gamma D_1\ln(K/\delta) \quad \forall\, t \in [T]$$

$$\sum_{s=1}^{t}\|\mathbf{v}_s\|^2 \leq C_M g^2 T \quad \forall t \in [T]\}$$

Note that by Lemma 16, $\mathbb{P}(E) \geq 1 - \delta$. We define the constant $A$ as follows:
$$A = \|\Sigma\|_2\sqrt{d_{\text{eff}}} + G\sqrt{\|\Sigma\|_2} = \sqrt{\|\Sigma\|_2}\left(\sqrt{\text{Tr}(\Sigma)} + G\right)$$

We now set the clipping level $\Gamma = \sqrt{\frac{AT}{\ln(K/\delta)}}$. For this choice of $\Gamma$, we now simplify the expression for $B$ as follows:
$$B = \sqrt{\frac{A\ln(K/\delta)}{T}} + \frac{G\left(\|\Sigma\|_2 d_{\text{eff}} + G^2\right)\ln(K/\delta)}{AT} \tag{29}$$

Similarly, the expression for $V$ can be simplified as follows
$$V = \|\Sigma\|_2 + \frac{G^2\ln(K/\delta)}{AT}\left(\|\Sigma\|_2 d_{\text{eff}} + G^2\right) \tag{30}$$

Using the above inequality, we derive the following upper bound for $g$:
$$g \leq \sqrt{\|\Sigma\|_2 d_{\text{eff}}} + \frac{2\Gamma\ln(K/\delta)}{\sqrt{T}} + \frac{V\sqrt{T}}{2\Gamma}$$
$$= \sqrt{\|\Sigma\|_2 d_{\text{eff}}} + 2\sqrt{A\ln(K/\delta)} + \frac{V\sqrt{\ln(K/\delta)}}{2\sqrt{A}}$$
$$= \sqrt{\|\Sigma\|_2 d_{\text{eff}}} + 2\sqrt{A\ln(K/\delta)} + \frac{\|\Sigma\|_2\sqrt{\ln(K/\delta)}}{2\sqrt{A}} + \frac{G^2\ln(K/\delta)^{3/2}}{A^{3/2}T}\left(\|\Sigma\|_2 d_{\text{eff}} + G^2\right)$$
$$\leq \sqrt{\|\Sigma\|_2 d_{\text{eff}}} + 3\sqrt{A\ln(K/\delta)} + \frac{G^2\ln(K/\delta)^{3/2}}{A^{3/2}T}\left(\|\Sigma\|_2 d_{\text{eff}} + G^2\right) = g' \tag{31}$$

We also prove the following uniform upper bound on the iterates $\mathbf{x}_t$

**Lemma 17** (Iterate Bound). *Let $\eta \leq c \min\{D_1/BT, D_1/g'\sqrt{T}, D_1/G\sqrt{T}\}$ where $c = \frac{1}{\sqrt{8C_M + 334}}$. Then, conditioned on the event $E$, $D_t \leq 2D_1 \; \forall t \in [T]$.*

Equipped with the above lemmas, we now prove the following theorem which is a formal restatement of Theorem 4

**Theorem 8** (Lipschitz Convex Objectives). *Let Assumptions Convexity, G-Lipschitzness and Bdd. $2^{nd}$ Moment be satisfied. Then, for any $\delta \in (0, 1/2)$ and $T \geq \ln(\ln(d))$, there exists an $\eta \in (0, G/\sqrt{T}]$ such that the average iterate of Algorithm 1 run for $T$ iterations with step-size $\eta_t = \eta$ and clipping level $\Gamma = \sqrt{\frac{T\sqrt{\|\Sigma\|_2}(\sqrt{\mathsf{Tr}(\Sigma)}+G)}{\ln(\ln(T)/\delta)}}$ satisfies the following with probability at least $1 - \delta$*

$$
F(\hat{\mathbf{x}}_T) - F(\mathbf{x}^*) \lesssim \frac{D_1 G}{\sqrt{T}} + D_1 \sqrt{\frac{\mathsf{Tr}(\Sigma) + \sqrt{\|\Sigma\|_2}\left(\sqrt{\mathsf{Tr}(\Sigma)} + G\right)\ln(K/\delta)}{T}}
$$
$$
+ \frac{D_1 G \ln(K/\delta)}{T}\sqrt{\frac{\mathsf{Tr}(\Sigma) + G^2}{\|\Sigma\|_2}} + \frac{D_1 G^2 \ln(1/\delta)^{3/2}}{T^{3/2}}\left(\frac{\mathsf{Tr}(\Sigma) + G^2}{\|\Sigma\|^3}\right)^{1/4}
$$

## E.1  Proof of Lemma 14

*Proof.* Since $\Pi_{\mathcal{C}}$ is a contractive operator

$$
\begin{aligned}
D_{t+1}^2 &= \|\mathbf{x}_{t+1} - \mathbf{x}^*\|^2 \leq D_t^2 - 2\eta\langle\partial F(\mathbf{x}_t) + \mathbf{b}_t + \mathbf{v}_t, \mathbf{x}_t - \mathbf{x}^*\rangle + \eta^2\|\nabla F(\mathbf{x}_t) + \mathbf{b}_t + \mathbf{v}_t\| \\
&\leq D_t^2 - 2\eta\langle\partial F(\mathbf{x}_t), \mathbf{x}_t - \mathbf{x}^*\rangle - 2\eta\langle\mathbf{b}_t, \mathbf{x}_t - \mathbf{x}^*\rangle - 2\eta\langle\mathbf{v}_t, \mathbf{x}_t - \mathbf{x}^*\rangle \\
&\quad + 2\eta^2\|\partial F(\mathbf{x}_t)\|^2 + 4\eta^2\|\mathbf{v}_t\|^2 + 4\eta^2\|\mathbf{b}_t\|^2 \\
&\leq D_t^2 - 2\eta[F(\mathbf{x}_t) - F(\mathbf{x}^*)] - 2\eta\langle\mathbf{b}_t, \mathbf{x}_t - \mathbf{x}^*\rangle - 2\eta\langle\mathbf{v}_t, \mathbf{x}_t - \mathbf{x}^*\rangle + 2\eta^2 G^2 + 4\eta^2\|\mathbf{b}_t\|^2 + 4\eta^2\|\mathbf{v}_t\|^2
\end{aligned}
$$

where the second inequality follows from the definition of the subgradient and the $G$ lipschitzness of $F$. Rearranging and taking averages on both sides

$$
\begin{aligned}
\sum_{t=1}^{T} F(\mathbf{x}_t) - F(\mathbf{x}^*) &\leq \frac{D_1^2}{2\eta T} - \frac{1}{T}\sum_{t=1}^{T}\langle\mathbf{b}_t, \mathbf{x}_t - \mathbf{x}^*\rangle - \frac{1}{T}\sum_{t=1}^{T}\langle\mathbf{v}_t, \mathbf{x}_t - \mathbf{x}^*\rangle \\
&\quad + \eta G^2 + \frac{2\eta}{T}\sum_{t=1}^{T}\|\mathbf{b}_t\|^2 + \frac{2\eta}{T}\sum_{t=1}^{T}\|\mathbf{v}_t\|^2
\end{aligned}
$$

Using the above inequality and the convexity of $F$, we conclude that

$$
\begin{aligned}
F(\hat{\mathbf{x}}_T) - F(\mathbf{x}^*) &= F\left(\tfrac{1}{T}\sum_{t=1}^{T}\mathbf{x}_t\right) - F(\mathbf{x}^*) \\
&\leq \frac{1}{T}\sum_{t=1}^{T} F(\mathbf{x}_t) - F(\mathbf{x}^*) \\
&\leq \frac{D_1^2}{2\eta T} - \frac{1}{T}\sum_{t=1}^{T}\langle\mathbf{b}_t, \mathbf{x}_t - \mathbf{x}^*\rangle - \frac{1}{T}\sum_{t=1}^{T}\langle\mathbf{v}_t, \mathbf{x}_t - \mathbf{x}^*\rangle \\
&\quad + \eta G^2 + \frac{2\eta}{T}\sum_{t=1}^{T}\|\mathbf{b}_t\|^2 + \frac{2\eta}{T}\sum_{t=1}^{T}\|\mathbf{v}_t\|^2
\end{aligned}
$$

$\square$

## E.2 Proof of Lemma 15

We recall that $\mathbf{b}_t = \mathbb{E}[\mathbf{g}_t|\mathcal{F}_{t-1}] - \mathbb{E}[\mathsf{clip}_\Gamma(\mathbf{g}_t)|\mathcal{F}_{t-1}]-$. Since $\mathsf{Cov}[\mathbf{g}_t|\mathcal{F}_{t-1}] \preceq \Sigma$ by Assumption Bdd. 2nd Moment, we obtain the following bound on $\|\mathbf{b}_t\|$ by an application of Lemma 4

$$\|\mathbf{b}_t\| \leq \frac{\|\Sigma\|_2\sqrt{d_{\mathsf{eff}}}}{\Gamma} + \frac{\|\partial F(\mathbf{x}_t)\|\sqrt{\|\Sigma\|_2}}{\Gamma} + \frac{\|\partial F(\mathbf{x}_t)\|^3}{\Gamma^2} + \frac{\|\Sigma\|_2 d_{\mathsf{eff}}\|\partial F(\mathbf{x}_t)\|}{\Gamma^2}$$

$$\leq \frac{\|\Sigma\|_2\sqrt{d_{\mathsf{eff}}}}{\Gamma} + \frac{G\sqrt{\|\Sigma\|_2}}{\Gamma} + \frac{G^3}{\Gamma^2} + \frac{\|\Sigma\|_2 d_{\mathsf{eff}} G}{\Gamma^2}$$

## E.3 Proof of Lemma 16

For any $s \in [T]$, we recall that $\mathbf{v}_s = \mathbb{E}\left[\mathsf{clip}_\Gamma(\mathbf{g}_s)|\mathcal{F}_{s-1}\right] - \mathsf{clip}_\Gamma(\mathbf{g}_s)$. Since $\mathbb{E}[\mathbf{g}_s|\mathcal{F}_{s-1}] = \partial F(\mathbf{x}_s)$ and $\mathsf{Cov}[\mathbf{g}_s|\mathcal{F}_{s-1}] \preceq \Sigma$, we obtain the following from Lemma 4

$$\|\mathbb{E}\left[\mathbf{v}_s\mathbf{v}_s^T|\mathcal{F}_{s-1}\right]\|_2 = \|\mathsf{Cov}\left[\mathsf{clip}_\Gamma(\mathbf{g}_s)|\mathcal{F}_{s-1}\right]\| \leq \|\Sigma\|_2 + \frac{\|\partial F(\mathbf{x}_s)\|^4}{\Gamma^2} + \frac{\|\partial F(\mathbf{x}_s)\|^2\mathsf{Tr}(\Sigma)}{\Gamma^2}$$

$$\leq \|\Sigma\|_2 + \frac{G^4}{\Gamma^2} + \frac{G^2\mathsf{Tr}(\Sigma)}{\Gamma^2}$$

$$\mathsf{Tr}\left(\mathbb{E}\left[\mathbf{v}_s\mathbf{v}_s^T|\mathcal{F}_{s-1}\right]\right) = \mathsf{Tr}\left(\mathsf{Cov}\left[\mathsf{clip}_\Gamma(\mathbf{g}_s)|\mathcal{F}_s\right]\right) \leq \mathsf{Tr}(\Sigma)$$

For $s \in [T]$, define $h_s = \langle \mathbf{v}_s, \mathbf{d}_s \rangle$. We note that

$$|h_s| \leq \|\mathbf{v}_s\| \cdot \|\mathbf{d}_s\| \leq 4\Gamma D_1$$

$$\mathbb{E}\left[h_s|\mathcal{F}_{s-1}\right] = \langle \mathbb{E}[\mathbf{v}_s|\mathcal{F}_{s-1}], \mathbf{d}_s \rangle = 0$$

$$\mathbb{E}\left[h_s^2|\mathcal{F}_{s-1}\right] = \mathbf{d}_s^T\mathbb{E}[\mathbf{v}_s\mathbf{v}_s^T]\mathbf{d}_s$$

$$= \mathbf{d}_s^T\Sigma_s\mathbf{d}_s$$

$$\leq \|\mathbf{d}_s\|^2\|\Sigma_s\| \leq 4D_1^2V$$

Hence, by Freedman's inequality (Lemma 3), we conclude that the following holds with probability at least $1 - \delta/2$:

$$\sum_{s=1}^t \langle \tilde{\mathbf{v}}_t, \mathbf{d}_t \rangle \leq 4D_1\sqrt{Vt\ln(K/\delta)} + 8\Gamma D_1\ln(K/\delta) \quad \forall\, t \in [T]$$

We now apply Corollary 6 with $p_s = V$, $q_s = \mathsf{Tr}(\Sigma)$ and $\tau = 2\Gamma$ to conclude that the following holds with probability at least $1 - \delta/2$ uniformly for every $t \in [T]$

$$\sum_{s=1}^t \|\tilde{\mathbf{v}}_s\|^2 \leq 4C_M\Gamma^2\ln(K/\delta)^2 + C_M\mathsf{UP}(t)\mathsf{Tr}(\Sigma) + \frac{C_M t\mathsf{UP}(t)V^2}{4\Gamma^2}$$

$$\leq 4C_M\Gamma^2\ln(K/\delta)^2 + C_M T\mathsf{Tr}(\Sigma) + \frac{C_M T^2V^2}{4\Gamma^2}$$

$$\leq C_M T\left(\|\Sigma\|_2 d_{\mathsf{eff}} + \frac{4\Gamma^2\ln(K/\delta)^2}{T} + \frac{V^2T}{4\Gamma^2}\right) = C_M g^2 T$$

where

$$g^2 = \|\Sigma\|_2 d_{\mathsf{eff}} + \frac{4\Gamma^2\ln(K/\delta)^2}{T} + \frac{V^2T}{4\Gamma^2}$$

The proof is concluded by a union bound

## E.4 Proof of Lemma 17

We prove the claim via induction. Clearly, the claim is true for $t = 1$. Now, suppose the claim holds for every $s \leq t$ for some $t \in [T]$. Since $\Pi_\mathcal{C}$ is a contractive operator

$$D_{t+1}^2 = \|\mathbf{x}_{t+1} - \mathbf{x}^*\|^2 \leq D_t^2 - 2\eta\langle\partial F(\mathbf{x}_t) + \mathbf{b}_t + \mathbf{v}_t, \mathbf{x}_t - \mathbf{x}^*\rangle + \eta^2\|\nabla F(\mathbf{x}_t) + \mathbf{b}_t + \mathbf{v}_t\|$$

$$\leq D_t^2 - 2\eta\langle\partial F(\mathbf{x}_t), \mathbf{x}_t - \mathbf{x}^*\rangle - 2\eta\langle\mathbf{b}_t, \mathbf{x}_t - \mathbf{x}^*\rangle - 2\eta\langle\mathbf{v}_t, \mathbf{x}_t - \mathbf{x}^*\rangle$$

$$+ 2\eta^2\|\partial F(\mathbf{x}_t)\|^2 + 4\eta^2\|\mathbf{v}_t\|^2 + 4\eta^2\|\mathbf{b}_t\|^2$$

$$\leq D_t^2 - 2\eta[F(\mathbf{x}_t) - F(\mathbf{x}^*)] - 2\eta\langle\mathbf{b}_t, \mathbf{x}_t - \mathbf{x}^*\rangle - 2\eta\langle\mathbf{v}_t, \mathbf{x}_t - \mathbf{x}^*\rangle + 2\eta^2 G^2 + 4\eta^2\|\mathbf{b}_t\|^2 + 4\eta^2\|\mathbf{v}_t\|^2$$

where the second inequality follows from the definition of the subgradient and the $G$ lipschitzness of $F$. Now, by the Cauchy Schwarz inequality and the fact that $ab \le a^2 + b^2/4$ we obtain the following:

$$-2\eta \langle \mathbf{b}_t, \mathbf{x}_t - \mathbf{x}^* \rangle \le \frac{D_t^2}{2T} + \eta^2 T \|\mathbf{b}_t\|^2$$

It follows that

$$D_{t+1}^2 \le \left(1 + \frac{1}{2T}\right) D_t^2 + 5\eta^2 T \|\mathbf{b}_t\|^2 + 2\eta^2 G^2 + 4\eta^2 \|\mathbf{v}_t\|^2 - 2\eta \langle \mathbf{v}_t, \mathbf{x}_t - \mathbf{x}^* \rangle$$

Unrolling the above recursion for $t$ steps and using the fact that $(1 + 1/2T)^T \le 2$, we obtain the following:

$$D_{t+1}^2 \le \left(1 + \frac{1}{2T}\right)^T D_1^2 + \sum_{s=1}^{t} \left(1 + \frac{1}{2T}\right)^{t-s} \left(5\eta^2 T \|\mathbf{b}_s\|^2 + 2\eta^2 G^2 + 4\eta^2 \|\mathbf{v}_s\|^2 - 2\eta \langle \mathbf{v}_s, \mathbf{x}_s - \mathbf{x}^* \rangle\right)$$

$$\le 2D_1^2 + 4\eta^2 G^2 T + \sum_{s=1}^{t} 10\eta^2 T \|\mathbf{b}_s\|^2 + 8\eta^2 \|\mathbf{v}_s\|^2 - 4\eta \langle \mathbf{v}_s, \mathbf{x}_s - \mathbf{x}^* \rangle$$

By the induction hypothesis, $\mathbb{1}\{E_s\} = 1 \; \forall s \in [t]$. Hence,

$$D_{t+1}^2 \le 2D_1^2 + 4\eta^2 G^2 T + 10\eta^2 T \sum_{s=1}^{t} \|\mathbf{b}_s\|^2 + 8\eta^2 \sum_{s=1}^{t} \|\mathbf{v}_s\|^2 - 4\eta \sum_{s=1}^{t} \langle \mathbf{v}_s, \mathbf{d}_s \rangle$$

$$\le 2D_1^2 + 4\eta^2 G^2 T + +10\eta^2 T^2 B^2 + 8 C_M \eta^2 g^2 T + 16\eta D_1 \left[ \sqrt{Vt \ln(K/\delta)} + 2\Gamma \ln(K/\delta) \right]$$

$$\le 3D_1^2 + 4\eta^2 G^2 T + +10\eta^2 T^2 B^2 + 8 C_M \eta^2 g^2 T + 64\eta^2 \left( \sqrt{Vt \ln(K/\delta)} + 2\Gamma \ln(K/\delta) \right)^2$$

$$\le 3D_1^2 + 4\eta^2 G^2 T + +10\eta^2 T^2 B^2 + 8 C_M \eta^2 g^2 T + 128\eta^2 VT \ln(K/\delta) + 1024\Gamma^2 \ln(K/\delta)^2$$

where the second inequality follows from the Lemma 15 and the fact that we have conditioned on $E$. Note that by definition of $g^2$ and the AM-GM inequality

$$g^2 T \ge 4\Gamma^2 \ln(K/\delta)^2 + \frac{V^2 T^2}{4\Gamma^2} \ge \max\{4\Gamma^2 \ln(K/\delta)^2, 2VT \ln(K/\delta)\}$$

It follows that

$$D_{t+1}^2 \le 3D_1^2 + 4\eta^2 G^2 T + 10\eta^2 T^2 B^2 + 8(C_M + 40)\eta^2 g^2 T$$
$$\le 3D_1^2 + 4c^2 D_1^2 + 10c^2 D_1^2 + c^2(8 C_M + 320) D_1^2$$
$$\le 4D_1^2$$

where the second inequality uses the definition of $\eta$ and the fact that $g'$ upper bounds $g$, and the last inequality sets $c = \frac{1}{\sqrt{8C_M + 334}}$. Hence, $D_{t+1} \le 2D_1$ which proves the claim by induction.

# F   Improved Martingale Concentration via PAC Bayes Theory

We have the following re-statement of Theorem 5 for the sake of readability.

**Theorem 9.** *Suppose $M_t$ for $t = 0, \ldots, T$ is an $\mathbb{R}^d$ valued martingale such that $M_0 = 0$ almost surely, the martingale difference sequence $\mathbf{v}_t := M_t - M_{t-1}$ is such that $\|\mathbf{v}_t\| \le \Gamma$ and $\mathbb{E}[\mathbf{v}_t \mathbf{v}_t^\intercal | \mathcal{F}_{t-1}] = \Sigma_t$ almost surely for every $t = 1, \ldots, T$ for some $\Gamma > 0$. Assume that there are deterministic sequences $p_1, \ldots, p_T$ and $q_1, \ldots, q_T$ such that $\mathsf{Tr}(\Sigma_t) \le q_t$ and $\|\Sigma_t\| \le p_t$ almost surely.*

*Let $\bar{q} := \frac{1}{T} \sum_{t=1}^{T} q_t$ and $\bar{p} := \frac{1}{T} \sum_{t=1}^{T} p_t$. Then, for any $\delta \in (0, \frac{1}{2})$*

$$\mathbb{P}(\sup_{t \le T} \|M_t\| \ge g(T, \delta)\sqrt{T}) \le \delta$$

*Where $g(T, \delta) = C \left[ \sqrt{\bar{q}} + \frac{\bar{p}\sqrt{T}}{\Gamma} + \frac{\Gamma}{\sqrt{T}} \log(\frac{K}{\delta}) \right]$ and $K = \log \Theta(\log((\frac{\sqrt{\bar{q}T}}{\Gamma} + 1)\log(d+1)))$*

Define the event $\mathcal{A}_t(g) := \{\|M_t\| \leq g\sqrt{T}\}$ and $\mathcal{B}_t(g) := \cap_{s=1}^t \mathcal{A}_s$. Consider the quantity $N_t := \|M_t\|^2 - \sum_{s=1}^t \|\mathbf{v}_s\|^2$.

**Theorem 10.** *Let $\delta \in (0, \frac{1}{2})$ and $g = g(T, \frac{\delta}{2})$ be as defined in Theorem 9. Under the conditions of Theorem 9, the following inequality holds for some large enough universal constant $C$.*

$$\mathbb{P}\left(\{\sup_{t \leq T} |N_t| > \Gamma C g\sqrt{T}\log(\tfrac{1}{\delta}) + \tfrac{C\nu g T^{3/2}}{\Gamma}\} \cap \mathcal{B}_T(g)\right) \leq \delta$$

The next corollary is a simple consequence of the Theorems 9 and 10.

**Corollary 6.** *Let $\delta \in (0, \frac{1}{2})$ and $g = g(T, \frac{\delta}{3})$ be as specified in Theorem 9. Under the conditions of Theorem 9, the following inequality holds with probability at-least $1 - \delta$:*

$$\sum_{t=1}^T \|\mathbf{v}_t\|^2 \leq Cg^2 T$$

## F.1 Proof of Theorem 9

The aim of this section is to prove the sharp concentration result given in Theorem 9. We now consider the concentration of norms of the martingale $\|M_t\|$. Define the event $\mathcal{A}_t := \{\|M_t\| \leq g\sqrt{T}\}$ and $\mathcal{B}_t = \cap_{s=1}^t \mathcal{A}_s$. Let $H$ be any stopping time for the martingale $M_t$. We have the following inequality which follows from PAC-Bayes theory (see Equation 5.2.1, Page 159 in [7]).

**Theorem 11.** *Suppose $\pi$ be any measure over $\mathbb{R}^d$ and let $\mathcal{M}_1(\mathbb{R}^d)$ denote the space of all probability measures over $\mathbb{R}^d$. Let $\gamma > 0$ be arbitrary. Then conditioned on $\mathcal{B}_T$, with probability at-least $1 - \delta$, the following inequality holds:*

$$\sup_{\rho \in \mathcal{M}_1(\mathbb{R}^d)} \mathbb{E}_{\theta \sim \rho} \gamma \left\langle M_{\min(H,T)}, \theta \right\rangle - \mathsf{KL}\left(\rho \| \pi\right) \leq \log\left(\mathbb{E}_M \mathbb{E}_{\theta \sim \pi} \frac{\exp(\gamma \left\langle M_{\min(H,T)}, \theta \right\rangle) \mathbb{1}(\mathcal{B}_T)}{\delta \mathbb{P}(\mathcal{B}_T)}\right) \quad (32)$$

We will now bound the exponential moment: $\mathbb{E}_M \mathbb{E}_{\theta \sim \pi} \exp(\gamma \left\langle M_t, \theta \right\rangle)$ whenever $\pi = \mathcal{N}(0, \mathbf{I})$.

**Theorem 12.** *Let $h(t) := \sum_{s=1}^t \log\left(1 + \frac{\gamma^2}{2} q_t \exp(\gamma^2 \Gamma^2) + \gamma^4 p_t g^2 T \exp(2\gamma^2 \Gamma g\sqrt{T})\right)$. Then,*

$$\mathbb{E}_{\theta \sim \pi} \exp(\gamma \langle M_t, \theta \rangle - h(t)) \mathbb{1}(\mathcal{B}_t)$$

*is a supermartingale with respect to the filtration $\mathcal{F}_t$*

*Proof.* Let $\Sigma_t := \mathbb{E}[\mathbf{v}_t \mathbf{v}_t^\mathsf{T} | \mathcal{F}_{t-1}]$ and $\nu_t := \|\Sigma_t\|$. First, consider $\mathbb{E}_{\theta \sim \pi} \exp(\gamma \langle M_t, \theta \rangle)$. By the properties of the Gaussians, we must have almost surely:

$$\mathbb{E}_{\theta \sim \pi} \exp(\gamma \langle M_t, \theta \rangle) \mathbb{1}(\mathcal{B}_t) = \exp(\tfrac{\gamma^2 \|M_t\|^2}{2}) \mathbb{1}(\mathcal{B}_t) \quad (33)$$

Using the fact that $\|M_t\|^2 = \|\mathbf{v}_t\|^2 + 2\langle \mathbf{v}_t, M_{t-1} \rangle + \|M_{t-1}\|^2$, we have:

$$\mathbb{E}\left[\exp(\tfrac{\gamma^2 \|M_t\|^2}{2}) \mathbb{1}(\mathcal{B}_t) \Big| \mathcal{F}_{t-1}\right] = \mathbb{E}\left[\exp(\tfrac{\gamma^2 \|M_{t-1}\|^2}{2} + \tfrac{\gamma^2 \|\mathbf{v}_t\|^2}{2} + \gamma^2 \langle \mathbf{v}_t, M_{t-1} \rangle) \mathbb{1}(\mathcal{B}_t)\right]$$

$$= \mathbb{E}\left[\exp(\tfrac{\gamma^2 \|\mathbf{v}_t\|^2}{2} + \gamma^2 \langle \mathbf{v}_t, M_{t-1} \rangle) \mathbb{1}(\mathcal{A}_t) \Big| \mathcal{F}_{t-1}\right] \exp(\tfrac{\gamma^2 \|M_{t-1}\|^2}{2}) \mathbb{1}(\mathcal{B}_{t-1}) \quad (34)$$

We will now bound the quantity: $\mathbb{E}\left[\exp(\frac{\gamma^2\|\mathbf{v}_t\|^2}{2} + \gamma^2\langle\mathbf{v}_t, M_{t-1}\rangle)\mathbb{1}(\mathcal{A}_t)\Big|\mathcal{F}_{t-1}\right]$. Using the convexity of $x \to \exp(x)$, we conclude:

$$\mathbb{E}\left[\exp(\frac{\gamma^2\|\mathbf{v}_t\|^2}{2} + \gamma^2\langle\mathbf{v}_t, M_{t-1}\rangle)\mathbb{1}(\mathcal{A}_t)\Big|\mathcal{F}_{t-1}\right]$$

$$\leq \mathbb{E}\left[\frac{1}{2}\exp(\gamma^2\|\mathbf{v}_t\|^2)\mathbb{1}(\mathcal{A}_t) + \frac{1}{2}\exp(2\gamma^2\langle\mathbf{v}_t, M_{t-1}\rangle)\mathbb{1}(\mathcal{A}_t)\Big|\mathcal{F}_{t-1}\right]$$

$$\leq \frac{1}{2}\left[1 + \gamma^2\mathsf{Tr}(\Sigma_t)\exp(\gamma^2\Gamma^2)\right] + \mathbb{E}\left[\frac{1}{2}\exp(2\gamma^2\langle\mathbf{v}_t, M_{t-1}\rangle)\mathbb{1}(\mathcal{A}_t)\Big|\mathcal{F}_{t-1}\right] \quad (35)$$

In the second step, we have used the fact that $\exp(\gamma^2\|\mathbf{v}_t\|^2)\mathbb{1}(\mathcal{A}_t) \leq 1 + \gamma^2\|\mathbf{v}_t\|^2\exp(\gamma^2\Gamma^2)$ almost surely using the power series expansion of the $\exp()$ function. Using the power series expansion of $\exp(x)$, we have:

$$\mathbb{E}\left[\exp(2\gamma^2\langle\mathbf{v}_t, M_{t-1}\rangle)\mathbb{1}(\mathcal{A}_t)\Big|\mathcal{F}_{t-1}\right] \leq \mathbb{E}\left[\exp(2\gamma^2\langle\mathbf{v}_t, M_{t-1}\rangle)\Big|\mathcal{F}_{t-1}\right]$$

$$= 1 + 2\gamma^2\mathbb{E}[\langle\mathbf{v}_t, M_{t-1}\rangle|\mathcal{F}_{t-1}] + \sum_{k\geq2}\frac{2^k\gamma^{2k}}{k!}\mathbb{E}[(\langle\mathbf{v}_t, M_{t-1}\rangle)^k|\mathcal{F}_{t-1}]$$

$$\leq 1 + \sum_{k\geq2}\frac{2^k\gamma^{2k}}{k!}\mathbb{E}[(\langle\mathbf{v}_t, M_{t-1}\rangle)^2\Gamma^{k-2}\|M_{t-1}\|^{k-2}|\mathcal{F}_{t-1}]$$

$$\leq 1 + \sum_{k\geq2}\frac{2^k\gamma^{2k}}{k!}\langle M_{t-1}, \Sigma_t M_{t-1}\rangle\Gamma^{k-2}\|M_{t-1}\|^{k-2}$$

$$\leq 1 + \sum_{k\geq2}\frac{2^k\gamma^{2k}}{k!}\nu_t\Gamma^{k-2}\|M_{t-1}\|^k \leq 1 + 2\gamma^4\nu_t\|M_{t-1}\|^2\exp(2\gamma^2\|M_{t-1}\|\Gamma) \quad (36)$$

Here, $\nu_t = \|\Sigma_t\|_{\mathsf{op}}$ In the second step we have used the fact that $\mathbb{E}[\mathbf{v}_t|\mathcal{F}_{t-1}] = 0$ and the fact that $\langle\mathbf{v}_t, M_{t-1}\rangle \leq \Gamma\|M_{t-1}\|$ almost surely. Plugging Equation (36) into Equation (35), we conclude:

$$\mathbb{E}\left[\exp(\frac{\gamma^2\|\mathbf{v}_t\|^2}{2} + \gamma^2\langle\mathbf{v}_t, M_{t-1}\rangle)\mathbb{1}(\mathcal{A}_t)\Big|\mathcal{F}_{t-1}\right]$$

$$\leq 1 + \frac{\gamma^2}{2}\mathsf{Tr}(\Sigma_t)\exp(\gamma^2\Gamma^2) + \gamma^4\nu_t\|M_{t-1}\|^2\exp(2\gamma^2\Gamma\|M_{t-1}\|) \quad (37)$$

Using Equation (37) and that under the event $\mathcal{B}_{t-1}$ we must have $\|M_{t-1}\| \leq g\sqrt{T}$, we conclude:

$$\mathbb{E}\left[\exp(\frac{\gamma^2\|M_t\|^2}{2})\mathbb{1}(\mathcal{B}_t)\Big|\mathcal{F}_{t-1}\right]$$

$$\leq \left(1 + \frac{\gamma^2}{2}q_t\exp(\gamma^2\Gamma^2) + \gamma^4 p_t g^2 T\exp(2\gamma^2\Gamma g\sqrt{T})\right)\exp(\frac{\gamma^2\|M_{t-1}\|^2}{2})\mathbb{1}(\mathcal{B}_{t-1})$$

$$= \exp(h(t) - h(t-1))\exp\left(\gamma^2\frac{\|M_{t-1}\|^2}{2}\right)\mathbb{1}(\mathcal{B}_{t-1}) \quad (38)$$

Therefore, by induction, we conclude the statement of the theorem.

$\square$

**Theorem 13.** *For any stopping time $H$,*

$$\mathbb{E}_M\mathbb{E}_{\theta\sim\pi}\exp(\gamma\langle M_{\min(H,T)}, \theta\rangle)\mathbb{1}(\mathcal{B}_T) \leq \exp(h(T)) \quad (39)$$

Where $h(T) = \sum_{t=1}^{T} \log\left(1 + \frac{\gamma^2 q_t}{2} \exp(\gamma^2\Gamma^2) + \gamma^4 p_t g^2 T \exp(2\gamma^2\Gamma g\sqrt{T})\right)$

*Proof.* From Theorem 12 and the optional stopping theorem, we conclude that the following quantity is a super-martingale:

$$M_t^{\text{exp}} := \mathbb{E}_{\theta\sim\pi} \exp(\gamma \langle M_{\min(H,t)}, \theta \rangle - h(\min(H,t))) \mathbb{1}(\mathcal{B}_{\min(H,t)})$$

Therefore, we have:

$$\mathbb{E}_{\theta\sim\pi} \exp(\gamma \langle M_{\min(H,T)}, \theta \rangle - h(T)) \mathbb{1}(\mathcal{B}_T) \leq M_T^{\text{exp}} \leq \mathbb{E} M_0^{\text{exp}} = 1$$

$\square$

Combining Theorem 13 and Equation (32), we conclude that the following inequality holds with probability at-least $1 - \delta$ when conditioned on $\mathcal{B}_T$:

$$\sup_{\rho\in\mathcal{M}_1(\mathbb{R}^d)} \mathbb{E}_{\theta\sim\rho} \gamma \langle M_{\min(T,H)}, \theta \rangle - \mathsf{KL}\left(\rho\|\pi\right) \leq h(T) + \log(\tfrac{1}{\delta\mathbb{P}(\mathcal{B}_T)})$$

In the RHS of the inequality above, we replace the supremum over $\mathcal{M}_1$ with the supremum over the set of all probability distributions $\{\mathcal{N}(\alpha\xi, \mathbf{I})$ such that $\xi \in \mathcal{S}^{d-1}, \alpha \geq 0\}$. We note that $\mathsf{KL}\left(\mathcal{N}(\alpha\xi, \mathbf{I})\|\pi\right) = \frac{\alpha^2}{2}$ to conclude that the following inequality holds with probability at-least $1 - \delta$ when conditioned on $\mathcal{B}_T$:

$$\sup_{\alpha>0} \gamma\alpha\|M_{\min(H,T)}\| - \frac{\alpha^2}{2} \leq h(T) + \log(\tfrac{1}{\delta\mathbb{P}(\mathcal{B}_T)})$$

That is:

$$\|M_{\min(H,T)}\| \leq \sqrt{\frac{2h(T) + 2\log(\frac{1}{\delta\mathbb{P}(\mathcal{B}_T)})}{\gamma^2}}$$

Now, note that by definition,

$$\begin{aligned}
\frac{h(T)}{T} &= \frac{1}{T} \sum_{t=1}^{T} \log\left(1 + \frac{\gamma^2}{2} q_t \exp(\gamma^2\Gamma^2) + \gamma^4 p_t g^2 T \exp(2\gamma^2\Gamma g\sqrt{T})\right) \\
&\leq \frac{\gamma^2}{2} \bar{q} \exp(\gamma^2\Gamma^2) + \gamma^4 \bar{p} g^2 T \exp(2\gamma^2\Gamma g\sqrt{T})
\end{aligned} \tag{40}$$

Therefore, whenever: $\gamma \leq \min\left(\frac{1}{\Gamma}, \frac{1}{2\sqrt{\Gamma g\sqrt{T}}}\right)$, we note with probability at-least $1 - \delta$ conditioned on the event $\mathcal{B}_T$:

$$\|M_{\min(H,T)}\| \lesssim \sqrt{T\bar{q} + \gamma^2 \bar{p} g^2 T^2 + \tfrac{1}{\gamma^2} \log\left(\tfrac{1}{\delta\mathbb{P}(\mathcal{B}_T)}\right)}$$

We therefore state the following theorem:

**Theorem 14.** *Suppose $\delta, \delta_1 \in (0, \frac{1}{2})$. If $M_t$ satisfies $(g, T, \delta)$ uniform concentration for some $\delta < \frac{1}{2}$. Then $M_t$ also satisfies $(g', T, \delta + \delta_1)$ concentration, where*

$$(g')^2 = C\left[\bar{q} + \gamma^2 \bar{p} g^2 T + \frac{\log(\frac{1}{\delta_1})}{\gamma^2 T}\right]$$

*for any $\gamma \leq \min\left(\frac{1}{\Gamma}, \frac{1}{2\sqrt{\Gamma g\sqrt{T}}}\right)$.*

Additionally, suppose $g \geq c_0 \frac{\Gamma}{\sqrt{T}}$ for some fixed constant $c_0 > 0$, then we have for some constant $C_{\text{iter}}(c_0)$:

$$(g')^2 = C_{\text{iter}}(c_0)[\bar{q} + g\left(\frac{\bar{p}\sqrt{T}}{\Gamma} + \frac{\Gamma}{\sqrt{T}}\log(\frac{1}{\delta_1})\right)]$$

*Proof.* Since $\delta \leq \frac{1}{2}$, we conclude that $\mathbb{P}(\mathcal{B}_T) \geq \frac{1}{2}$. Given that $M_t$ satisfies $(g, T, \delta)$ uniform concentration. We conclude from the discussion above that for some universal constant $C$ and any $\gamma \leq \min\left(\frac{1}{\Gamma}, \frac{1}{2\sqrt{\Gamma g \sqrt{T}}}\right)$, we have:

$$\sup_H \mathbb{P}(\|M_{\min(H,T)}\|^2 \geq C[T\bar{q} + \gamma^2 \bar{p} g^2 T^2 + \frac{1}{\gamma^2}\log\left(\frac{1}{\delta_1}\right)]|\mathcal{B}_T) \leq \delta_1$$

Picking $H$ to be the stopping time given by $H = \inf\{t \geq 0 : \|M_t\|^2 \geq C[T\bar{q} + \gamma^2 \bar{p} g^2 T^2 + \frac{1}{\gamma^2}\log\left(\frac{1}{\delta_1}\right)]\}$ where $C$ is the same constant as in the equation above, we conclude:

$$\mathbb{P}(\sup_{t \leq T} \|M_t\|^2 \geq C[T\bar{q} + \gamma^2 \bar{p} g^2 T^2 + \frac{1}{\gamma^2}\log\left(\frac{1}{\delta_1}\right)]|\mathcal{B}_T) \leq \delta_1$$

Only in this proof, call the event $\mathcal{G} := \{\sup_{t \leq T} \|M_t\|^2 \geq C[T\bar{q} + \gamma^2 \bar{p} g^2 T^2 + \frac{1}{\gamma^2}\log\left(\frac{1}{\delta_1}\right)]\}$. We have:

$$\mathbb{P}(\mathcal{G}) = \mathbb{P}(\mathcal{G} \cap \mathcal{B}_T) + \mathbb{P}(\mathcal{G} \cap \mathcal{B}_T^\complement) \leq \mathbb{P}(\mathcal{G}|\mathcal{B}_T) + \mathbb{P}(\mathcal{B}_T^\complement) \leq \delta_1 + \delta$$

Whenever $g \geq c_0 \frac{\Gamma}{\sqrt{T}}$, we can pick $\lambda = \frac{c_1(c_0)}{\sqrt{\Gamma g \sqrt{T}}}$ and conclude the result.

$\square$

We now state consider Lemma 11 from [2].

**Lemma 18.** *Suppose $\alpha, \beta \leq 0$ with $\alpha + \beta > 0$. Consider the function $f : \mathbb{R}^+ \to \mathbb{R}^+$ given by $f(u) = \alpha + \beta\sqrt{u}$. Then, $f$ has the unique fixed point: $u^* := \left(\frac{\beta + \sqrt{\beta^2 + 4\alpha}}{2}\right)^2$. For $t \in \mathbb{N}$, denoting $f^{(t)}$ to be the $t$ fold composition of $f$ with itself, we have for any $u \in \mathbb{R}^+$:*

$$|f^{(t)}(u) - u^*| \leq \beta^{(2 - \frac{1}{2^{t-1}})}|u - u^*|^{\frac{1}{2^t}}.$$

We are now ready to prove the main theorem 9

*Proof of Theorem 9.* It is sufficient to show that there exists $K = \log \Theta(\log(\Gamma T d \log(\frac{1}{\delta})))$ such that $M_t$ obeys $(g, T, \delta)$ uniform concentration where $g = C \max(\frac{\Gamma}{\sqrt{T}}, \bar{q} + \frac{\bar{p}\sqrt{T}}{\Gamma} + \frac{\Gamma}{\sqrt{T}}\log(\frac{K}{\delta}))$

Let $K \in \mathbb{N}$ be any fixed integer. By Theorem 6, we conclude that the martingale $M_t$ is $(g_0(\frac{\delta}{K}), T, \frac{\delta}{K})$ uniformly concentrated. Fix some $c_0 > 0$ and $C_{\text{iter}}(c_0)$ be as in Theorem 14.

Define the sequence $g_i := \sqrt{C_{\text{iter}}(c_0)\bar{q}} + \sqrt{C_{\text{iter}}(c_0)g_{i-1}G}$ where $G = \frac{\bar{p}\sqrt{T}}{\Gamma} + \frac{\Gamma}{\sqrt{T}}\log(\frac{K}{\delta})$.

If $g_0 \leq c_0 \frac{\Gamma}{\sqrt{T}}$, then the statement of the theorem follows. Suppose there exists $K_1 \leq K - 1$ such that $g_{K_1} \leq \frac{c_0 \Gamma}{\sqrt{T}}$ and suppose that it is the first such integer. If $K_1 = 0$, the statement of the theorem follows from $(g_0(\frac{\delta}{K}), T, \frac{\delta}{K})$ uniform concentration of $M_t$. Suppose $1 \leq K_1 \leq K - 1$. Then, $\min(g_0, \ldots, g_{K_1-1}) \geq c_0 \frac{\Gamma}{\sqrt{T}}$. Then, by Theorem 14, the fact that $\sqrt{x + y} \leq \sqrt{x} + \sqrt{y}$ and induction, we conclude that $M_t$ obeys $(g_i, T, \frac{(i+1)\delta}{K})$ for every $i \leq K_1$. Thus we conclude the statement of the theorem.

Suppose such a $K_1$ does not exist. Then, $\min(g_0, \ldots, g_{K-1}) \geq c_0 \frac{\Gamma}{\sqrt{T}}$. Then, by Theorem 14, the fact that $\sqrt{x+y} \leq \sqrt{x} + \sqrt{y}$ and induction, we conclude that $M_t$ obeys $(g_i, T, \frac{(i+1)\delta}{K})$ for every $i \leq K - 1$. Therefore, it obeys $(g_K, T, \delta)$ uniform concentration.

Consider the function $f$ in Lemma 18 with $\alpha = \sqrt{C_{\text{iter}}(c_0)\bar{q}}$ and $\beta = \sqrt{C_{\text{iter}}(c_0)G}$ and let the corresponding fixed point be denoted by $g^*$. It is easy to show that the fixed point $g^* \lesssim \sqrt{\bar{q}} + G$. After $K$ iterations, we must have:

$$g_K \leq g^* + (C_{\text{iter}}(c_0)G^{1-\frac{1}{2K}})|g_0 - g^*|^{\frac{1}{2K}} \lesssim g^* + (G^{1-\frac{1}{2K}})|g_0|^{\frac{1}{2K}}$$

We can show that picking $K = \log \Theta(\log((1 + \frac{\sqrt{\bar{q}}T}{\Gamma}) \log d))$, and the bound on $\Gamma$, we conclude the result. $\qquad \square$

### F.2  Proof of Theorem 10

*Proof of Theorem 10.* Recall that $\Sigma_t := \mathbb{E}[\mathbf{v}_t \mathbf{v}_t^\mathsf{T} | \mathcal{F}_{t-1}]$, $\nu_t := \|\Sigma_t\|$ and $N_t := \|M_t\|^2 - \sum_{s=1}^t \|\mathbf{v}_t\|^2$. Note that $\nu_t \leq p_t$ and $\mathsf{Tr}(\Sigma_t) \leq p_t$ almost surely.

Let $\gamma \in \mathbb{R}$. Define $h_N(t) := \sum_{s=1}^t \log\left(1 + 4\gamma^2 p_s g^2 T \exp(2|\gamma|\Gamma g\sqrt{T})\right)$ with empty sum denoting 0. We first show that $N_t^{\text{exp}} = \exp(\gamma N_t - h_N(t))\mathbb{1}(\mathcal{B}_T)$ is a super martingale with respect to the filtration $\mathcal{F}_t$ for $0 \leq t \leq T$. For $T \geq t > 1$, we have:

$$\mathbb{E}[\exp(\gamma N_t)\mathbb{1}(\mathcal{B}_t)|\mathcal{F}_{t-1}] = \exp(\gamma N_{t-1})\mathbb{1}(\mathcal{B}_{t-1})\mathbb{E}[\exp(2\gamma\langle \mathbf{v}_t, M_{t-1}\rangle)\mathbb{1}(\mathcal{B}_t)|\mathcal{F}_{t-1}]$$

$$\leq \exp(\gamma N_{t-1})\mathbb{1}(\mathcal{B}_{t-1})\mathbb{E}[\sum_{k=0}^\infty \frac{1}{k!}2^k\gamma^k\langle \mathbf{v}_t, M_{t-1}\rangle^k\mathbb{1}(\mathcal{B}_{t-1})|\mathcal{F}_{t-1}]$$

$$= \exp(\gamma N_{t-1})\mathbb{1}(\mathcal{B}_{t-1})\mathbb{E}[\mathbb{1}(\mathcal{B}_{t-1}) + \sum_{k=2}^\infty \frac{1}{k!}2^k\gamma^k\langle \mathbf{v}_t, M_{t-1}\rangle^k\mathbb{1}(\mathcal{B}_{t-1})|\mathcal{F}_{t-1}]$$

$$\leq \exp(\gamma N_{t-1})\mathbb{1}(\mathcal{B}_{t-1})\mathbb{E}[1 + \sum_{k=2}^\infty \frac{1}{k!}2^k|\gamma|^k\langle \mathbf{v}_t, M_{t-1}\rangle^2\Gamma^{k-2}\|M_{t-1}\|^{k-2}\mathbb{1}(\mathcal{B}_{t-1})|\mathcal{F}_{t-1}]$$

$$\leq \exp(\gamma N_{t-1})\mathbb{1}(\mathcal{B}_{t-1})\mathbb{E}[1 + 4\gamma^2\langle \mathbf{v}_t, M_{t-1}\rangle^2 \exp(2|\gamma|\Gamma\|M_{t-1}\|)\mathbb{1}(\mathcal{B}_{t-1})|\mathcal{F}_{t-1}]$$

$$\leq \exp(\gamma N_{t-1})\mathbb{1}(\mathcal{B}_{t-1})\mathbb{E}[1 + 4\gamma^2\nu_t\|M_{t-1}\|^2 \exp(2|\gamma|\Gamma\|M_{t-1}\|)\mathbb{1}(\mathcal{B}_{t-1})|\mathcal{F}_{t-1}]$$

$$\leq \exp(\gamma N_{t-1})\mathbb{1}(\mathcal{B}_{t-1})\left(1 + 4\gamma^2\nu_t g^2 T \exp(2|\gamma|\Gamma g\sqrt{T})\right)$$

$$= \exp(\gamma N_{t-1} + h_N(t) - h_N(t-1))\mathbb{1}(\mathcal{B}_{t-1}) \tag{41}$$

This shows that $N_t^{\text{exp}}$ is a super-martingale. Using the fact that $N_1^{\text{exp}} = 1$ almost surely, the optional stopping theorem and the Chernoff bound, we conclude that for any stopping time $H$, we have for any $\alpha, \gamma > 0$

$$\begin{aligned}
\mathbb{P}(\{N_{\min(T,H)} > \alpha\} \cap \mathcal{B}_T) &\leq \mathbb{E}[\exp(\gamma N_{\min(T,H)} - \gamma\alpha)\mathbb{1}(\mathcal{B}_T)] \\
&\leq \mathbb{E}[\exp(\gamma N_{\min(T,H)} - \gamma\alpha)\mathbb{1}(\mathcal{B}_{\min(T,H)})] \\
&\leq \mathbb{E}[N_{\min(T,H)}^{\text{exp}}]\exp(h_N(T) - \gamma\alpha) \\
&\leq \exp(h_N(T) - \gamma\alpha) \tag{42}
\end{aligned}$$

Taking $\gamma = \frac{1}{2\Gamma g\sqrt{T}}$ allows us to conclude:

$$\mathbb{P}(\{N_{\min(T,H)} > \Gamma Cg\sqrt{T}\log(\frac{2}{\delta}) + \frac{C\nu gT^{3/2}}{\Gamma}\} \cap \mathcal{B}_T) \leq \frac{\delta}{2}$$

Let $\alpha = \Gamma Cg\sqrt{T}\log(\frac{2}{\delta}) + \frac{C\nu gT^{3/2}}{\Gamma}$ and take $H$ to be the stopping time $\min(\inf_t\{t > 0 : N_t > \alpha\}, T)$ where infimum of an empty set is taken to be infinity. We note that $\{\sup_{t \leq T} N_t > \alpha\} = \{N_{\min(T,H)} > \alpha\}$. We thus conclude:

$$\mathbb{P}(\{\sup_{t \leq T} N_t > \Gamma C g \sqrt{T} \log(\tfrac{2}{\delta}) + \tfrac{C \nu g T^{3/2}}{\Gamma}\} \cap \mathcal{B}_T) \leq \frac{\delta}{2}$$

Taking $\gamma$ negative gives the analogous proof for $N_t < -\alpha$.

$\square$

### F.3 Proof of Corollary 5

*Proof.* Consider the set $S = \{\mathsf{UP}(t) : 0 \leq t \leq T\}$. The, $|S| \leq \log_2(T) + 1$. By Corollary 6, we have for any $t_0 \in S$, the following is true with probability $1 - \frac{\delta}{1+\log_2(T)}$

$$\sum_{s=1}^{t_0} \|\mathbf{v}_s\|^2 \leq t_0 g^2 (t_0, \frac{\delta}{3(1 + \log_2(T))})$$

Therefore, by union bound of the above event over every $t_0 \in S$, we have with probability $1 - \delta$:

$$\sup_{t_0 \in S} \sum_{s=1}^{t_0} \|\mathbf{v}_s\|^2 \leq t_0 g^2 (t_0, \frac{\delta}{3(1 + \log_2(T))}) \leq 0$$

Now, note that $\sum_{s=1}^t \|\mathbf{v}_s\|^2 \leq \sum_{s=1}^{\mathsf{UP}(t)} \|\mathbf{v}_s\|^2$ almost surely for every $t \in [T]$ since $t \leq \mathsf{UP}(t)$. Therefore, we conclude that with probability at-least $1 - \delta$, the following holds for all $t \in [T]$ simultaneously:

$$\sum_{s=1}^t \|\mathbf{v}_s\|^2 \leq g^2 \left( \mathsf{UP}(t), \tfrac{\delta}{3(1+\log_2(T))} \right) \mathsf{UP}(t)$$

Using the definition of $g(,)$ from Theorem 9, we conclude the result. $\square$

## G Applications to Streaming Heavy Tailed Statistical Estimation

### G.1 Streaming Heavy Tailed Mean Estimation : Proof of Corollary 1

*Proof.* Recall that for this problem, $\Xi = \mathcal{C}$, $\mathbb{E}_{\xi \sim P}[\xi] = \mathbf{m} \in \mathcal{C}$ and $\mathsf{Cov}[\xi] \preceq \Sigma$. Consider the following quadratic loss function $f : \mathcal{C} \to \mathbb{R}$:

$$f(\mathbf{x}; \xi) = \tfrac{1}{2}\|\mathbf{x} - \xi\|^2, \qquad \xi \sim P$$

The associated population risk function $F$ is given by

$$F(\mathbf{x}) = \frac{1}{2} \cdot \mathbb{E}_{\xi \sim P} \left[ \|\mathbf{x} - \xi\|^2 \right] = F(\mathbf{x}) = \frac{1}{2}\|\mathbf{x} - \mathbf{m}\|^2 + \mathsf{Tr}(\mathsf{Cov}_{\xi \sim P}[\xi])$$

Note that $F$ is $L$-smooth and $\mu$-strongly convex with $L = \mu = 1$. Thus, $\kappa = 1$. Furthermore, $\mathbf{m}$ is the unique minimizer of $F$. Hence, solving the streaming heavy tailed mean estimation problem is equivalent to solving the SCO problem for $F$. To this end, we consider the following stochastic gradient oracle:

$$g(\mathbf{x}; \xi) = \mathbf{x} - \xi$$

It is easy to see that $\mathbb{E}_{\mathbf{y}}[g(\mathbf{x}; \xi)] = \nabla F(\mathbf{x})$, i.e., the stochastic gradient estimate is unbiased. The associated stochastic gradient noise $\mathbf{n}(\mathbf{x}; \xi)$ is given by

$$\mathbf{n}(\mathbf{x}; \xi) = \nabla F(\mathbf{x}) - \nabla f_{\mathbf{y}}(\mathbf{x}) = \mathbf{y} - \mathbf{m}$$

We now note that

$$\Sigma(\mathbf{x}) = \mathbb{E}[\mathbf{n}(\mathbf{x}; \xi)\mathbf{n}(\mathbf{x}; \xi)^T] = \mathbb{E}[(\mathbf{y} - \mathbf{m})(\mathbf{y} - \mathbf{m})^T] = \mathsf{Tr}(\mathsf{Cov}_{\xi \sim P}[\xi]) \preceq \Sigma$$

Hence, we note that the Bdd. $2^{\mathsf{nd}}$ Moment assumption is satisfied. Hence, the result follows by an application of Theorem 1 $\square$

## G.2 Streaming Heavy Tailed Linear Regression : Proof of Corollary 2

We use $\theta \in \mathcal{C}$ to denote the parameter of $F$. Recall from Section 5.2 that $\Xi = \mathbb{R}^d \times \mathbb{R}$, and given a target parameter $\theta^* \in \mathcal{C}$, $P$ defines the following linear model:

$$\mathbf{x} \sim Q, \; \mathbb{E}[\mathbf{x}] = 0, \; \mathbb{E}[\mathbf{x}\mathbf{x}^T] = \Sigma \succ 0; \qquad y = \langle \mathbf{x}, \theta^* \rangle + \epsilon, \; \mathbb{E}[\epsilon|\mathbf{x}] = 0, \; \mathbb{E}[\epsilon^2|\mathbf{x}] \leq \sigma^2$$

In addition, we make the following bounded $4^{\text{th}}$ moment asumption on the covariates $\mathbf{x}$

$$\mathbb{E}[\langle \mathbf{x}, \mathbf{v} \rangle^4] \leq C_4 (\mathbb{E}[\langle \mathbf{x}, \mathbf{v} \rangle^2])^2 \qquad \forall \, \mathbf{v} \in \mathbb{R}^d$$

for some numerical constant $C_4 \geq 1$. Recall that the sample loss function is given by:

$$f(\theta; \mathbf{x}, \mathbf{y}) = \frac{1}{2} (\langle \theta, \mathbf{x} \rangle - \mathbf{y})^2 = \frac{1}{2} (\langle \theta - \theta^*, \mathbf{x} \rangle - \epsilon)^2$$

Using the fact that $\mathbb{E}[\epsilon|\mathbf{x}] = 0, \mathbb{E}[\mathbf{x}] = 0$ and $\mathbb{E}[\mathbf{x}\mathbf{x}^T] = \Sigma$

$$F(\theta) = \frac{1}{2} (\theta - \theta^*)^T \mathbb{E}[\mathbf{x}\mathbf{x}^T](\theta - \theta^*) + \mathbb{E}[\epsilon^2]$$

$$= \frac{1}{2} (\theta - \theta^*)^T \Sigma (\theta - \theta^*) + \mathbb{E}[\epsilon^2]$$

We note that $\mathbb{E}[\epsilon^2] \leq \sigma^2$ as per our assumption hence $F$ is well defined. Furthermore.

$$\nabla F(\theta) = \Sigma(\theta - \theta^*)$$
$$\nabla^2 F(\theta) = \Sigma$$

Thus, the population risk $F$ is $L$-smooth and $\mu$-strongly convex with $L = \|\Sigma\|_2$ and $\mu = \lambda_{\min}(\Sigma)$, i.e., $\kappa = \frac{\|\Sigma\|_2}{\lambda_{\min}(\Sigma)}$. Furthermore, the unique minimizer of $F$ is $\theta^*$. Hence, $\kappa = \frac{\|\Sigma\|_2}{\lambda_{\min}(\Sigma)}$ the linear regression task of estimating $\theta^*$ is equivalent to solving SCO for the above objective.

The associated stochastic gradient oracle $g(\theta; \mathbf{x}, \mathbf{y})$ at any $\theta \in \mathcal{C}$ is given by:

$$g(\theta; \mathbf{x}, \mathbf{y}) = \nabla f(\theta; \mathbf{x}, \mathbf{y}) = \mathbf{x}(\langle \theta, \mathbf{x} \rangle - \mathbf{y}) = \mathbf{x}(\langle \theta - \theta^*, \mathbf{x} \rangle - \epsilon)$$

$$= \mathbf{x}\mathbf{x}^T (\theta - \theta^*) - \mathbf{x}\epsilon$$

We first show that $g(\theta; \mathbf{x}, \mathbf{y}))$ is indeed an unbiased estimate of $\nabla F(\theta)$

$$\mathbb{E}[g(\theta; \mathbf{x}, \mathbf{y})] = \mathbb{E}[\mathbf{x}\mathbf{x}^T](\theta - \theta^*) - \mathbb{E}[\mathbf{x}\mathbb{E}[\epsilon|\mathbf{x}]] = \Sigma(\theta - \theta^*) = \nabla F(\theta)$$

The associated stochastic gradient noise $\mathbf{n}(\theta; \mathbf{x}, \mathbf{y})(\theta)$ is given by

$$\mathbf{n}(\theta; \mathbf{x}, \mathbf{y})(\theta) = g(\theta; \mathbf{x}, \mathbf{y})(\theta) - \nabla F(\mathbf{x})$$

$$= \left(\mathbf{x}\mathbf{x}^T - \Sigma\right)(\theta - \theta^*) - \mathbf{x}\epsilon$$

$\Sigma(\theta) = \mathbb{E}[\mathbf{n}(\theta; \mathbf{x}, \mathbf{y})\mathbf{n}(\theta; \mathbf{x}, \mathbf{y})]$. For convenience, we use $\mathbf{M} = \mathbf{x}\mathbf{x}^T - \Sigma$ and $\mathbf{d}_\theta = \theta - \theta^*$ and note that $\mathbf{M}$ is symmetric. It follows that:

$$\Sigma(\theta) = \mathbb{E}\left[(\mathbf{M}\mathbf{d}_\theta - \mathbf{x}\epsilon)(\mathbf{M}\mathbf{d}_\theta - \mathbf{x}\epsilon)^T\right]$$

$$= \mathbb{E}\left[\mathbf{M}\mathbf{d}_\theta \mathbf{d}_\theta^T \mathbf{M}\right] + \mathbb{E}\left[\mathbf{x}\mathbf{x}^T \cdot \mathbb{E}\left[\epsilon^2|\mathbf{x}\right]\right] - \mathbb{E}[\mathbf{x}\mathbf{d}_\theta^T \mathbf{M} \cdot \mathbb{E}[\epsilon|\mathbf{x}]] - \mathbb{E}[\mathbf{M}\mathbf{d}_\theta \mathbf{x}^T \cdot \mathbb{E}[\epsilon|\mathbf{x}]]$$

$$\preceq \mathbb{E}\left[\mathbf{M}\mathbf{d}_\theta \mathbf{d}_\theta^T \mathbf{M}\right] + \sigma^2 \Sigma$$

where we use the fact that $\mathbb{E}[\epsilon|\mathbf{x}] = 0, \mathbb{E}[\epsilon^2|\mathbf{x}] \leq \sigma^2$ and $\mathbb{E}[\mathbf{x}\mathbf{x}^T] = \Sigma$.

We shall now upper bound $\|\Sigma(\theta)\|_2$. To do so, we define $\mathbf{A}(\theta) = \mathbb{E}\left[\mathbf{M}\mathbf{d}_\theta \mathbf{d}_\theta^T \mathbf{M}\right]$ and note that $\mathbf{A}(\theta)$ is a PSD matrix since for any $\mathbf{v} \in \mathbb{R}^d$, $\mathbf{v}^T \mathbf{A}(\theta)\mathbf{v} = \mathbb{E}\left[(\mathbf{v}^T \mathbf{M}\mathbf{d}_\theta)^2\right] \geq 0$. Without loss of generality,

we assume $\theta \neq \theta^*$ and observe that

$$
\begin{aligned}
\sup_{\|\mathbf{v}\|=1} \mathbb{E}[\mathbf{v}^T \mathbf{A}(\theta)\mathbf{v}] &= \sup_{\|\mathbf{v}\|=1} \mathbb{E}[\langle \mathbf{d}_\theta, \mathbf{M}\mathbf{v} \rangle^2] \\
&= \|\mathbf{d}_\theta\|^2 \sup_{\|\mathbf{v}\|=1} \mathbb{E}[\langle \tfrac{\mathbf{d}_\theta}{\|\mathbf{d}_\theta\|}, \mathbf{M}\mathbf{v} \rangle^2] \\
&\leq \|\mathbf{d}_\theta\|^2 \sup_{\|\mathbf{v}\|=1,\|\mathbf{w}\|=1} \mathbb{E}[\langle \mathbf{w}, \mathbf{M}\mathbf{v} \rangle^2] \\
&= \|\mathbf{d}_\theta\|^2 \sup_{\|\mathbf{v}\|=1,\|\mathbf{w}\|=1} \mathbb{E}[(\mathbf{w}^T (\mathbf{x}\mathbf{x}^T - \Sigma)\mathbf{v})^2] \\
&\leq \|\mathbf{d}_\theta\|^2 \sup_{\|\mathbf{v}\|=1,\|\mathbf{w}\|=1} \mathbb{E}\left[(\langle \mathbf{w}, \mathbf{x} \rangle \cdot \langle \mathbf{v}, \mathbf{x} \rangle - \mathbf{w}^T \Sigma \mathbf{v})^2\right] \\
&\leq \|\mathbf{d}_\theta\|^2 \sup_{\|\mathbf{v}\|=1,\|\mathbf{w}\|=1} 2(\mathbf{w}^T \Sigma \mathbf{v})^2 + 2\mathbb{E}\left[\langle \mathbf{w}, \mathbf{x} \rangle^2 \langle \mathbf{v}, \mathbf{x} \rangle^2\right] \\
&\leq 2\|\mathbf{d}_\theta\|^2 \left( \|\Sigma\|_2^2 + \sup_{\|\mathbf{v}\|=1,\|\mathbf{w}\|=1} \sqrt{\mathbb{E}[\langle \mathbf{w}, \mathbf{x} \rangle^4]}\sqrt{\mathbb{E}[\langle \mathbf{v}, \mathbf{x} \rangle^4]} \right) \\
&\leq 2\|\mathbf{d}_\theta\|^2 \left( \|\Sigma\|_2^2 + C_4 \sup_{\|\mathbf{v}\|=1,\|\mathbf{w}\|=1} \mathbb{E}[\langle \mathbf{w}, \mathbf{x} \rangle^2] \cdot \mathbb{E}[\langle \mathbf{v}, \mathbf{x} \rangle^2] \right) \\
&\leq 2\|\mathbf{d}_\theta\|^2 \left( \|\Sigma\|_2^2 + C_4 \sup_{\|\mathbf{w}\|=1} \mathbf{w}^T \Sigma \mathbf{w} \cdot \sup_{\|\mathbf{v}\|=1} \mathbf{v}^T \Sigma \mathbf{v} \right) \\
&\leq \|\mathbf{d}_\theta\|^2 \cdot 2\|\Sigma\|^2(C_4 + 1)
\end{aligned}
$$

where we use the fourth moment assumption on the covariates in the eighth step. Note that the above bound also holds when $\theta = \theta^*$ since in that case $\mathbf{A}(\theta) = 0$ and $\mathbf{d}_\theta = 0$. It follows that

$$
\begin{aligned}
\|\Sigma(\theta)\| &\leq \|A(\theta)\| + \sigma^2\|\Sigma\| \\
&\leq 2(C_4 + 1)\|\Sigma\|^2\|\theta - \theta^*\|^2 + \sigma^2\|\Sigma\|
\end{aligned}
$$

We shall now derive an upper bound for $\mathsf{Tr}(\Sigma(\theta))$ as follows:

$$
\begin{aligned}
\mathsf{Tr}(\Sigma(\theta)) &= \mathbb{E}[\|\mathbf{n}(\theta; \mathbf{x}, \mathbf{y})\|^2] \\
&= \mathbb{E}\left[\|\mathbf{M}\mathbf{d}_\theta - \mathbf{x}\epsilon\|^2\right] \\
&= \mathbb{E}[\|\mathbf{M}\mathbf{d}_\theta\|^2] - 2\mathbb{E}[\langle \mathbf{M}\mathbf{d}_\theta, \mathbf{x} \rangle \mathbb{E}[\epsilon|\mathbf{x}]] + \mathbb{E}[\|\mathbf{x}\|^2\mathbb{E}[\epsilon^2|\mathbf{x}]] \\
&\leq \mathbb{E}[\|\mathbf{M}\mathbf{d}_\theta\|^2] + \sigma^2\mathsf{Tr}(\Sigma)
\end{aligned}
$$

We now control $\mathbb{E}[\|\mathbf{M}\mathbf{d}_\theta\|^2]$. Note that $\mathbb{E}[\|\mathbf{M}\mathbf{d}_\theta\|^2] = 0$ if $\theta = \theta^*$ so we shall now consider the case when $\theta \neq \theta^*$. To this end, let $\mathbf{e}_1, \ldots, \mathbf{e}_d$ be an orthonormal basis of $\mathbb{R}^d$ such that $\mathbf{e}_1 = \frac{\mathbf{d}_\theta}{\|\mathbf{d}_\theta\|}$.

For the remainder of the proof, we use $\Sigma_{ij}$ to denote $\Sigma_{ij} = \mathbf{e}_i^T \Sigma \mathbf{e}_j$ where $i, j \in [d]$, which implies that $\mathsf{Tr}(\Sigma) = \sum_{i=1}^d \Sigma_{ii}$. We also note that for any two symmetric matrices $\mathbf{B}, \mathbf{C}$, $(\mathbf{B} - \mathbf{C})^2 \preceq 2\mathbf{B}^2 + 2\mathbf{C}^2$.

Hence,

$$\mathbb{E}[\|\mathbf{M}\mathbf{d}_\theta\|^2] = \|\mathbf{d}_\theta\|^2 \mathbb{E}\left[\|\mathbf{M}\mathbf{e}_1\|^2\right]$$

$$= \|\mathbf{d}_\theta\|^2 \mathbb{E}\left[\mathbf{e}_1^T(\Sigma - \mathbf{x}\mathbf{x}^T)^2\mathbf{e}_1\right]$$

$$\leq 2\|\mathbf{d}_\theta\|^2 \mathbb{E}\left[\mathbf{e}_1^T\left(\Sigma^2 + (\mathbf{x}\mathbf{x}^T)^2\right)\mathbf{e}_1\right]$$

$$\leq 2\|\mathbf{d}_\theta\|^2 \left(\mathbf{e}_1^T\Sigma^2\mathbf{e}_1 + \mathbb{E}\left[\langle\mathbf{e}_1,\mathbf{x}\rangle^2\|\mathbf{x}\|^2\right]\right)$$

$$\leq 2\|\mathbf{d}_\theta\|^2 \left(\|\Sigma^2\| + \mathbb{E}\left[\langle\mathbf{e}_1,\mathbf{x}\rangle^2\sum_{i=1}^d\langle\mathbf{e}_i,\mathbf{x}\rangle^2\right]\right)$$

$$\leq 2\|\mathbf{d}_\theta\|^2 \left(\|\Sigma\|^2 + \mathbb{E}\left[\langle\mathbf{e}_1,\mathbf{x}\rangle^4\right] + \sum_{i=2}^d\mathbb{E}[\langle\mathbf{e}_1,\mathbf{x}\rangle^2\langle\mathbf{e}_i,\mathbf{x}\rangle^2]\right)$$

$$\leq 2\|\mathbf{d}_\theta\|^2 \left(\|\Sigma\|^2 + \mathbb{E}\left[\langle\mathbf{e}_1,\mathbf{x}\rangle^4\right] + \sum_{i=2}^d\sqrt{\mathbb{E}\left[\langle\mathbf{e}_1,\mathbf{x}\rangle^4\right]\mathbb{E}\left[\langle\mathbf{e}_i,\mathbf{x}\rangle^4\right]}\right)$$

$$\leq 2\|\mathbf{d}_\theta\|^2 \left(\|\Sigma\|^2 + C_4\mathbb{E}\left[\langle\mathbf{e}_1,\mathbf{x}\rangle^2\right]^2 + C_4\sum_{i=2}^d\mathbb{E}\left[\langle\mathbf{e}_1,\mathbf{x}\rangle^2\right]\mathbb{E}\left[\langle\mathbf{e}_i,\mathbf{x}\rangle^2\right]\right)$$

$$\leq 2\|\mathbf{d}_\theta\|^2 \left(\|\Sigma\|^2 + C_4\sum_{i=1}^d\mathbb{E}\left[\langle\mathbf{e}_1,\mathbf{x}\rangle^2\right]\mathbb{E}\left[\langle\mathbf{e}_i,\mathbf{x}\rangle^2\right]\right)$$

$$\leq 2\|\mathbf{d}_\theta\|^2 \left(\|\Sigma\|^2 + C_4(\mathbf{e}_1^T\Sigma\mathbf{e}_1)\sum_{i=1}^d(\mathbf{e}_i^T\Sigma\mathbf{e}_i)\right)$$

$$\leq 2\|\mathbf{d}_\theta\|^2 \left(\|\Sigma\|^2 + C_4(\mathbf{e}_1^T\Sigma\mathbf{e}_1)\sum_{i=1}^d\Sigma_{ii}\right)$$

$$\leq 2\|\mathbf{d}_\theta\|^2 \left(\|\Sigma\|\mathsf{Tr}(\Sigma) + C_4\|\Sigma\|\mathsf{Tr}(\Sigma)\right)$$

$$\leq 2(C_4+1)\|\Sigma\|_2\mathsf{Tr}(\Sigma)\|\mathbf{d}_\theta\|^2$$

Clearly, the above bound holds even when $\theta = \theta^*$. Hence, we infer that

$$\mathsf{Tr}(\Sigma(\theta)) \leq 2(C_4+1)\|\Sigma\|_2\mathsf{Tr}(\Sigma)\|\theta - \theta^*\|^2 + \sigma^2\mathsf{Tr}(\Sigma)$$

From these bounds, we can conclude the following

$$\|\Sigma(\theta)\| \leq 2(C_4+1)\|\Sigma\|_2^2\|\theta - \theta^*\|^2 + \sigma^2\|\Sigma\|$$

$$\mathsf{Tr}(\Sigma(\theta)) \leq \frac{\mathsf{Tr}(\Sigma)}{\|\Sigma\|_2}\left[2(C_4+1)\|\Sigma\|_2^2\|\theta - \theta^*\|^2 + \sigma^2\|\Sigma\|\right]$$

Thus, the stochastic gradient oracle satisfies Assumption QG 2$^{\text{nd}}$ Moment with $\alpha = 2(C_4+1)\|\Sigma\|_2^2$, $\beta = \sigma^2\|\Sigma\|$ and $d_{\text{eff}} = \mathsf{Tr}(\Sigma)/\|\Sigma\|$. Hence, the result follows by an application of Theorem 2

### G.3  Heavy Tailed Streaming Logistic Regression : Proof of Corollary 3

Recall from Section 5.4 that $\Xi = \mathbb{R}^d \times \{0,1\}$ and $P$ denotes the following linear-logistic model:

$$\mathbf{x} \sim Q, \ \mathbb{E}[\mathbf{x}] = 0, \ \mathbb{E}[\mathbf{x}\mathbf{x}^T] \preceq \Sigma; \qquad y \sim \mathsf{Bernoulli}(\phi(\langle\theta^*,\mathbf{x}\rangle))$$

where $\phi(t) = (1 + e^{-t})^{-1}$. The covariates $\mathbf{x}$ are heavy tailed, with only bounded second moments.

The sample-level loss is given by the negative log likelihood of $y|\mathbf{x}$ as follows:

$$f(\theta;\mathbf{x},y) = \ln(1 + \exp(\langle\mathbf{x},\theta\rangle)) - y\langle\mathbf{x},\theta\rangle$$

The associated population loss and stochastic gradient oracle is given by

$$F(\theta) = \mathbb{E}_{\mathbf{x},y\sim P}\left[\ln(1 + \exp(\langle\mathbf{x},\theta\rangle)) - y\langle\mathbf{x},\theta\rangle\right]$$

$$g(\theta;\mathbf{x},\mathbf{y}) = \phi(\langle\mathbf{x},\theta\rangle)\mathbf{x} - y\mathbf{x}$$

We now compute the gradient and the Hessian of $F$

$$\nabla F(\theta) = \mathbb{E}\left[\frac{\exp(\langle \mathbf{x}, \theta \rangle)}{1 + \exp(\langle \mathbf{x}, \theta \rangle)} \cdot \mathbf{x} - \phi(\langle \mathbf{x}, \theta^* \rangle)\mathbf{x}\right]$$
$$= \mathbb{E}\left[(\phi(\langle \mathbf{x}, \theta \rangle) - \phi(\langle \mathbf{x}, \theta^* \rangle))\,\mathbf{x}\right]$$
$$\nabla^2 F(\theta) = \mathbb{E}[\phi'(\langle \mathbf{x}, \theta \rangle)\mathbf{x}\mathbf{x}^T]$$
$$= \mathbb{E}[\phi(\langle \mathbf{x}, \theta \rangle)(1 - \phi(\langle \mathbf{x}, \theta \rangle))\mathbf{x}\mathbf{x}^T]$$

Since $0 \le \phi(t) \le 1$ for every $t \in \mathbb{R}$, we note that $0 \preceq \nabla^2 F(\theta) \preceq \mathbb{E}[\mathbf{x}\mathbf{x}^T] \preceq \Sigma$ (as $E[\mathbf{x}] = 0$). Hence, $F$ is convex and $L$ smooth with $L = \|\Sigma\|_2$. Furthermore, since $\nabla F(\theta^*) = 0$ and $F$ is convex, we conclude that $\theta^*$ is a minimizer of $F$.

It is easy to see that $\mathbb{E}\left[g(\theta; \mathbf{x}, y)\right] = \mathbb{E}\left[(\phi(\langle \mathbf{x}, \theta \rangle) - \phi(\langle \mathbf{x}, \theta^* \rangle))\,\mathbf{x}\right] = \nabla F(\theta)$, i.e., the stochastic gradient is unbiased. Let $\mathbf{n}(\theta; \mathbf{x}, y)$ denote the stochastic gradient noise, i.e.,:

$$\mathbf{n}(\theta; \mathbf{x}, y) = g(\theta; \mathbf{x}, y) - \nabla F(\theta)$$
$$= \phi(\langle \mathbf{x}, \theta \rangle)\mathbf{x} - \mathbb{E}\left[\phi(\langle \mathbf{x}, \theta \rangle)\mathbf{x}\right] + \mathbb{E}[\phi(\langle \mathbf{x}, \theta^* \rangle)\mathbf{x}] - y\mathbf{x}$$

We shall now control the stochastic gradient covariance $\Sigma(\theta) = \mathbb{E}[\mathbf{n}(\theta; \mathbf{x}, y)\mathbf{n}(\theta; \mathbf{x}, y)^T]$. To this end, we define $\mathbf{a_x}(\theta)$ and $\mathbf{c_{x,y}}(\theta)$ as follows:

$$\mathbf{a_x}(\theta) = \phi(\langle \mathbf{x}, \theta \rangle)\mathbf{x} - \mathbb{E}\left[\phi(\langle \mathbf{x}, \theta \rangle)\mathbf{x}\right]$$
$$\mathbf{c_{x,y}}(\theta) = \mathbb{E}[\phi(\langle \mathbf{x}, \theta^* \rangle)\mathbf{x}] - y\mathbf{x}$$

We note that $\mathbb{E}\left[\mathbf{c_{x,y}}(\theta)|\mathbf{x}\right] = 0$ and $\mathbb{E}[\mathbf{a_x}(\theta)] = 0$. Since $\mathbf{n_{x,y}}(\theta) = \mathbf{a_x}(\theta) + \mathbf{b_{x,y}}(\theta)$, it follows that:

$$\Sigma(\theta) == \mathbb{E}[\mathbf{n}(\theta; \mathbf{x}, y)\mathbf{n}(\theta; \mathbf{x}, y)^T] = \mathbb{E}\left[\mathbf{a_x}(\theta)\mathbf{a_x}(\theta)^T\right] + \mathbb{E}\left[\mathbf{c_{x,y}}(\theta)\mathbf{c_{x,y}}(\theta)^T\right]$$

We now control each of the terms in the RHS as follows:

$$\mathbb{E}\left[\mathbf{a_x}(\theta)\mathbf{a_x}(\theta)^T\right] = \mathbb{E}[\phi(\langle \mathbf{x}, \theta \rangle)^2\mathbf{x}\mathbf{x}^T] - \mathbb{E}\left[\phi(\langle \mathbf{x}, \theta \rangle)\mathbf{x}\right]\mathbb{E}\left[\phi(\langle \mathbf{x}, \theta \rangle)\mathbf{x}\right]^T$$
$$\preceq \mathbb{E}[\phi(\langle \mathbf{x}, \theta \rangle)^2\mathbf{x}\mathbf{x}^T]$$
$$\preceq \mathbb{E}[\mathbf{x}\mathbf{x}^T] \preceq \Sigma$$

where we use the fact that $\phi(t) \le 1$. Similarly,

$$\mathbb{E}\left[\mathbf{c_{x,y}}(\theta)\mathbf{c_{x,y}}(\theta)^T\right] = \mathbb{E}[y^2\mathbf{x}\mathbf{x}^T] - \mathbb{E}[\phi(\langle \mathbf{x}, \theta^* \rangle)\mathbf{x}]\mathbb{E}[\phi(\langle \mathbf{x}, \theta^* \rangle)\mathbf{x}]^T$$
$$\preceq \mathbb{E}[\phi(\langle \mathbf{x}, \theta^* \rangle)\mathbf{x}\mathbf{x}^T]$$
$$\preceq \mathbb{E}[\mathbf{x}\mathbf{x}^T] \preceq \Sigma$$

where we use the fact that $\mathbb{E}[y^2|\mathbf{x}] = \phi(\langle \mathbf{x}, \theta^* \rangle) \le 1$. It follows that

$$\Sigma(\theta) \preceq 2\Sigma$$

Thus, the stochastic gradient oracle satisfies the Bdd. 2$^{\text{nd}}$ Moment assumption. Hence, the stochastic gradient oracle satisfies the Bdd. 2$^{\text{nd}}$ Moment assumption. Thus, the following result, which is a formal version of Corollary 3, is implied by Theorem 7

**Corollary 7** (Heavy Tailed Logistic Regression). *Under the stochastic subgradient oracle described above, realized using $N \gtrsim \ln(\ln(d))$ i.i.d samples from $P$, the average iterate of Algorithm 1, when run under the parameter settings of Theorem 4 satisfies the following with probability at least $1 - \delta$:*

$$F(\hat{\theta}_N) - F(\theta^*) \lesssim D_1\sqrt{\frac{\mathsf{Tr}(\Sigma) + \sqrt{\|\Sigma\|_2}\left(\sqrt{\mathsf{Tr}(\Sigma)} + \|\Sigma\|_2 D_1\right)\ln(\ln(N)/\delta)}{N}} + \frac{\|\Sigma\|_2 D_1^2}{N}$$
$$+ \frac{D_1^2 \ln(\ln(N)/\delta)}{N}\sqrt{\|\Sigma\|_2\mathsf{Tr}(\Sigma) + \|\Sigma\|^3 D_1^2} + \frac{\|\Sigma\|_2^{5/4} D_1^3 \ln(\ln(N)/\delta)^{3/2}}{N^{3/2}}\left(\mathsf{Tr}(\Sigma) + \|\Sigma\|_2^2 D_1^2\right)^{1/4}$$

### G.4 Proof of Corollary 4

Recall from Section 5.4 that $\Xi = \mathbb{R}^d \times \mathbb{R}$ and given a target parameter $\theta^* \in \mathcal{C}$, $P$ defines the following linear model:

$$\mathbf{x} \sim Q, \ \mathbb{E}[\mathbf{x}] = 0, \ \mathbb{E}[\mathbf{x}\mathbf{x}^T] \preceq \Sigma; \qquad y = \langle \mathbf{x}, \theta^* \rangle + \epsilon, \ \mathsf{Median}(\epsilon|\mathbf{x}) = 0$$

We allow both the covariate $\mathbf{x}$ and target $y$ to be heavy tailed, assuming only bounded second moments for $\mathbf{x}$. We do not assume any moment bounds on $\epsilon|\mathbf{x}$. The Least Absolute Deviation (LAD) Regression problem involves estimating $\theta$ by solving SCO with the following sample loss

$$f(\theta; \mathbf{x}, y) = |\langle \mathbf{x}, \theta \rangle - y|$$

The associated population risk and one possible realization of a stochastic subgradient oracle is given by:

$$F(\theta) = \mathbb{E}\left[|\langle \theta - \theta^*, \mathbf{x} \rangle - \epsilon|\right]$$
$$g(\theta; \mathbf{x}, \mathbf{y}) = \mathsf{sgn}(\langle \theta, \mathbf{x} \rangle - \mathbf{y})\mathbf{x}$$

where $\mathsf{sgn}(t) = \frac{t}{\|t\|}$ for $t \neq 0$ and $\mathsf{sgn}(0) = 0$. We note that for every $(\mathbf{x}, \mathbf{y}) \in \mathbb{R}^d \times \mathbb{R}$, $f(\theta; \mathbf{x}, y)$ is a convex function in $\theta$, and thus, the population risk $F$ is a convex function, whose subgradient is given by:

$$\partial F(\theta) = \mathbb{E}\left[\mathsf{sgn}\left(\langle \theta - \theta^*, \mathbf{x} \rangle - \epsilon\right)\mathbf{x}\right]$$

We now show that $F$ is a Lipschitz function by bounding $\partial F(\theta)$ as follows:

$$\begin{aligned}
\|\partial F(\theta)\| &= \|\mathbb{E}\left[\mathsf{sgn}\left(\langle \theta - \theta^*, \mathbf{x} \rangle - \epsilon\right)\mathbf{x}\right]\| \\
&\leq \mathbb{E}\left[|\mathsf{sgn}\left(\langle \theta - \theta^*, \mathbf{x} \rangle - \epsilon\right)| \cdot \|\mathbf{x}\|\right] \\
&\leq \sqrt{\mathbb{E}\left[\|\mathbf{x}\|^2\right]} \\
&\leq \sqrt{\mathsf{Tr}(\Sigma)}
\end{aligned}$$

where the second step follows from Jensen's inequality, the third step uses the fact that $|\mathsf{sgn}(t)| \leq 1$ and applies the Cauchy Schwarz inequality. Hence, $F$ is $G$-Lipschitz with $G = \sqrt{\mathsf{Tr}(\Sigma)}$. We now show that $\partial F(\theta^*) = 0$ which would imply that $\theta^*$ is a minimizer of $F$ (as $F$ is convex)

$$\nabla F(\theta^*) = \mathbb{E}\left[\mathsf{sgn}(\epsilon)\mathbf{x}\right] = \mathbb{E}\left[\mathbf{x} \cdot \mathbb{E}\left[\mathsf{sgn}(\epsilon)|\mathbf{x}\right]\right] = 0$$

where we use the fact that $\mathbb{E}[\mathsf{sgn}(\epsilon)|\mathbf{x}] = 0$, because $\epsilon|\mathbf{x}$ is a continuous random variable with zero median.

For the stochastic gradient oracle described above, the associated stochastic gradient noise $\mathbf{n}(\theta; \mathbf{x}, y)$ and its covariance $\Sigma(\theta)$ are given as follows:

$$\mathbf{n}(\theta; \mathbf{x}, y) = \mathsf{sgn}(\langle \theta - \theta^*, \mathbf{x} \rangle - \epsilon)\mathbf{x} - \mathbb{E}\left[\mathsf{sgn}(\langle \theta - \theta^*, \mathbf{x} \rangle - \epsilon)\mathbf{x}\right]$$

$$\begin{aligned}
\Sigma(\theta) &= \mathbb{E}\left[\mathsf{sgn}(\langle \theta - \theta^*, \mathbf{x} \rangle - \epsilon)^2\mathbf{x}\mathbf{x}^T\right] - \mathbb{E}\left[\mathsf{sgn}(\langle \theta - \theta^*, \mathbf{x} \rangle - \epsilon)\mathbf{x}\right]\mathbb{E}\left[\mathsf{sgn}(\langle \theta - \theta^*, \mathbf{x} \rangle - \epsilon)\mathbf{x}\right]^T \\
&\preceq \mathbb{E}\left[\mathsf{sgn}(\langle \theta - \theta^*, \mathbf{x} \rangle - \epsilon)^2\mathbf{x}\mathbf{x}^T\right] \\
&\preceq \mathbb{E}\left[\mathbf{x}\mathbf{x}^T\right] \preceq \Sigma
\end{aligned}$$

Hence, the stochastic gradient oracle satisfies the Bdd. 2$^{\text{nd}}$ Moment assumption. Thus, the following result, which is a formal version of Corollary 4, is implied by Theorem 8

**Corollary 8** (Heavy Tailed LAD Regression)**.**

$$F(\hat{\theta}_N) - F(\theta^*) \lesssim D_1 \sqrt{\frac{\mathsf{Tr}(\Sigma) + \sqrt{\|\Sigma\|_2 \mathsf{Tr}(\Sigma)} \ln\left(\ln(N)/\delta\right)}{N}} + \frac{D_1 \mathsf{Tr}(\Sigma) \ln\left(\ln(N)/\delta\right)}{N\sqrt{\|\Sigma\|_2}} + \frac{D_1 \mathsf{Tr}(\Sigma)^{5/4} \ln\left(\ln(N)/\delta\right)^{3/2}}{N^{3/2}\|\Sigma\|^{3/4}}$$

