# OpenReview forum: "Near-Optimal Streaming Heavy-Tailed Statistical Estimation with Clipped SGD"
_NeurIPS.cc/2024/Conference — NeurIPS 2024 poster_

### Official Review · Reviewer_cbSj · 2024-06-22

**Soundness:** 3
**Presentation:** 3
**Contribution:** 3
**Rating:** 7
**Confidence:** 3

**Summary:**

The paper gives a new concentration method to improve the bounds for the convergence of clipped-SGD when the noise has bounded second moments in the online/streaming setting. The method has also applications in streaming heavy-tailed statistical estimation, including streaming mean estimation and regression. The main idea of the method is by bootstrapping Freedman's inequality via iterative refinement using a PAC Bayesian argument. This new concentration method allows to obtain better bounds, closer to the optimal bounds than previous methods.

**Strengths:**

- Overall, I think the paper has made a good contribution in sharpening known bounds for heavy-tailed clipped-SGD. The idea of bootstrapping Freedman's inequality via PAC Bayesian is very interesting and I agree with the authors that it would be interesting to further investigate the applications of this argument.
- The proposed method has a wide range of applications and the authors have presented them quite thoroughly.

**Weaknesses:**

- The second moment bound assumption used in this paper appears to be stronger than the variance bound used in prior work for analyzing clipped-SGD. My understanding, for example from the work of Gorbunov et al, is that with the bounded variance assumption they use the simple scalar version of Freedman's so the dependence on the dimension might not be as good as stated here in the paper. It would be good if the authors can discuss this more clearly, for example what if we just use the high dimension version of Freedman's?
- Another issue with current assumption 1 is that it seems not easy to extend the work to the bounded $p$-moment case, as we also know that we can do this very easily from the work of Gorbunov et al and follow up works
- The clipping method requires to know the time horizon, which we know is unnecessary in some prior work, eg, Nguyen et al. The bounds are also sub-optimal in $T$ (with an extra $\log\log T$ factor).
- In terms of presentation, I think proofs, especially section F, can be written in a more streamlined way  and for example, with an outline.
- Some minor details
Line 82: There should be a square root in the bound
Line 106: We not(e) that

Reference

E. Gorbunov, M. Danilova, and A. Gasnikov. Stochastic optimization with heavy-tailed noise via accelerated gradient clipping, NeuRIPS, 2020

T. D. Nguyen, T. H. Nguyen, A. Ene, and H. Nguyen.  Improved Convergence in High Probability of Clipped Gradient Methods with Heavy Tailed Noise, NeuRIPS, 2023

**Questions:**

- I'm a bit confused about the result in table 1. I didn't do the calculation myself but the bound shown in the last line mentioned both $T$ and $\epsilon$, which is odd; and the dependence $\log^2 1/\delta$ in the lower order term is also higher than other bounds. Could the authors explain?

**Limitations:**

The authors have adequately addressed the limitations of the paper.

---

> ### Author Rebuttal · Authors · 2024-08-06
>
> $\newcommand{\Tr}{\mathsf{Tr}} \newcommand{\deff}{d_{\mathsf{eff}}} \newcommand{\vx}{\mathbf{x}} \newcommand{\vz}{\mathbf{z}} \newcommand{\vw}{\mathbf{w}} \newcommand{\vn}{\mathbf{n}} \newcommand{\bE}{\mathbb{E}} \newcommand{\bR}{\mathbb{R}} \newcommand{\dotp}[2]{\left\langle #1, #2 \right \rangle} \newcommand{\vv}{\mathbf{v}} \newcommand{\vS}{\mathbf{S}}$ Thank you for your helpful feedback. We hope to address your concerns below:
> ### Assumption 1 and Gorbunov et. al. [3]
>
> Assumption 1 of our work is actually *equivalent to & more fine-grained than* the bounded second moment assumption of [3]. Here $\Tr(\Sigma)$ is the same as $\sigma^2$ in [3], as explained below. Assumption 1 is more fine-grained as it allows us to obtain bounds in terms of both $\Tr(\Sigma)$ and $\||\Sigma\||_2$.
>
> Let $\vz \in \bR^d$ be a zero-mean random vector satisfying $\bE[\||\vz\||^2] \leq \sigma^2$ for some $\sigma \geq 0$ . By linearity of expectation, $\Tr(\bE[\vz \vz^T]) = \bE[\||\vz\||^2] \leq \sigma^2$. Hence, $\bE[\vz \vz^T]$ is a well-defined PSD matrix with $\Tr(\bE[\vz \vz^T]) \leq \sigma^2$. It follows that there exists some PSD matrix $\Sigma$ satisfying $\bE[\vz \vz^T] \preceq \Sigma$ such that $\Tr(\Sigma) \leq \sigma^2$. The converse is argued similarly.
>
> ### Iterative Refinement vs High Dimensional Freedman
>
> As discussed in Section 6, a standalone application of the high-dimensional Freedman's inequality (i.e., Matrix Freedman) in conjunction with Assumption 1, is by itself insufficient to obtain the near-optimal rate, and instead implies a suboptimal rate of $\sqrt{\frac{\Tr(\Sigma) \ln(d/\delta)}{T}}$. This necessitates the development of our iterative refinement technique, which recursively improves the crude bound implied by Matrix Freedman, boosting it to a near-optimal rate of $\sqrt{\frac{\Tr(\Sigma) + \sqrt{\||\Sigma\||_2 \Tr(\Sigma)}\ln(\ln(T)/\delta)}{T}}$. We have updated our draft to emphasize this point more explicitly in our proof outline.
>
> ### Extending Assumption 1
>
> Assumption 1 can be extended to bounded $p^{\mathsf{th}}$ moments for any $p > 1$ by assuming that there exists a PSD matrix $\vS$ satisfying $\bE_{\xi \sim P}[|\dotp{\vn(\vx;\xi)}{\vv}|^p] \leq (\vv^T \vS \vv)^{p/2} \ \forall \ \vv \in \bR^d$. We note that this generalization recovers Assumption 1 for $p = 2$ and is similar to the weak moment assumption considered by [1], which, to our knowledge, is the only work that analyzes statistical rates for heavy tailed mean estimation under bounded $p^{\mathsf{th}}$ moments ($p \leq 2$).
>
> We also note that beyond the specific case of mean estimation studied in [1], very little is known about the optimal statistical rates of heavy-tailed estimation under bounded $p^{\mathsf{th}}$ moments, even in the full batch setting. Thus, it is unknown whether existing works that analyze clipped SGD and its variants under bounded $p^{\mathsf{th}}$ moments achieve statistical optimality.
>
> ### Known Time Horizon
>
> Our assumption of a known time horizon was primarily made for the sake of clarity, and motivated by the fact that the time horizon / sample budget is typically known apriori in most ML and statistics applications. We believe our results can be extended to the anytime setting by using a time-varying clipping level of the form $\lambda_t = \Theta(\sqrt{t})$ (similar to [2]). However, incorporating this adjustment would significantly increase the length and technical complexity of our proofs, and subsequently obscure our main contributions. To this end, our results are presented assuming a fixed time horizon, similar to prior works on heavy tailed stochastic optimization [3,4].
>
> ### Extra $\ln(\ln(T))$ Term
>
> We emphasize that the extra  $\ln(\ln(T))$ term is not a major weakness of our work since our obtained rates continue to significantly outperform the previous state of the art in all practical regimes. To observe this, let $\deff = \tfrac{\Tr(\Sigma)}{\||\Sigma\||_2}$  Note that $1 \leq \deff \leq d$. Furthermore, $\forall \delta \in (0,1)$ and $T \leq e^{\exp({(\sqrt{\deff}-1)\ln(1/\delta)})}$ , we have $\sqrt{\deff} \ln(\ln(T)/\delta) \leq \deff \ln(1/\delta)$. Hence, our obtained rate of $\sqrt{\frac{\Tr(\Sigma) + \sqrt{\||\Sigma\||_2 \Tr(\Sigma)} \ln(\ln(T)/\delta)}{T}}$ significantly outperforms the previous best known rate of $\sqrt{\frac{\Tr(\Sigma)\ln(1/\delta)}{T}}$ unless the time-horizon / sample budget $T$ exceeds $ e^{\exp({(\sqrt{\deff}-1)\ln(1/\delta)})}$, which is quite impractical (as it involves a double exponential dependence on $\deff$)
>
> ### Proof Outline
>
>  Thank you for this helpful pointer. We have updated our draft to add a brief proof outline for each of our key results in the appropriate sections of the Appendix. Furthermore, in accordance with Reviewer 86zv's feedback,  our updated draft contains an expanded proof sketch section at the beginning of the Appendix, which explains our iterative refinement technique more clearly, and also describes how its applied to prove Theorem 3 (clipped SGD for smooth convex objectives).
>
> ### Table 1
>
> Thanks for pointing out this typo. The sample complexity bound in the last row of the table is derived from Equation 1; it is obtained by finding the $T$ for which the error in function value is equal to $\epsilon$. The $\ln(T)$ term in the table should be replaced by $\ln(1/\epsilon),$ which makes the lower order (in $\epsilon$) term $\frac{D_1\ln^2(\delta^{-1}\ln{1/\epsilon})}{\sqrt{\epsilon}}$
>
> ### Lines 82 and 106
>
> Thanks for pointing these out. Our updated draft corrects these typos.
>
> ***
> 1. Cherapanamjeri et. al., Optimal Mean Estimation without a Variance, COLT 2022
> 2. Nguyen et. al., Improved Convergence in High Probability of Clipped Gradient Methods with Heavy Tails, NeurIPS 2023
> 3. Gorbunov et. al., Stochastic Optimization with Heavy Tailed Noise via Accelerated Gradient Clipping, NeurIPS 2020
> 4. Tsai et. al., Streaming Heavy Tailed Statistical Estimation, AISTATS 2022

---

> > ### Comment · Reviewer_cbSj · 2024-08-13
> >
> > I thank the authors for the thoughtful response. I hope to see the above discussion in the paper. I maintain my current score.

---

### Official Review · Reviewer_rk1f · 2024-06-30

**Soundness:** 3
**Presentation:** 3
**Contribution:** 2
**Rating:** 5
**Confidence:** 3

**Summary:**

This paper addresses the problem of high-dimensional heavy-tailed statistical estimation in a streaming setting, which is more challenging than the batch setting due to memory constraints.

The authors cast the problem as stochastic convex optimization (SCO) with heavy-tailed stochastic gradients.

They demonstrate that the Clipped Stochastic Gradient Descent (Clipped-SGD) algorithm attains near-optimal sub-Gaussian statistical rates when the second moment of the stochastic gradient noise is finite.

The rate is better than the previous result in terms of the fluctuation term that depends on $1/\delta$ in the smooth and strongly convex case and fills the blank for smooth convex and Lipschitz convex cases.

**Strengths:**

1. **Improved Error Bounds**: The paper improves the known error bounds for Clipped-SGD, bringing them closer to the optimal sub-Gaussian rates.

2. **Extension to Various Objectives**: The results are extended to smooth convex and Lipschitz convex objectives, broadening the applicability of Clipped-SGD to a wider range of optimization problems.

3. **Novel Iterative Refinement Strategy**: The introduction of an iterative refinement strategy for martingale concentration is new to me. It might inspire similar techniques.

4. **Good Clarity**: The paper is overall well-written and easy to follow.

**Weaknesses:**

1. **Missing Reference**: There is one paper from my perspective that is related but not cited.

2. **How to Generalize the Established Results**: The paper contains many new theoretical results. It is unclear how others could use these proposed techniques for their interested cases. There are also some cases that the author didn't mention. See the Questions for the details.

**Questions:**

1. **Missing Reference**: The paper should include a citation to [1*], which modifies adaptive Huber regression for online bandit and MDP with heavy-tailed rewards. This work is related and should be cited in the "Heavy-tailed Estimation" part of Section 1.2.

    - [1*] Li, Xiang, and Qiang Sun. "Variance-aware decision making with linear function approximation under heavy-tailed rewards." Transactions on Machine Learning Research.

2. **Extension to Strongly Convex and Lipschitz Cases**: Is it possible to extend the results to the strongly convex and Lipschitz case? If so, how would these results differ from those already derived in this paper?

3. **Optimization Error and Statistical Error**: The error rate (e.g., $\|x_{T+1}-x^{\star}\|$ in Theorem 1) typically comprises two parts: the optimization error (i.e., $\frac{\gamma D_1}{T + \gamma}$ in (1)) and the statistical error (the other term in (1)). The paper's most significant contribution is on the statistical error, ensuring it behaves like a sub-Gaussian rate even when the gradient noise has only bounded variance. Two questions arise:
   - Can the optimization error be accelerated using the techniques developed in this paper to handle heavy-tailed issues? This might be achieved by another algorithm. I just wonder whether the analysis technique, i.e., the iterative refinement strategy, could be used to tighten the analysis on the statistical error.
   - Can the iterative refinement strategy be extended to cases with bounded $1+\delta_0$ moments, where $\delta_0 \in (0, 1]$, rather than just bounded variance?

4. **Comparison to Freedman's and Bernstein’s Inequalities**: The improvement in this paper is analogous to the improvement from Hoeffding's inequality to Bernstein’s inequality. However, to account for the Markov structure, Freedman’s inequality and the iterative refinement strategy are used to decouple dependencies. Is this understanding correct?

**Limitations:**

See the Questions.

---

> ### Author Rebuttal · Authors · 2024-08-06
>
> $\newcommand{\Tr}{\mathsf{Tr}}$ Thank you for your helpful feedback. We hope to address your concerns below:
>
> ### Missing Reference
>
> Thank you for pointing out this helpful reference. The work is indeed quite relevant and we have updated our draft to include the citation in Section 1.2.
>
> ### Extension to Lipschitz Strongly Convex Problems
>
> Indeed, our proof techniques can be applied to Lipschitz Strongly Convex problems to obtain a guarantee similar to Theorem 1 for the weighted average iterate. However, due to space constraints, we decided to focus on results that directly led to interesting statistical applications such as mean estimation, linear regression, binary classification and LAD regression.
>
> ### Improving the Optimization Error
>
> Indeed, we believe our iterative refinement strategy can be used to analyze accelerated stochastic optimization algorithms (such as the clipped SSTM algorithm of [1]) to obtain an improved optimization error rate. Accelerating the optimization error rate whilst simultaneously obtaining a (near)-optimal statistical rate is an interesting avenue of future work which we intend to look into.
>
> ### Bounded $1 + \delta_0$ Moments
>
> We believe our iterative refinement technique is quite general and can be extended to handle bounded $1 + \delta_0$ moments for $\delta_0 \in (0, 1]$ by carefully modifying the MGF bounds involved in the proof of Theorem 5, along with a re-derivation of the Martix Freedman inequality with $1+\delta_0$ moment. This is a very promising avenue of future work which we are currently looking into.
>
> We highlight that, unlike the well-studied case of bounded second moments, very little is known about the optimal statistical rates for heavy tailed estimation tasks under bounded $1 + \delta_0$ moments, *even in the full batch setting*. To the best of our knowledge, [2] is the only work that analyzes statistical rates in this framework (under the full batch setting) for the specific task of mean estimation. It is still unknown whether existing works that analyze clipped SGD and its variants under bounded $1 + \delta_0$ moments achieve statistical optimality.
>
> ### Comparison to Freedman and Bernstein Inequalities
>
> To the best of our understanding, the improvement obtained via our iterative refinement technique actually surpasses the typical Hoeffding-to-Bernstein improvement. To observe this, we note the following:
>
>  -  Transitioning from Hoeffding to Bernstein leads to a sharper "variance proxy" (i.e. it gives us sharper concentration inequalities where the typical fluctuation is of the order of the variance instead of the almost sure bound on the random variable), which is necessary for obtaining an improved statistical rate.
>
>  -  Freedman's Inequality leads to further improvement by allowing the martingale / Markov structure of the noise to be incorporated within Bernstein's Inequality.
>
>  -  However, we find that the above steps alone are insufficient for obtaining the desired near-optimal statistical rate. In particular, as discussed in Section 6, a direct use of (Matrix) Freedman's inequality implies a rate of $\sqrt{\frac{\Tr(\Sigma) \ln(d/\delta)}{T}}$ which is far from optimal. To overcome this, we develop the iterative refinement technique which recursively improves upon the coarse bound implied by Freedman's inequality. As a result, we finally obtain the desired near-optimal statistical rate of $\sqrt{\frac{\Tr(\Sigma) + \sqrt{\||\Sigma\||_2 \Tr(\Sigma)}\ln(\ln(T)/\delta)}{T}}$.
> ***
> ### References
>
> 1. Gorbunov et. al., Stochastic Optimization with Heavy Tailed Noise via Accelerated Gradient Clipping, NeurIPS 2020.
>
> 2. Cherapanamjeri et. al., Optimal Mean Estimation without a Variance, COLT 2022

---

### Official Review · Reviewer_86zv · 2024-07-07

**Soundness:** 3
**Presentation:** 2
**Contribution:** 4
**Rating:** 6
**Confidence:** 3

**Summary:**

The paper introduces significant advancements in the field of statistical estimation and optimization for heavy-tailed data. It improves the convergence complexity of the widely-used clipped-SGD algorithm in both strongly convex and general convex settings, achieving near-optimal sub-Gaussian statistical rates when the second moment of the stochastic gradient noise is finite.

The paper also introduces refined concentration guarantees for vector-valued martingales using the Donsker-Varadhan Variational Principle, enhancing the PAC-Bayes bounds of Catoni and Giulini.
Additionally, it provides a fine-grained analysis of clipped SGD for heavy-tailed SCO problems, achieving nearly subgaussian performance in a streaming setting with improved complexity over previous work.
Furthermore, the paper develops streaming estimators for various heavy-tailed statistical problems, including mean estimation and regression (linear, logistic, and LAD), all exhibiting nearly subgaussian performance and improving upon prior guarantees.

**Strengths:**

The paper presents several noteworthy contributions to the field of statistical estimation and optimization for heavy-tailed data, particularly in improving complexity.

The paper introduces novel improvements to the convergence complexity of the clipped-SGD algorithm, applicable to both strongly convex and general convex settings, as well as various assumption combinations. Achieving near-optimal sub-Gaussian statistical rates is a significant advancement in the field.
The technical rigor of the paper is evident in its thorough analysis and detailed proofs, with over 40 pages in the appendix. The authors systematically build on existing literature, as summarized in Table 1, providing a clear and logical progression of their arguments.
The fine-grained analysis of clipped SGD for heavy-tailed SCO problems is meticulously executed, demonstrating nearly subgaussian performance in a streaming setting. This improvement over previous work indicates a high level of quality in the research methodology and execution.
The paper is well-organized and clearly written, making complex theoretical concepts accessible. The authors effectively explain the significance of their contributions and how they build upon existing work, enhancing overall readability.

The improvements to the clipped-SGD algorithm have broad implications for practical applications in machine learning and statistical estimation, particularly in scenarios involving heavy-tailed data distributions.
By developing streaming estimators for various heavy-tailed statistical problems, the paper addresses a critical need in the field, providing robust solutions that exhibit nearly subgaussian performance. This has the potential to significantly impact both theoretical research and real-world applications.

Overall, the paper excels in originality, quality, clarity, and significance, making substantial contributions to the field of statistical estimation and optimization for heavy-tailed data.

**Weaknesses:**

I am not an expert in this area and do not have time to check the proofs in detail. Therefore, my understanding relies solely on the content provided in the writing.
Despite its significant contributions, the paper has several areas that could benefit from improvement:
1. Lack of Technique-Proof Sketches in the Main Paper: While the paper presents numerous theorems, there is a noticeable absence of technique-proof sketches. These sketches would provide insights into how the authors build upon existing results compared to previous works. Including such sketches would enhance the clarity and transparency of the theoretical developments, making it easier for readers to grasp the novelty and progression of the contributions.
The current structure dedicates only half a page to showcasing the main technique in Section 6, which can be challenging for readers. A more effective approach might involve presenting one key theory with a detailed technique description in the main paper, and relegating supplementary theories to either the main paper with abbreviated details or the appendix.
2. Theoretical Overemphasis: The paper may overly focus on presenting multiple theories without sufficiently integrating them into a cohesive narrative or demonstrating their practical application. A clearer alignment of the theoretical developments with practical implications would strengthen the paper's impact and relevance.
For instance, Section 5, which features many applications, could benefit from highlighting some of them in the background to help readers understand the importance and scenarios of the problems studied.
3. Absence of Experimental Validation: Statistical estimation typically requires extensive data and iterations to observe convergence reliably. The lack of experimental validation in the paper limits its ability to demonstrate the practical effectiveness of the proposed algorithms.
Incorporating experiments with diverse datasets and comparisons with existing methods would provide empirical evidence of the proposed techniques' efficacy and robustness in real-world scenarios.

Addressing these points would enhance the paper’s clarity, relevance, and applicability, bridging the gap between theoretical developments and practical implementations in statistical estimation and optimization for heavy-tailed data.
I welcome further discussion on these points to ensure a comprehensive evaluation. Thanks!

**Questions:**

See the Weaknesses part.

**Limitations:**

The paper is theoretical, and I believe there is no need to confirm societal impact.

---

> ### Author Rebuttal · Authors · 2024-08-06
>
> Thank you for your helpful feedback. We hope to address your concerns below:
>
> ### Proof Sketches
>
> Thank you for this helpful pointer which has helped improve our work. We have updated our draft to add a detailed proof sketch in the beginning of our Appendix. This new section expands upon the proof sketch of the iterative refinement technique presented in Section 6, and also outlines a proof of Theorem 3 (i.e. clipped SGD for smooth convex objectives) by applyinh the iterative refinement technique. Since camera ready versions of accepted papers are allowed to have an extra page, we would be happy to shift this section to the main paper if our work is accepted.
>
> ### Theoretical Overemphasis
>
> Thank you for pointing this out. Based on your helpful feedback, we have updated our presentation in Section 1 to emphasize the statistical applications, and added references to Section 5 wherever appropriate.
>
> ### Empirical Validation
>
> We note that gradient clipping is a standard technique in machine learning whose empirical performance very well investigated. Indeed, gradient clipping is widely used in several empirical deployments such as LLMs, Vision Transformers, GANs and PPO. On the contrary, a sharp statistical analysis of gradient clipping which justifies its empirical effectiveness has been relatively limited. To this end, our work focuses on presenting a thorough analysis of the theory behind gradient clipping by sharply characterizing its performance on a wide variety of streaming heavy tailed statistical estimation tasks.

---

> > ### Comment · Reviewer_86zv · 2024-08-10
> >
> > Thank you for the response. I maintain my current evaluation of the paper as it is highly technical and covers areas I am unfamiliar with. I believe the paper could be improved based on the current modifications shown in the rebuttal. Good luck to you.

---

### Official Review · Reviewer_Wj6A · 2024-07-13

**Soundness:** 3
**Presentation:** 3
**Contribution:** 3
**Rating:** 6
**Confidence:** 3

**Summary:**

The paper studies the clipping SGD algorithm and shows a refined analysis to improve the dependence on the variance. The authors also provide different applications of their new results.

**Strengths:**

The new concentration bounds are interesting, which is the key novel part of the work.

**Weaknesses:**

1. The key drawback is that the algorithm still requires many problem-dependent parameters like $D_1$ and $\mathrm{Tr}(\Sigma)$, which are hard to know, especially in a streaming setting.

1. The time horizon $T$ is also requested to make the algorithm work. Is it possible to extend to the any-time setting?

1. There is an extra $\log{\log{T}}$ term in the bounds.

1. Please specify the stepsize $\eta$ in Theorems 3 and 4.

1. Line 82, missing a square root?

**Questions:**

See **Weaknesses**.

**Limitations:**

Not applicable.

---

> ### Author Rebuttal · Authors · 2024-08-06
>
> $\newcommand{\Tr}{\mathsf{Tr}} \newcommand{\deff}{d_{\mathsf{eff}}}$ Thank you for your insightful feedback. We hope to address your concerns below:
>
>
> ### Problem Dependent Parameters
>
> While our results assume knowledge of some problem-dependent quantities, we emphasize that such assumptions are **not unique to our work**. To the best of our knowledge, setting the step-size and related algorithmic parameters as a function of problem dependent quantities like $D_1$ and $\Tr(\Sigma)$ is a standard practice in the analysis of stochastic optimization algorithms [1; 2 Ch. 6 Thms 6.1 - 6.3]. Naturally, this is also prevalent in almost all prior works on clipped SGD [3, 4, 5]. For instance, [3, Thm 3.1] requires knowledge of $D_1$, $T$ and $\Tr(\Sigma)$ (corresponding to $R_0$, $N$ and $\sigma^2$ respectively as per their notation) to set the batch-size, clipping level and step-size for clipped SGD. Moreover, our results can be easily adapted to settings where only upper bounds of problem-dependent quantities are available while the precise values are unknown.
>
> We note that design of *parameter-free optimization algorithms* (i.e. algorithms that don't require prior knowledge of problem-specific parameters) is a sub-field of its own [6] involving significant technical challenges and algorithmic modifications that are orthogonal to the focus of our work. We believe that the techniques developed in our work could be used in conjunction with ideas from the parameter-free optimization literature to develop streaming parameter-free heavy tailed statistical estimators. Investigating this would be an interesting avenue of future work.
>
> ### Known Time Horizon
>
> Our assumption of a known time horizon was primarily made for the sake of clarity, and motivated by the fact that the time horizon (or equivalently, the sample budget) is typically known apriori in most ML and statistics applications (at least up to constant factors). We believe our results can be extended to the anytime setting by using a time-varying clipping level of the form $\lambda_t = \Theta(\sqrt{t})$ (similar to [5]). However, incorporating this adjustment would significantly increase the length and technical complexity of our proofs, and subsequently obscure our key technical contributions. To this end, our results are presented assuming a fixed time horizon, similar to prior works on heavy tailed stochastic optimization [3,4].
>
> ### Extra $\ln(\ln(T))$ Term
>
> We respectfully disagree with the claim that the extra  $\ln(\ln(T))$ term is a major weakness of our work. We emphasize that, despite this extra term, our obtained rates continue to **significantly outperform** the previous state of the art in all practical regimes. To observe this, let $\deff = \tfrac{\Tr(\Sigma)}{\||\Sigma\||_2}$ where $\Sigma$ is the noise covariance. Note that $1 \leq \deff \leq d$. Furthermore, $\forall \delta \in (0,1)$ and $T \leq e^{\exp({(\sqrt{\deff}-1)\ln(1/\delta)})}$ , we have $\sqrt{\deff} \ln(\ln(T)/\delta) \leq \deff \ln(1/\delta)$. Hence, our obtained rate of $\sqrt{\frac{\Tr(\Sigma) + \sqrt{\||\Sigma\||_2 \Tr(\Sigma)} \ln(\ln(T)/\delta)}{T}}$ significantly outperforms the previous best known rate of $\sqrt{\frac{\Tr(\Sigma)\ln(1/\delta)}{T}}$ unless the time-horizon / sample budget $T$ exceeds $ e^{\exp({(\sqrt{\deff}-1)\ln(1/\delta)})}$, which is quite impractical (as it involves a *double exponential* dependence on the effective dimension)
>
> ### $\eta$ in Theorems 3 and 4
>
> Thank you for your feedback. We have updated our draft to specify $\eta$ in the statement of Theorems 3 and 4.
>
> ### Line 82
>
> Thank you for the pointer. Line 82 is indeed missing a square root and we have updated our draft to rectify this.
>
> ***
> ### References
>
> 1. Chi Jin, Optimization for ML Lecture Notes :  [URL](https://drive.google.com/file/d/1BKqs34avawbcw7WDWgJpq-xkHxNEL4c5/view)
> 2. Sebastien Bubeck, Convex Optimization : Algorithms and Complexity
> 3. Gorbunov et. al., Stochastic Optimization with Heavy Tailed Noise via Accelerated Gradient Clipping, NeurIPS 2020
> 4. Tsai et. al., Streaming Heavy Tailed Statistical Estimation, AISTATS 2022
> 5. Nguyen et. al., Improved Convergence in High Probability of Clipped Gradient Methods with Heavy Tails, NeurIPS 2023
> 6. Orabona and Cutkosky, ICML 2020 Tutorial on Parameter Free Online Learning : [URL](https://parameterfree.com/icml-tutorial/)

---

> ### Comment · Reviewer_Wj6A · 2024-08-10
>
> I first thank the author's detailed response.
>
> **Problem-dependent parameters and $T$.** I understand some existing works require these conditions. However, doing research needs to explore new possibilities instead of sticking to old ways. More importantly, as far as I know, at least two different ways can partially remove them at least in the non-smooth and convex case.
>
>   1. Parameter-free algorithms from the online learning community: For example, see [1], which can remove the dependence on $D_1$. However, as mentioned by the authors, I agree this may require additional algorithmic modifications.
>
>   2. Recent advances in stochastic optimization: Another simpler way is to combine the DoG technique proposed by [2]. For example, see [3], which not only removes $D_1$ but also removes $T$ in a simpler way.
>
> Hence, I highly encourage the authors to consider how to remove these parameters (even partially), which can significantly improve the quality of the paper.
>
> In addition, the authors mentioned that the results can be easily adapted to settings where only the upper bounds of problem-dependent quantities are known. However, I am not very convinced since it is hard to imagine that this can be guaranteed in a streaming setting. Could you provide some concrete examples?
>
> **Extra $\log\log T$.**
>
>   1. I clarify that I didn't claim this is a major weakness. Instead, I put it in the weakness only because it is undesired and suboptimal. I believe the final goal should always be to find the optimal bound.
>
>   2. Moreover, I also would like to point out that the optimal rate without any extra logarithmic has already been achieved in prior works like [4] (which is even mentioned by the authors) and [3]. But I understand the double $\log$ factor here may appear due to different reasons, which could require different ways to remove. However, as discussed above, this is still a weakness in my opinion.
>
>   3. I appreciate the authors' discussion on this weakness, which I recommend adding as a remark in the paper.
>
> **References**
>
> [1] Zhang, J., & Cutkosky, A. (2022). Parameter-free regret in high probability with heavy tails. Advances in Neural Information Processing Systems, 35, 8000-8012.
>
> [2] Ivgi, M., Hinder, O., & Carmon, Y. (2023, July). Dog is sgd’s best friend: A parameter-free dynamic step size schedule. In International Conference on Machine Learning (pp. 14465-14499). PMLR.
>
> [3] Liu, Z., & Zhou, Z. (2023). Stochastic Nonsmooth Convex Optimization with Heavy-Tailed Noises: High-Probability Bound, In-Expectation Rate and Initial Distance Adaptation. arXiv preprint arXiv:2303.12277.
>
> [4] Nguyen, T. D., Nguyen, T. H., Ene, A., & Nguyen, H. (2023). Improved convergence in high probability of clipped gradient methods with heavy tailed noise. Advances in Neural Information Processing Systems, 36, 24191-24222.

---

> ### Author Response · Authors · 2024-08-13
> **Some notes regarding parameter free algorithms**
>
> Thank you for your feedback about removing problem dependent parameters. Obtaining fully parameter free optimization algorithms would be a great direction for future research and we will follow the helpful references provided by the reviewer. Below, we note some additional complexities which arise in the context of our work and argue that this requires additional techniques, beyond the scope of our work..
>
> ### Parameter Dependence on $D_1$ and $T$:
>
> The reviewer points to the work [3] as an example to improve parameter dependence in the algorithm. We want to note that [3, Theorem 1] which removes dependence on $T$ in the parameters achieves a sub-optimal bound with extra $\log^2 T$ terms, whereas [3, Theorem 2] which has $T$ dependent parameters removes these extra factors. Since our sharp analysis is meant to remove such extra logarithmic factors from the leading order term, we believe a straightforward adaptation of such techniques might not work in our case.
>
> Note that [3] considers a fixed upper bound on the stochastic gradient noise. However, Assumption 2 in our work allows for noise whose covariance can grow with distance from the optimum, which is necessary for important applications like linear regression. Thus, removing the dependence on the initial distance in our case might require additional technical considerations.
>
>
> ### Regarding $\log \log T$ factors:
> Thank you for the clarification. After a careful examination of our technical results, removing the $\log \log T$ factors does not seem straightforward. We also note that the results given in [4, Theorem 3.1] and [3, Theorem 3] show that they achieve rates with a dependence of $d\log(1/\delta)$. We compare this to $d + \sqrt{d}\log(\tfrac{\log T}{\delta})$ dependence which we achieve in our work. Note that  $\sigma^2$ in these works corresponds to $\mathsf{Tr}(\Sigma)$ in our work which we take to be $\Theta(d)$ for the sake of comparison. Thus our $\log\log T$ dependence is in the lower order term.

---

> > ### Comment · Reviewer_Wj6A · 2024-08-13
> >
> > I thank the authors for further discussion. Though some parts of the work are suboptimal and undesirable in my view, I think the refined analysis for SGD could potentially bring new insights into the optimization community. As such, I would increase my score to 6.

---

> > > ### Author Response · Authors · 2024-08-14
> > > **Thank you**
> > >
> > > Thank you very much. We hope to work on the directions pointed out by the reviewer in the future.

---

### Decision · Program_Chairs · 2024-09-25

**Decision:**

Accept (poster)

**Comment:**

The paper studies the streaming heavy-tailed statistical estimation problem. The authors cast the problem as stochastic convex optimization with heavy-tailed stochastic gradients and prove that the clipped-SGD algorithm attains near-optimal sub-Gaussian statistical rates whenever the second moment of the stochastic gradient noise is finite. The main idea of the method is an iterative refinement for martingale concentration. All the reviewers are positive (one suggests accept, two weak accept, and one borderline accept). I decide to accept the paper.